# Steady-State Behavior of Constant-Stepsize Stochastic Approximation: Gaussian Approximation and Tail Bounds

Yuyang Wang [1]  Felix Wang [2]  Zedong Wang [3]  Ijay Narang [4]  Yuzhou Wang [2]  Siva Theja Maguluri [3]

## Abstract

Constant-stepsize stochastic approximation is widely used in learning for computational efficiency. For a fixed stepsize, the iterates typically admit a stationary distribution that is rarely tractable. Prior work shows that as the stepsize $\alpha \downarrow 0$, the centered-and-scaled steady state converges weakly to a Gaussian random vector. However, for fixed $\alpha$, this weak convergence offers no usable error bound for approximating the steady-state by its Gaussian limit. This paper provides explicit, non-asymptotic error bounds for fixed $\alpha$. We study (i) stochastic gradient descent on smooth strongly convex objectives, (ii) linear SA, and (iii) contractive nonlinear SA, and we treat both i.i.d. and Markovian noise models to ensure broad applicability. Our main results first give dimension- and stepsize-dependent, explicit bounds in Wasserstein distance between the centered-scaled steady state and its Gaussian limit, with errors that vanish as $\alpha \downarrow 0$. We further derive sharp tail control, comparing the steady-state tail probability to Gaussian tails with an explicit error term that decays in both the deviation level and stepsize $\alpha$. Our analysis combines steady-state Stein's method with moment bounds on the SA iterations, and uses Poisson equation techniques to manage temporal dependence in the Markovian noise setting. We adapt the same toolkit to SGD with general convex objectives and suggest non-Gaussian limiting behavior, which is validated via numerical experiments.

[1]Department of Mathematics, University of Michigan Ann Arbor, MI, USA [2]School of Mathematics, Georgia Institute of Technology, Atlanta, GA, USA [3]H. Milton Stewart School of Industrial & Systems Engineering, Georgia Institute of Technology, Atlanta, GA, USA [4]School of Computer Science, Georgia Institute of Technology, Atlanta, GA, USA. Correspondence to: Zedong Wang <zwang3524@gatech.edu>, Siva Theja Maguluri <siva.theja@gatech.edu>.

*Proceedings of the 43rd International Conference on Machine Learning*, Seoul, South Korea. PMLR 306, 2026. Copyright 2026 by the author(s).

## 1. Introduction

Stochastic approximation (SA) algorithm is a fundamental framework for large-scale machine learning (ML), e.g., stochastic gradient descent (SGD) and temporal-difference methods in reinforcement learning (RL). Its iterations typically take the form

$$X_{k+1}^{(\alpha)} \;=\; X_k^{(\alpha)} + \alpha\big(F(X_k^{(\alpha)}) + \xi_k\big), \qquad (1)$$

with stepsize $\alpha > 0$. Practitioners often prefer constant stepsize SA/SGD in training modern learning models since it can reach useful solutions faster under fixed computational budget (Bottou et al., 2018). With constant $\alpha$, the iterations $\{X_k^{(\alpha)}\}_{k \geq 0}$ typically form an ergodic Markov chain with a stationary law $X^{(\alpha)}$. Given that practitioners often employ the algorithms for long periods, this paper studies the long-run performance of constant-stepsize SA algorithms, i.e., the steady state $X^{(\alpha)}$. The existence and stability conditions for stationary distributions have been extensively studied in the literature (Meyn & Tweedie, 2009). Yet due to the unknown nature of noise in SA/SGD iterations, the stationary distribution is often intractable except in special cases, e.g., linear SA with Gaussian noise. The focus of this paper is to analytically characterize the steady state $X^{(\alpha)}$, particularly the fluctuation $X^{(\alpha)} - x^*$ around the true solution $x^*$.

An attempt for the characterization is studied in (Chen et al., 2022) from an asymptotic viewpoint as $\alpha \downarrow 0$. Under standard smoothness assumptions for SA algorithms, the authors prove that the centered-and-scaled steady state $Y^{(\alpha)} := (X^{(\alpha)} - x^*)/\sqrt{\alpha}$ converges weakly to a Gaussian distribution as $\alpha \downarrow 0$. This limit suggests that for small $\alpha$, steady state $X^{(\alpha)}$ has an approximately Gaussian fluctuation around the true solution $x^*$. However, the limit result does not describe the pre-limit $\alpha$ regime, i.e., when $\alpha$ is small but fixed in practice.

Motivated by this gap, this paper targets at an *explicit pre-limit* characterization for the fluctuation $X^{(\alpha)} - x^*$. Our goal is two fold. We first study, for each fixed $\alpha$, an explicit approximation error bound in Wasserstein metric between the law of $Y^{(\alpha)}$ and its Gaussian limit. Explicit here refers to a clear dependency on all parameters including the stepsize $\alpha$ and the dimension $d$ without any unspecified constants.

Moreover, we want the approximation error to vanish as $\alpha \downarrow 0$, directly implying the prior limit results. Moreover, while Chen et al. (2022) only studies the limit results under i.i.d. noise model, we further extend the analysis to Markovian noise settings, which is more realistic in RL applications.

We further translate the approximation error into an interpretable tail characterization for the steady state fluctuation, namely the tail probability $\mathbb{P}(\|X^{(\alpha)} - x^*\| \geq a)$ for $a > 0$. Particularly, anchored on the Gaussian limit, we sandwich this tail probability between the corresponding Gaussian tail probability plus/minus an explicit error term vanishing as $\alpha \downarrow 0$. Moreover, the resulting tail bound is *non-uniform* in $a$, i.e., the error term depends on $a$ and decreases as $a$ increases, reflecting a non-uniform Berry-Esseen (Bikelis, 1966) type behavior in the tails.

### 1.1. Main contributions

Our main contribution is as follows.

1. For i.i.d. noise, we first characterize the pre-limit Gaussian approximation error for the steady-state fluctuation of constant-stepsize SA in Wasserstein distance. We obtain the errors for three representative modeling: (i) SGD with strongly convex and smooth objectives, (ii) linear SA, and (iii) nonlinear SA under a contraction condition. We achieve Wasserstein upper bounds of the form $c\alpha^{1/2} \log(1/\alpha)$ for small stepsize $\alpha$ with explicit constant $c$ defined later in the paper. Taking $\alpha \downarrow 0$ recovers the limit result in (Chen et al., 2022).

2. We achieve non-asymptotic tail bounds based on the Wasserstein approximation error. In particular, we show a bound of non-uniform Berry-Esseen type,

$$\Phi_\zeta^c(a) - C_d \frac{\alpha^{\frac{1}{4}} \log^{\frac{1}{2}}(1/\alpha)}{a}$$

$$\leq \mathbb{P}\big(\langle \zeta, Y^{(\alpha)} \rangle \geq a\big) \leq \Phi_\zeta^c(a) + C_d \frac{\alpha^{\frac{1}{4}} \log^{\frac{1}{2}}(1/\alpha)}{a},$$

with $\Phi_\zeta^c(\cdot)$ the target Gaussian tail probability and $\zeta$ an arbitrary unit vector.

3. We extend our analysis to Markovian noise for all three SA models above, capturing temporally correlated noise common in reinforcement learning. This extension is achieved by adding a Poisson equation technique to handle the Markovian correlation. We obtain the same order of Wasserstein bounds $c'\alpha^{1/2} \log(1/\alpha)$, and establish corresponding non-asymptotic tail bounds. As a by-product we achieve an asymptotic normality result for Markovian noise.

4. We generalize our steady state analysis to discuss SGD for general convex instead of strongly convex objec-

tives. Assuming a stability and Stein's equation hypothesis, we prove that the limit distribution of scaled $X^{(\alpha)} - x^*$ is no longer Gaussian, and we identify both the right scaling of this fluctuation and the limiting Gibbs law. We quantify another pre-limit approximation error in Wasserstein distance between properly scaled fluctuation and its Gibbs limit. We verify the proposed scaling and limit via numerical experiments.

### 1.2. Our Approach

To quantify the pre-limit approximation error in Wasserstein metric, our approach has two key ingredients. Firstly, we employ Stein's method via generator comparison to compare the steady state $Y^{(\alpha)}$ and its Gaussian limit $Y$ by bounding the difference of their associated Stein operators. Particularly, we choose the operator of $Y^{(\alpha)}$ from the one-step update of the SA iterations, and choose the Ornstein-Uhlenbeck (OU) generator for the Gaussian limit $Y$. First introduced in (Barbour, 1990) for functional approximation, this generator-based Stein's method has been developed into steady state analysis for queueing models (Gurvich, 2014). In our SA setting, we take a second-order Taylor expansion to align the leading terms of the two generators, and control the remainder terms using appropriate moment estimates of $Y^{(\alpha)}$ with Stein solution bounds (see (Ross, 2011)).

Secondly, to extend the same Wasserstein analysis from i.i.d to Markovian noise, we use a Poisson equation technique to re-express the correlated noise in terms of a one-step transition. In the generator comparison, this re-expression is essential since a linear and quadratic noise term no longer vanish automatically as in the i.i.d case. We apply the Poisson equation once to control the linear noise term, and again to identify the long-run covariance structure from the quadratic noise term. We finally bound the remainders explicitly using the same generator-based Stein's method framework as above.

## 2. Problem Setup

Formally, we study the constant-stepsize stochastic approximation (SA) recursion in Iteration (1) with $X_0^{(\alpha)} \in \mathbb{R}^d$, $k \in \mathbb{N}$, $F : \mathbb{R}^d \to \mathbb{R}^d$ a deterministic differentiable update direction, $\alpha > 0$ a constant stepsize, and $\{\xi_k\}_{k \geq 0}$ a noise sequence with mild moment control. Recursion (1) can be viewed as a noisy iterative method for solving the root-finding problem $F(x) = 0$, which covers many algorithms in machine learning and reinforcement learning. For example, when $F(x) = -\nabla f(x)$, recursion (1) covers the family of stochastic first-order optimizers, including SGD and variants (Sutskever et al., 2013). In reinforcement learning, a common choice is $F(x) = T^\star(x) - x$, where $T^\star$ is the Bellman optimality operators. In this case (1) captures

value-based updates including temporal-difference methods and Q-learning (Sutton et al., 1998).

Under standard stability conditions (Meyn & Tweedie, 2009), the chain $\{X_k^{(\alpha)}\}$ admits a unique invariant distribution $\pi_\alpha$. Let $X^{(\alpha)} \sim \pi_\alpha$ and $x^*$ satisfy $F(x^*) = 0$. Recall that results from (Chen et al., 2022) show the centered-scaled fluctuation $Y^{(\alpha)} := (X^{(\alpha)} - x^*)/\sqrt{\alpha}$ converges weakly to a Gaussian vector $Y$ as $\alpha \downarrow 0$. A central question we address is to describe the *pre-limit* behavior of this approximation. As mentioned, we quantify the Wasserstein metric between the laws of $Y^{(\alpha)}$ and $Y$, defined as

$$\mathcal{W}(\mu, \nu) := \inf_{\gamma \in \Gamma(\mu, \nu)} \int_{\mathbb{R}^d \times \mathbb{R}^d} \|x - y\|_2 \, d\gamma(x, y) \quad (2)$$

$$\overset{(a)}{=} \sup_{f \in \mathrm{Lip}_1} \left| \int_{\mathbb{R}^d} f(x) \, d\mu(x) - \int_{\mathbb{R}^d} f(x) \, d\nu(x) \right|, \quad (3)$$

with $\mu, \nu$ being the distributions of $Y^{(\alpha)}$ and $Y$, respectively. The norm $\|\cdot\|_2$ the standard Euclidean norm, $\mathrm{Lip}_1$ the 1-Lipschitz functions with respect to $\|\cdot\|_2$, and $\Gamma(\mu, \nu)$ the set of all couplings of $\mu$ and $\nu$. We will work with the dual form in (a) which follows from Kantorovich-Rubinstein duality (Villani et al., 2009). Hereafter, we denote $d_W(X, Y) := \mathcal{W}(\mathcal{L}(X), \mathcal{L}(Y))$ for random variables $X$ and $Y \in \mathbb{R}^d$ with laws $\mathcal{L}(X)$ and $\mathcal{L}(Y)$, respectively.

## 3. I.I.D. Noise Case

In this section, we present our results on the pre-limit approximation error in terms of Wasserstein distance and tail bounds for the steady state fluctuation of SA under i.i.d. noise. We first state the assumptions on the noise sequence, then present the theorems and discussion for the three representative SA models.

**Assumption 3.1.** The noise sequence $\{\xi_k\}_{k \geq 0}$ consists of independent and identically distributed random vectors with $\mathbb{E}[\xi_k] = 0$ and positive definite covariance matrix $\Sigma \in \mathbb{R}^{d \times d}$. Moreover, $\mathbb{E}[\|\xi_k\|_2^3] < \infty$. $\|\cdot\|_2$ denotes the Euclidean norm.

Practically, the i.i.d noise is realized by sampling batches of data or trajectories independently for ML/RL. Analytically, since we are approximating a Gaussian limit for the stationary law of SA algorithm, it is standard to assume the noise sequence has a finite second moment so that the covariance matrix of the Gaussian limit is well-defined (Chen et al., 2022). This assumption allows us to consider arbitrary noise distributions beyond the Gaussian noise assumption in (Mandt et al., 2017). Here, we further assume the noise has a finite third moment to achieve explicit approximation error bounds and tail bounds, which is comparable to the classical 3-rd moment condition in the Berry-Esseen theorem for CLT of i.i.d summation (Petrov, 2012).

### 3.1. SGD for Strongly Convex and Smooth Objective

Suppose mapping $F$ in (1) is given by $F(x) = -\nabla f(x)$ for some objective function $f : \mathbb{R}^d \to \mathbb{R}$. Then, the SA algorithm becomes classical SGD

$$X_{k+1}^{(\alpha)} = X_k^{(\alpha)} + \alpha(-\nabla f(X_k^{(\alpha)}) + \xi_k) \quad (4)$$

for minimizing $f(x)$. To proceed, we first impose standard smoothness and strong convexity assumptions on the objective function $f$.

**Assumption 3.2.** The objective function $f : \mathbb{R}^d \to \mathbb{R}$ is twice differentiable and satisfies: (i) $L$-smoothness: $f(y) \leq f(x) + \langle \nabla f(x), y - x \rangle + \frac{L}{2}\|x - y\|_2^2$ for all $x, y \in \mathbb{R}^d$; and (ii) $\sigma$-strong convexity: $f(y) \geq f(x) + \langle \nabla f(x), y - x \rangle + \frac{\sigma}{2}\|x - y\|_2^2$ for all $x, y \in \mathbb{R}^d$, where $L > \sigma > 0$ are constants and $\|\cdot\|_2$ denotes the Euclidean norm.

From Assumption 3.2, the function $f$ has a unique minimizer $x^*$ such that $\nabla f(x^*) = 0$. Recalling the centered scaled iterate $Y_k^{(\alpha)} = (X_k^{(\alpha)} - x^*)/\sqrt{\alpha}$. We first develop existence of the stationary distribution of $\{Y_k^{(\alpha)}\}$. The following lemma is justified in Section C.6.

**Lemma 3.3.** *If Assumptions 3.1 and 3.2 hold, then there exists some $a' > 0$ such that for a fixed $\alpha \in (0, a')$, the sequence of random vectors $\{Y_k^{(\alpha)}\}$ converges weakly to some random vector $Y^{(\alpha)}$, where $\mathbb{E}[\|Y^{(\alpha)}\|_2^3] < \infty$.*

To proceed with the error analysis, we assume the following extra regularity on the objective function $f$.

**Assumption 3.4.** *The objective function $f$ is thrice differentiable and $\sup_{x \in \mathbb{R}^d} \left\| \frac{\partial^3 f}{\partial x_i \partial x_j \partial x_k}(x) \right\| = M < \infty$ for all $i, j, k \in \{1, \ldots, d\}$ and some $M \in \mathbb{R}$.*

Assumption 3.4 is a standard quantitative smoothness condition that controls Lipschitzness of the Hessian matrix of $f$. This assumption is key to upgrade mere weak convergence of $Y^{(\alpha)}$ to an explicit approximation error in Wasserstein metric (see (Bach & Moulines, 2011, H6) for similar use in non-asymptotic SA analysis).

Having outlined all assumptions, we are ready to present our theorem for SGD.

**Theorem 3.5.** *Consider the iterates generated by the recursion (4) and suppose Assumptions 3.1, 3.2, and 3.4 hold. Let $H_f(x^*) \in \mathbb{R}^{d \times d}$ denote the Hessian matrix of $f$ evaluated at $x^*$, and let $\Sigma_Y \in \mathbb{R}^{d \times d}$ be the unique positive definite solution to the following Lyapunov equation:*

$$H_f(x^*)\Sigma_Y + \Sigma_Y H_f(x^*)^\top = \Sigma, \quad (5)$$

*where $\Sigma$ is the covariance matrix of the noise from Assumption 3.1. Define Gaussian vector $Y \sim \mathcal{N}(0, \Sigma_Y)$. Then there exists an $\alpha' > 0$ such that for all $\alpha \in (0, \alpha')$,*

the Markov chain $\{X_k^{(\alpha)}\}_{k \geq 0}$ defined in (4) has a unique stationary distribution $\pi_\alpha^X$. Let $X^{(\alpha)} \sim \pi_\alpha^X$ and define $Y^{(\alpha)} := (X^{(\alpha)} - x^*)/\sqrt{\alpha}$, we have the following results:

1. the Wasserstein metric between $Y^{(\alpha)}$ and $Y$ is bounded

$$d_W(Y^{(\alpha)}, Y) \leq U_1 \alpha^{\frac{1}{2}} \log \frac{1}{\alpha} \qquad (6)$$

where constant $U_1$ is defined in (54).

2. for any test direction $\zeta \in \mathbb{R}^d$ with $\|\zeta\| = 1$ and deviation $a > 0$, we have

$$|\mathbb{P}(\langle Y^{(\alpha)}, \zeta \rangle > a) - \mathbb{P}(Z_\zeta > a)|$$
$$\leq U_1' \frac{\alpha^{1/4} \log^{1/2}(1/\alpha)}{a}, \qquad (7)$$

where constant $U_1'$ is defined in (55). Random variable $Z_\zeta \sim \mathcal{N}(0, \zeta^\top \Sigma_Y \zeta)$.

Note that $\Sigma$ is positive semidefinite from Assumptions 3.1, and the matrix $H_f(x^*)$ is positive definite from Assumption 3.2. Thus, the solution $\Sigma_Y$ to the Lyapunov equation (5) is well-defined and positive semidefinite (Haddad & Chellaboina, 2011), and the Gaussian vector $Y$ is therefore well posed.

From the Wasserstein bound (6) in Theorem 3.5, we obtain a quantitative approximation error between $Y^{(\alpha)}$ and $Y$ that vanishes as $\alpha \to 0$. Since the Wasserstein metric metrizes weak convergence (Villani et al., 2009) and satisfies $d_W(X, Z) = 0$ if and only if $X \overset{d}{=} Z$, Theorem 3.5 therefore implies that $Y^{(\alpha)}$ converges in distribution to $Y$ as $\alpha \downarrow 0$, thus recovering the weak convergence statement and settles the uniqueness conjecture in (Chen et al., 2022). Moreover, Theorem 3.5 studies small but fixed stepsize $\alpha$, providing a non-asymptotic bound with explicit dependence on stepsize $\alpha$. We also note that the leading $\sqrt{\alpha}$ scaling is consistent with the canonical CLT theorem for i.i.d sum. In the quadratic case $X_{k+1} = (1 - \alpha)X_k + \alpha\xi_k$, the stationary scaled error $Y^{(\alpha)} \overset{d}{=} \sqrt{\alpha} \sum_{j \geq 0}(1 - \alpha)^j \xi_j$, a weighted sum with length $\sum_{j \geq 0}(1 - \alpha)^j \asymp 1/\alpha$. Thus the Gaussian approximation of $Y^{(\alpha)}$ is comparable to a classical CLT with $n \asymp 1/\alpha$, where Wasserstein errors are in general of optimal order $n^{-1/2}$, so one should not expect an improvement to $o(\alpha^{1/2})$ without additional structure, see (Rio, 2009).

The logarithmic factor $\log(1/\alpha)$ in (42) arises as a technical artifact of high-dimensional Stein's method analysis (Gallouët et al., 2018). Similar logarithmic factor also appears in Markov Chain CLT (Srikant, 2025, Theorem 2). One attempt of improving the logarithmic factor is via comparison with CLT for i.i.d. sum, as a concurrent work on CLT shows $1/\sqrt{n}$ rate without logarithmic factor (Zhang & Xie, 2026).

Yet such comparison requires diminishing step size rather than fixed $\alpha$. We believe removing this logarithmic factor requires substantially new techniques in high-dimensional Stein's method, and is beyond the scope of this paper.

Complementing the Wasserstein bound, we further develop a pre-limit tail bound for the one-dimensional projections of $Y^{(\alpha)}$ in (7). This inequality (7) directly implies a tail bound on the fluctuation of $X^{(\alpha)}$ around $x^*$ as

$$\mathbb{P}(\langle X^{(\alpha)} - x^*, \zeta \rangle > b) \leq \Phi^c(\frac{b}{\sqrt{\alpha \zeta^\top \Sigma_Y \zeta}})$$
$$+ U_1' \frac{\alpha^{3/4} \log^{1/2}(1/\alpha)}{b},$$

with $b > 0$ and $\Phi^c(\cdot)$ the complementary CDF of standard normal distribution. Thus it enables statistical inference for fluctuation errors of SGD iterates around the minimizer $x^*$. For the two-sided tail bound in (7), we first note that for any unit vector $\zeta \in \mathbb{R}^d$, the right-hand side of (7) vanishes as $\alpha \downarrow 0$ for any fixed $a > 0$, showing that $\langle Y^{(\alpha)}, \zeta \rangle$ converges in distribution to $\langle Y, \zeta \rangle$ as $\alpha \downarrow 0$. Since we pick $\zeta$ arbitrarily, from Cramér-Wold Theorem (Samanta, 1989), this implies $Y^{(\alpha)} \Rightarrow Y$ as $\alpha \downarrow 0$, again showing the weak convergence. More importantly, unlike Berry-Esseen type of bound on $\sup_{a \in \mathbb{R}} |\mathbb{P}(Y^{(\alpha)} \cdot \zeta > a) - \mathbb{P}(Z > a)|$ that is uniform over $a$, our bound in (7) decays as $a$ increases, reflecting a non-uniform Berry-Esseen type behavior (Bikelis, 1966). The monotone decay enables the right-hand side of (7) to vanish to zero as $a \to \infty$ for any fixed $\alpha > 0$, aligning with the intuition that both $\mathbb{P}(\langle Y^{(\alpha)}, \zeta \rangle > a)$ and $\mathbb{P}(Z > a)$ decay to zero as $a \to \infty$. Thus the non-uniform improvement with $a$ is highly suitable for large-deviation analysis, ensuring that the approximation error is more precise when we probe the tails to higher reliability targets (e.g., $\mathbb{P}(\cdot) \leq \delta$ for smaller $\delta$). Thereby, the tail bound (7) enables a high-confidence on rare event control that is crucial for risk-sensitive applications (Rockafellar & Uryasev, 2002).

We comment on how the bounds depend on the problem parameters. The constants $U_1$ and $U_1'$ in (6) and (7) are given in fully explicit forms rather than merely order-level scaling. In particular, their polynomial dependence on dimension $d$ is consistent with the high-dimensional Wasserstein metric. In the primal form (2), the transportation cost of Wasserstein metric is Euclidean, so the error naturally aggregates and depends on all $d$ coordinates. Our bounds quantify this dependence explicitly while achieving the $\mathcal{O}(\alpha^{1/2} \log(1/\alpha))$ stepsize scaling.

We conclude this subsection by commenting on the small-deviation regime, where the factor $1/a$ in tail bound (7) may be large. In this case, one can merge a bound on Kolmogorov distance $d_K(a^\top X, a^\top Y)$ and the non-uniform bound (7) by taking the minimum. Since the Kolmogorov distance is uniform over $a$, it gives better control when $a$

is small. Such a Kolmogorov bound can be derived by the same proof strategy. Because the argument is applied to the one-dimensional projection $a^\top X$, the only change is to use the one-dimensional Stein factor bound for Kolmogorov distance in place of the bound used in Theorem 3.5 (see (Ross, 2011, Section 3.6)). We therefore do not expect additional technical difficulty and omit this extension.

## 3.2. Linear SA

Suppose the mapping $F$ in (1) is linear, i.e. of the form

$$F(x) = Ax + b,$$

where $A \in \mathbb{R}^{d \times d}$ is a matrix and $b \in \mathbb{R}^d$ is a vector. Then the SA recursion (1) specializes to

$$X_{k+1}^{(\alpha)} = X_k^{(\alpha)} + \alpha\left(AX_k^{(\alpha)} + b + \xi_k\right). \qquad (8)$$

This recursion can be interpreted as a noisy iterative method for finding root for the system of linear equations $Ax + b = 0$. Since $A$ is not necessarily symmetric, the mapping $F(x) = Ax + b$ may fail to be the gradient of any objective function, thus falling outside the scope of Theorem 3.5. A canonical example of such linear SA in RL is the temporal-difference (TD) learning with linear function approximation (Tsitsiklis & Van Roy, 1996). For clarity, we first treat the i.i.d. noise setting, and defer the Markovian noise case, which is more directly aligned with RL, to Section 4. We first impose a standard structural condition below on $A$ to ensure stability of the noise-free dynamics.

**Assumption 3.6.** The matrix $A \in \mathbb{R}^{d \times d}$ is Hurwitz; that is, every eigenvalue $\lambda$ of $A$ has negative part, $\mathrm{Re}(\lambda) < 0$.

A Hurwitz matrix is necessarily non-singular. Therefore, Assumption 3.6 guarantees that the equation $Ax + b = 0$ has a unique solution, which we denote by $x^*$. In the following, we study the centered, scaled process $Y_k^{(\alpha)} := (X_k^{(\alpha)} - x^*)/\sqrt{\alpha}$, develop the existence of stationarity, and characterize the steady state fluctuation $X^{(\alpha)} - x^*$. For linear SA, under identity $Ax^* + b = 0$, we can obtain a clean recursion for $\{Y_k^{(\alpha)}\}_{k \geq 1}$ as follows.

$$Y_{k+1}^{(\alpha)} = (I + \alpha A)\, Y_k^{(\alpha)} + \sqrt{\alpha}\, \xi_k. \qquad (9)$$

We now present the theorem for pre-limit approximation.

**Theorem 3.7.** *Consider the iterates $\{Y_k^{(\alpha)}\}_{k \geq 0}$ generated by the recursion (9). Suppose Assumption 3.1 and 3.6, there exists $\alpha' > 0$ such that for all $\alpha \in (0, \alpha')$, the chain $\{Y_k^{(\alpha)}\}_{k \geq 0}$ admits a unique stationary distribution. Moreover, with this $\alpha'$, let $Y^{(\alpha)}$ denote a random vector distributed according to this stationary distribution. Let $\Sigma_Y \in \mathbb{R}^{d \times d}$ be the unique positive semidefinite solution to the Lyapunov equation: $A\,\Sigma_Y + \Sigma_Y\,A^\mathsf{T} + \Sigma = 0$, and define $Y \sim \mathcal{N}(0, \Sigma_Y)$. Then the following hold with $\alpha < \alpha'$,*

1. *the Wasserstein distance between $Y^{(\alpha)}$ and $Y$ is bounded in the following way:*

$$d_W(Y^{(\alpha)}, Y) \leq U_2 \alpha^{\frac{1}{2}} \log \frac{1}{\alpha}, \qquad (10)$$

*where the constant $U_2$ is defined in (60).*

2. *for $\zeta \in \mathbb{R}^d$, $\|\zeta\| = 1$ and deviation $a > 0$, we have*

$$|\mathbb{P}(\langle Y^{(\alpha)}, \zeta\rangle > a) - \mathbb{P}(Z_\zeta > a)|$$
$$\leq U_2' \frac{\alpha^{1/4} \log^{1/2}(1/\alpha)}{a}, \qquad (11)$$

*with constant $U_2'$ defined in (61), $Z_\zeta \sim \mathcal{N}(0, \zeta^\top \Sigma_Y \zeta)$.*

Note that under the Hurwitz property of $A$ in Assumption 3.6, the Lyapunov equation has a unique solution $\Sigma_Y \succeq 0$ thus the Gaussian vector $Y$ is well posed.

The Wasserstein bound (10) again provides an explicit pre-limit approximation error between $Y^{(\alpha)}$ and $Y$, recovering the weak convergence and solving the uniqueness conjecture in (Chen et al., 2022). Meanwhile, we complement this Wasserstein error bound with the tail estimate in (11), which yields a non-uniform Berry–Esseen style bound.

With TD learning and linear function approximation as the motivating context, the scalar projection $\langle X^{(\alpha)} - x^*, \zeta\rangle$ represents a linear query of the steady-state fluctuation of the learned action-value function around its Bellman fixed point $x^*$ along the direction $\zeta \in \mathbb{R}^d$. Tail bound in (11) is thus describing the probability with which this linear query $\langle X^{(\alpha)} - x^*, \zeta\rangle$ deviates beyond a threshold $a\sqrt{\alpha}$. Such linear queries are ubiquitous in RL. A choice is $\xi = \frac{1}{\sqrt{2}}(e_i - e_j)$ for $i, j \in \{1, \ldots, d\}$, capturing pairwise action-value comparisons that enhance algorithm robustness (Farahmand, 2011), with other choices representing risk-constrained RL safety margins (Achiam et al., 2017; Gangapurwala et al., 2020). The non-uniform tail bound (14) thus provides refined control on the linear-query than uniform bounds, valuable for risk-aware RL that is based on estimated action gaps (Köse & Ruszczyński, 2021).

## 3.3. SA with Contractive Nonlinear Operator

Suppose the drift term can be written as a residual of a nonlinear vector field, namely $F(x) = \mathcal{T}(x) - x$ with $\mathcal{T} : \mathbb{R}^d \to \mathbb{R}^d$ defined via its components

$$\mathcal{T}(x) = (f^{(1)}(x), \ldots, f^{(d)}(x))^T$$

Then our SA recursion (1) becomes

$$X_{k+1}^{(\alpha)} = X_k^{(\alpha)} + \alpha\left(\mathcal{T}(X_k^{(\alpha)}) - X_k^{(\alpha)} + \xi_k\right). \qquad (12)$$

Many RL algorithms can be cast into this form, e.g., when $\mathcal{T}$ is the discounted Bellman operator on action-value functions, (12) corresponds to the sample-based Bellman iteration underlying Q-learning (Sutton et al., 1998). Corresponding to the discounted RL setting, we study the case where $\mathcal{T}$ is a contraction mapping, assumed below.

**Assumption 3.8.** Let $\mu_1, \ldots, \mu_d \in \mathbb{R}^+$ be positive weights and define the weighted Euclidean norm as $\|x\|_\mu :=$ $\left(\sum_{i=1}^d \mu_i x_i^2\right)^{1/2}$, for $x \in \mathbb{R}^d$. The vector field $\mathcal{T}$ is continuously differentiable and there exists $\gamma \in (0, 1)$ such that $\|\mathcal{T}(x_1) - \mathcal{T}(x_2)\|_\mu \leq \gamma \|x_1 - x_2\|_\mu$ for all $x_1, x_2 \in \mathbb{R}^d$.

From assumption 3.8, the vector field $\mathcal{T}$ is a strict contraction with respect to the weighted Euclidean norm $\|\cdot\|_\mu$. Contraction is a standard stability condition in SA analysis for RL. It implies a unique fixed point $x^*$ by the Banach fixed-point theorem. In discounted RL, Bellman operators are classically contractive, which prevents bootstrapping error from blowing up. The weighted Euclidean norm $\|\cdot\|_\mu$ reflects that different state-action coordinates are visited and updated at different long-run frequencies, and the weights $\mu$ is proportional to visitation distribution. Thus, the contraction is assumed under $\|\cdot\|_\mu$ to ensure that errors shrink faster on the coordinates that are updated more frequently (Bertsekas & Tsitsiklis, 1995).

We first study the property for this vector field $\mathcal{T}$. We let $J \in \mathbb{R}^{d \times d}$ denote the Jacobian of $\mathcal{T}$ evaluated at the fixed point $x^*$. We present the following lemma on the spectral radius of $J$ implied by Assumption 3.8, which is later used to identify the covariance matrix of the Gaussian limit.

**Lemma 3.9.** Let $r(J) := \max_{1 \leq i \leq d} |\lambda_i(J)|$ be the spectral radius of $J$. Then $r(J) < 1$.

Similar to SGD, we need an extra regularity condition for $\mathcal{T}$ to upgrade from weak convergence to approximation error.

**Assumption 3.10.** Each component of the vector field $\mathcal{T}$, the $f^{(i)}$ for $i = 1, \ldots, d$, is twice continuously differentiable. There exists a constant $M < \infty$ such that

$$\sup_{1 \leq i \leq d} \sup_{x \in \mathbb{R}^d} \|f_{jk}^{(i)}(x)\|_\infty \leq M,$$

where $f_{jk}^{(i)}$ denotes the partial derivative $\frac{\partial^2 f^{(i)}}{\partial x_j \partial x_k}$.

Having outlined the assumptions, we now state the main results for SA under a contractive drift.

**Theorem 3.11.** *Consider the recursion* (12) *and suppose Assumptions 3.1, 3.8, and 3.10 hold. Then there exists $\alpha'' \in (0, 1]$ such that for every $\alpha \in (0, \alpha'')$, the Markov chain $\{X_k^{(\alpha)}\}_{k \geq 0}$ admits a unique stationary distribution $\pi_\alpha^{\mathcal{T}}$. Let $X^{(\alpha)} \sim \pi_\alpha^{\mathcal{T}}$ and define $Y^{(\alpha)} := (X^{(\alpha)} - x^\star)/\sqrt{\alpha}$. Let $J := D\mathcal{T}(x^\star)$ and let $\Sigma$ be the noise covariance from Assumption 3.1. The Lyapunov equation $(J - I)\Sigma_Y +$*

$\Sigma_Y(J - I)^\top + \Sigma = 0$ *admits a unique symmetric positive definite solution $\Sigma_Y$. Defining $Y \sim \mathcal{N}(0, \Sigma_Y)$, then the following holds,*

1. *we have the following Wasserstein bound,*

$$d_W\big(\mathcal{L}(Y^{(\alpha)}), \mathcal{L}(Y)\big) \leq U_3 \sqrt{\alpha} \log(1/\alpha), \quad (13)$$

*where constant $U_3$ is defined in* (67).

2. *For any test direction $\zeta \in \mathbb{R}^d$ with $\|\zeta\| = 1$ and deviation $a > 0$, we have*

$$|\mathbb{P}(\langle Y^{(\alpha)}, \zeta \rangle > a) - \mathbb{P}(Z_\zeta > a)|$$
$$\leq U_3' \frac{\alpha^{1/4} \log^{1/2}(1/\alpha)}{a}, \quad (14)$$

*with $U_3'$ defined in* (68), *$Z_\zeta \sim \mathcal{N}(0, \zeta^\top \Sigma_Y \zeta)$.*

Note that by the contraction assumption 3.8, Lemma 3.9 implies that $J - I$ is a Hurwitz matrix and thus the Lyapunov equation has a unique solution $\Sigma_Y \succ 0$. Therefore the Gaussian vector $Y$ is well posed.

The Wasserstein bound controls distributional error, while the non-uniform tail bound quantifies one-dimensional projection accuracy. These bounds decay as $\alpha \downarrow 0$, establishing explicit pre-limit error for Gaussian approximation and corresponding weak convergence for SA under contraction under both i.i.d. and Markovian noise. The non-uniform $a$-dependence is particularly valuable for risk-aware Q-learning that requires high-confidence control of rare events (Chow et al., 2018).

## 4. Markovian Noise Setting

In this section, we extend our results to the Markovian noise setting. Specifically, we consider the noise sequence $\{\xi\}_{k \geq 0}$ generated by an underlying *exogenous* Markov chain $(Z_k)_{k \geq 0}$, i.e., $\xi_k = \xi(Z_k)$ (rather than depending directly on $X_k$). This captures sequentially generated data streams, particularly in RL where $Z_k$ can represent the state–action trajectory and lead to temporally correlated updates (Sutton et al., 1998). We let $(Z_k)_{k \geq 0}$ be a time-homogeneous Markov chain on a measurable state space $(\mathsf{Z}, \mathcal{Z})$, and use $P$ to denote both the one-step transition kernel and the associated Markov operator $(Pf)(z) := \mathbb{E}[f(Z_1) \mid Z_0 = z]$. Let $\xi : \mathsf{Z} \to \mathbb{R}^d$ be a measurable function and $\xi_k := \xi(Z_k)$, the SA recursion (1) in Markovian noise setting becomes

$$X_{k+1}^{(\alpha)} = X_k^{(\alpha)} + \alpha\big(F(X_k^{(\alpha)}) + \xi(Z_k)\big), \ k \in \mathbb{N}, \quad (15)$$

with $X_0^{(\alpha)} \in \mathbb{R}^d$. We further define the following standard objects associated with the Markov noise.

**Definition 4.1** (Long-Run Covariance and Poisson Equations). Let $(Z_k)_{k\geq 0}$ be stationary with distribution $\pi_Z$. If the series below is entrywise convergent, define

$$\Sigma_M := \mathbb{E}\big[\xi(Z_0)\xi(Z_0)^\top\big]$$
$$+ \sum_{m=1}^{\infty} \mathbb{E}\big[\xi(Z_0)\xi(Z_m)^\top + \xi(Z_m)\xi(Z_0)^\top\big].$$

Next, let $V : \mathsf{Z} \to \mathbb{R}^d$ be any measurable solution to the Poisson equation $\xi = V - PV$. Given such a $V$, define

$$\Phi(z) := -V(z)\xi(z)^\top + \tfrac{1}{2}\Sigma_M + \tfrac{1}{2}\xi(z)\xi(z)^\top,$$

and let $W : \mathsf{Z} \to \mathbb{R}^{d\times d}$ be any measurable solution to the second Poisson equation

$$W - PW = \Phi - \mathbb{E}_{\pi_Z}[\Phi(Z_0)].$$

Note that when $(Z_k)_{k\geq 0}$ are i.i.d., $\Sigma_M$ reduces to the noise covariance $\Sigma$ in Assumption 3.1, the long-run covariance $\Sigma_M$ thus generalizes the noise covariance $\Sigma$ in the i.i.d. case. It will be used to characterize the asymptotic variance of the Gaussian limit. The Poisson equations are standard tools that re-express the correlated noise by one-step transition (Benveniste et al., 2012; Haque & Maguluri, 2024). With these definitions, we now state the assumptions for Markovian noise.

**Assumption 4.2** (Markovian noise). With the definitions of $\Sigma_M$ and the Poisson equations given above, assume:

1. (Marginal Stationarity) The chain $(Z_k)_{k\geq 0}$ is uniformly ergodic with unique stationary distribution $\pi_Z$. We assume $Z_0 \sim \pi_Z$ and $\mathbb{E}_{\pi_Z}[\xi(Z_0)] = 0$. The series defining $\Sigma_M$ is entrywise absolutely convergent and $\Sigma_M$ is positive definite.

2. (Regularity Conditions) There exist measurable solutions $V$ and $W$ to the two Poisson equations above such that

$$\sup_{z\in\mathsf{Z}}\big(\|V(z)\|_2 + \|\xi(z)\|_2 + \|W(z)\|_2\big) < \infty,$$

   where $\|\cdot\|_2$ denotes the Euclidean norm (and its induced matrix norm).

3. (Joint Stationarity) There exists $\alpha_0 > 0$ such that for all $\alpha \in (0, \alpha_0)$, the joint chain $(X_k^{(\alpha)}, Z_k)_{k\geq 0}$ induced by (15) admits a unique stationary distribution.

The above assumptions are standard in SA with Markovian noise. We assume the long-run covariance $\Sigma_M$ exists to ensure the covariance matrix of the Gaussian limit is well-defined. We also assume the conditions on Poisson solutions $V$ and $W$ to control the correlation structure. Such existence and regularity of the Poisson solutions $V$ and $W$ can be verified under uniform ergodic conditions on $(Z_k)$ (Meyn & Tweedie, 2009). Rather than repeating those technical verifications, we state them explicitly as assumptions. We impose a boundedness condition on $V$, $\xi$, and $W$ to control the moments of steady states. Finally, we assume the joint chain $(X_k^{(\alpha)}, Z_k)$ admits a unique stationary distribution to ensure the steady state is well defined. Such uniqueness can be established under standard minorization conditions for SGD (Yu et al., 2020, Assumption 2.3), or the uniform ergodicity of $(Z_k)$ in linear SA (Huo et al., 2023, Assumption 1). To keep the main text focused, we take this unique stationarity as an assumption here. In Section D.4, we assume a minorization condition and provide a proof of the unique stationarity of $(X_k^{(\alpha)}, Z_k)$. In practice, these assumptions are satisfied in tabular Q-/TD-learning, where the environment is typically modeled or discretized as a irreducible and aperiodic finite-state Markov chain.

Compared with the i.i.d. noise setting, the Markovian noise setting necessarily imposes stronger conditions on the noise process, e.g., boundedness to control temporal correlations. In contrast, the i.i.d. noise case has fewer restrictions on the noise distribution and supports analytically stronger results (requiring only finite third moments), so it is of independent theoretical interest. Despite being more restrictive on assumptions, the Markovian noise is highly practical, as it naturally captures temporally correlated noise common in reinforcement learning applications where data arrives sequentially along trajectories. With above assumptions, we now state the main theorem for Markovian noise.

**Theorem 4.3.** *Consider the recursion* (15) *under Assumption 4.2. Further assume the model-specific conditions for: (i) SGD, i.e., $F = -\nabla f$ with Assumptions 3.2 and 3.4; or (ii) linear SA, i.e., $F(x) = Ax + b$ with Assumption 3.6; or (iii) contractive SA with Assumptions 3.8 and 3.10. Let $\Sigma_Y$ be the unique solution to the corresponding Lyapunov equation in Theorem 3.5, 3.7, or 3.11, respectively, with $\Sigma$ replaced by the long-run covariance $\Sigma_M$. Let $x^*$ be the fixed point and define $Y \sim \mathcal{N}(0, \Sigma_Y)$. Let $X^{(\alpha)}$ follow the stationary law of $\{X_k^{(\alpha)}\}_{k\geq 0}$ and set $Y^{(\alpha)} := (X^{(\alpha)} - x^*)/\sqrt{\alpha}$. Then there exists $\alpha' > 0$ such that for all $\alpha \in (0, \alpha')$:*

1. *$d_W\big(Y^{(\alpha)}, Y\big) \leq U_4\,\alpha^{1/2}\log(1/\alpha)$, where $U_4$ is given in* (87) *with constant $L$ defined for different models in Section 4.3.*

2. *for any unit $\zeta \in \mathbb{R}^d$ and any $a > 0$,*

$$|\mathbb{P}(\langle Y^{(\alpha)}, \zeta\rangle > a) - \mathbb{P}(Z_\zeta > a)|$$
$$\leq U_4' \frac{\alpha^{1/4}\log^{1/2}(1/\alpha)}{a},$$

   *where $U_4'$ is given in* (93) *and $Z_\zeta \sim \mathcal{N}(0, \zeta^\top\Sigma_Y\zeta)$.*

Theorem 4.3 covers SGD, linear SA, and contractive SA

under Markovian noise. Relative to the i.i.d. setting, the only change is replacing $\Sigma$ by $\Sigma_M$ in the Lyapunov equation, e.g., in (5), which defines the Gaussian comparator $Y$. This Markovian formulation is natural in RL, where samples are generated along trajectories. It yields explicit pre-limit Gaussian approximation and tail-comparison bounds for RL-motivated SA schemes such as TD learning or Q learning, while the implications of such bounds were discussed in Section 3.

# 5. SGD for general objectives and Gibbs limit distribution

For i.i.d. noise sequence $\{\xi_k\}_{k \geq 0}$, beyond the strongly convex and L-smooth objectives treated in Theorem 3.5, the work in (Chen et al., 2022, Section 3) suggested that the scaling factor $\sqrt{\alpha}$ may no longer be appropriate for more general convex objectives, and the limiting distribution of SGD can be non-Gaussian. With the objective $f(x) = \frac{x^4}{4}$, they use numerical experiments to exhibit that the limiting distribution has density proportional to $e^{-x^4/c}$ for some constant $c > 0$, and the scaling factor is $\alpha^{\frac{1}{4}}$ instead of $\sqrt{\alpha}$. In this section, we extend our theoretical framework to justify their findings under the following conditions.

**Assumption 5.1.** The objective function $f : \mathbb{R} \to \mathbb{R}$ is convex and of class $C^h(\mathbb{R})$ for some even integer $h > 0$. It has a unique minimizer at $x^*$, and satisfies $\lim_{x \to x^*} f^{(k)}(x) = 0$ for all $1 \leq k \leq h - 1$, while $f^{(h)}(x^*) > 0$ and $f^{(h+1)}(x)$ uniformly bounded by constant $M$.

Since we assume a unique minimizer at $x^*$, the first nonvanishing derivative at $x^*$ must be of even order, we therefore restrict $h$ to be even. A concrete example is the quartic objective $f(x) = \frac{x^4}{4}$ with $h = 4$, which corresponds to the least-mean-fourth algorithm in signal processing (Walach & Widrow, 2003), and it reproduces the numerical example from (Chen et al., 2022). To proceed, we first impose the following moment condition on the noise sequence that is analogous to Assumption 3.1.

**Assumption 5.2.** The noise sequence $\{\xi_k\}_{k \geq 0}$ consists of i.i.d. random variables satisfying $\mathbb{E}[\xi_k^3] < \infty$.

Under Assumption 5.1, the objective $f$ need not be L-smooth or strongly convex. It has a local $h$-th order polynomial growth near $x^*$, since it implies $\lim_{x \to x^*} \frac{f(x) - f(x^*)}{|x - x^*|^h} > 0$ while $\lim_{x \to x^*} \frac{f(x) - f(x^*)}{|x - x^*|^k} = 0$ for all $1 \leq k \leq h-1$. Motivated by the stationary fixed-point relation and $\mathbb{E}|\xi_k|^3 < \infty$, we naturally require a cubic-integrability in steady state, leading to our following conjecture.

**Conjecture 5.3.** *Under Assumption 5.1 and 5.2, there exists some $a > 0$ such that for all stepsize $\alpha \in (0, a)$, the sequence of random variables $\{X_k^{(\alpha)}\}_{k \geq 0}$ generated by the*

*SGD iteration* (4) *converges in distribution to a random variable $X^{(\alpha)}$ as $k \to \infty$. Furthermore, there exists a constant $C_h > 0$ such that $\sup_{\alpha \in (0,a)} \mathbb{E}|X^{(\alpha)} - x^*|^{3h} \leq C_h$.*

Conjecture 5.3 is natural given the order-$h$ local behavior of $f$ near $x^*$ and the requirement of cubic integrability. More broadly, it can be viewed as a higher-order analogue of the stability results in (Yu et al., 2020). To implement our Stein framework for the order-$h$ Gibbs target, we additionally assume the following regularity condition of the Stein solution associated with the target Gibbs.

**Conjecture 5.4.** *For any $h \in \mathrm{Lip}_1(\mathbb{R})$, the Stein equation*

$$h(y) - \mathbb{E}[h(Z)] = -y^h g_h'(y) + \mathbb{E}[\xi_k^2] g_h''(y) \qquad (16)$$

*admits a solution $g_h : \mathbb{R} \to \mathbb{R}$ with uniformly bounded derivatives: $\sup_{y \in \mathbb{R}}(|g_h'(y)| + |g_h''(y)| + |g_h'''(y)|) < R$ for some constant $R$.*

The right-hand-side in (16) corresponds to the generator of the one-dimensional Langevin diffusion $dY_t = -Y_t^h dt + \sqrt{2\mathbb{E}[\xi_k^2]} \, dW_t$, whose invariant law is the Gibbs measure with density proportional to $\exp\{-|y|^{h+1}/((h+1)\mathbb{E}[\xi_k^2])\}$. For multivariate Gibbs distributions, uniform gradient bounds for this Stein equation are only known under strong log-concavity (Mackey & Gorham, 2016, Theorem 2.1) or dissipativity conditions (**?**, Assumption 2.1). However, the polynomial Gibbs potential in Conjecture 5.4 does not satisfy these conditions. Thus the conjectured gradient bound would require substantially new techniques, which is beyond the scope of the present paper. That said, in one-dimensional setting we can solve the ODE directly and derive gradient bound under mild tail conditions (Ley et al., 2017), thus supporting the plausibility of Conjecture 5.4. Having established the stability and regularity conjectures, we can now state our main result.

**Proposition 5.5.** *Under Assumptions 5.1–5.2 and Conjectures 5.3–5.4, consider SGD on the one-dimensional objective* (4). *Consider the steady state $X^{(\alpha)}$ for stepsize $\alpha \in (0, \bar{\alpha})$ whose existence is guaranteed by Conjecture 5.3. Let $Y$ follow the Gibbs distribution with density proportional to $\exp\left[-\frac{2f^{(h)}(x^*)}{h\mathbb{E}[\xi_k^2]} y^h\right]$. Then there exists $\alpha' > 0$ such that for all $\alpha \in (0, \bar{\alpha})$,*

$$d_W\left((X^{(\alpha)} - x^*)/\alpha^{1/h}, Y\right) \leq U_5 \alpha^{1/h}, \qquad (17)$$

*where $U_5$ is given by* (140).

Proposition 5.5 provides a proper scaling function $g(\alpha) = \alpha^{\frac{1}{h}}$ for general objectives satisfying Assumption 5.1, and characterizes the limiting distribution of the scaled steady state $Y^{(\alpha)}$ by a non-Gaussian Gibbs distribution $\pi$. It also quantifies the pre-limit approximation error in Wasserstein distance as $\mathcal{O}(\alpha^{1/h})$ that vanishes as $\alpha \to 0$. To the best of our knowledge, Proposition 5.5 is the first constant-stepsize

steady-state diffusion approximation result for SA/SGD that yields a non-Gaussian limiting law, together with an explicit Wasserstein error bound. It provides theoretical support for the scaling and non-Gaussian limits observed numerically in (Chen et al., 2022), going beyond the classical Gaussian regime for smooth strongly convex objectives.

In Section E.2, we empirically test the two main predictions: (i) the scaling $\alpha^{1/h}$ and (ii) convergence to the Gibbs law. The experiments provide additional evidence supporting the stability and Stein-solution regularity postulated in Conjectures 5.3–5.4.

At a technical level, we bound the Wasserstein distance using the derivative bounds in Conjecture 5.4, together with the moment control in Conjecture 5.3. This demonstrates that our Stein framework extends beyond Gaussian limits and can accommodate diffusion approximations with non-quadratic potentials.

## 6. Future Work

There are several possible avenues for future work. A natural next step is to complete the proof of Proposition 5.5. The main challenges lie in obtaining regularity bounds for the solution to the associated Stein equation in this general case, as well as establishing moment bounds on the steady state. One potential approach is to explore the iterative method in (Döbler et al., 2017). In addition, our numerical experiments (see Section E.2)indicate that the tail bounds in main theorems may be overly conservative, especially for small values of $a$. Refining the dependence on $a$ in the tail bounds, potentially by bounding Wasserstein-$p$ distances for $p > 1$ (Wang & Maguluri, 2026), could yield sharper results. Finally, analyzing the tightness of the dimension-dependence in our Wasserstein bounds is another promising direction. Specifically, whether the polynomial dependence on dimension $d$ in all the constants can be improved to logarithmic dependence would clarify the curse of dimensionality in constant-stepsize SA.

## Impact Statement

This paper advances the theory of constant-stepsize stochastic approximation (including SGD and RL algorithms) by giving explicit pre-limit stepsize guarantees for steady-state fluctuations and tail behavior. These results may support more fine-grained tuning and reliability analysis of long-run ML/RL training dynamics. The work is methodological and does not involve new datasets or deployments; we do not anticipate ethical or societal impacts beyond the well-known downstream consequences of improving widely used ML optimization tools.

## Acknowledgements

This work is partially supported by NSF grants EPCN-2144316 and CPS-2240982.

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

# A. Structure of the Appendix

In this appendix, we describe the framework we use to obtain our results, and present proofs. The Appendix is structured as follows:

- Section B contains an overview of related literature.

- Section C introduces presents the necessary prerequisites for Stein's method and our analysis under iid noise. The proofs of 3.5, 3.7, and 3.11 are contained here.

- Section D contains the proofs of our results in the case of Markovian noise, proving 4.3.

- Section E provides a discussion of the limiting law and behavior of general Convex functions. In it, we prove 5.5 given the assumptions in 5 and also provide numerical simulations to justify our conjectures.

# B. Related Literature

**Background**   Stochastic approximation (SA) was introduced by Robbins and Monro as a method for solving fixed-point and stochastic optimization problems such as $\mathbb{E}[h(x, \xi)] = 0$ and $\min_x \mathbb{E}[\ell(x, \xi)]$, where only noisy samples are available (Robbins & Monro, 1951; Kushner & Yin, 2003; Nemirovski et al., 2009). This template includes classical SA recursions and stochastic gradient descent (SGD) used in large-scale learning (Kiefer & Wolfowitz, 1952; Bottou et al., 2018; Kushner & Yin, 2003). It also underlies widely used reinforcement-learning pipelines, where policy improvement relies on stochastic-gradient updates driven by noisy temporal-difference, including RLHF-style post-training for large language models (Tsitsiklis, 1994; Schulman et al., 2017; Ouyang et al., 2022). On the theory side, the SA literature is commonly developed along several complementary routes. The first route is convergence and performance guarantee via martingale/ODE methods and finite-time bounds. The second is fluctuation theory and distributional approximations for the iterations, especially CLT-type asymptotic normality under scaling. Our work follows the CLT/fluctuation route, particularly the Gaussian approximation for the *stationary* law of constant-stepsize SA. Thus, in the following we survey the relevant literature on fluctuation theory first, and then discuss the broader SA context.

**Fluctuation theory and distributional approximations for SA/SGD.**   A useful way to organize the "CLT / fluctuation" literature is along the limiting routes in Figure (1) as follows.

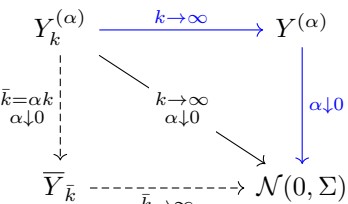

*Figure 1.* Fluctuation theory and distributional approximation diagram.

For the blue route, which is the route taken in this work, the literature typically studies constant stepsize $\alpha > 0$ first, and then lets $\alpha \downarrow 0$ after reaching stationarity. Such route approximates the stationary law of SA iterations $X_k$ when $k \to \infty$ by, e.g., a Gaussian fluctuation around the true solution, under small stepsizes regimes. Specifically, Mandt et al. (2017) popularizes the Markov chain viewpoint for constant stepsize SGD and models the noise $\xi_k$ as near Gaussian noise with nearly constant covariance. Under this condition, the stationary law of $Y_k$ is approximately Gaussian with covariance characterized by the Lyapunov equation (5). Dieuleveut et al. (2020) further exhibits the asymptotic normality of the stationary law for this SA Markov chain. They study constant-step SGD under strong convexity and utilize Poisson equation techniques to achieve a clear $\alpha$ dependency on stationary statistics, i.e., for sufficiently smooth functions $g$, $\mathbb{E}[g(X^{(\alpha)})] = g(x_*) + \alpha C_1(g)$. Meanwhile, Bianchi et al. (2022) studies minimizing a nonsmooth, nonconvex objective $f(x) = \mathbb{E}[\ell(x, \xi)]$, where each update uses a sample subgradient of $\ell(\cdot, \xi)$. They show that when $\alpha$ is small, the continuous-time interpolation of the iterates behaves like the differential inclusion $\dot{x} \in -\partial f(x)$, and the stationary distributions of the discrete algorithm converge (in a

set-valued sense) to invariant measures of this limiting inclusion, so the iterates tend to concentrate on the set of critical points, rather than exhibiting a Gaussian fluctuation limit. Building on the OU heuristics for smooth strongly convex SGD and contractive SA dynamics, Chen et al. (2022) directly show the weak convergence of the normalized stationary law $Y^{(\alpha)}$ to a Gaussian limit as $\alpha \downarrow 0$ under a conjecture of uniqueness for the solution to the PDE characterizing the stationary law. For nonsmooth contractive SA, Zhang et al. (2024) establish pre-limit coupling and steady-state convergence with relaxed differentiability assumptions on function $f$. They solve the uniqueness conjecture in (Chen et al., 2022) by metrizing the weak convergence using Wasserstein metric, i.e., $W_2(\mathcal{L}(Y^{(\alpha)}), \mathcal{L}(Y)) = o(1)$ with some limiting distribution $\mathcal{L}(Y)$ for general nonsmooth contractive SA. Yet they lack providing a convergence rate between $Y^{(\alpha)}$ and its Gaussian limit with explicit dependency on $\alpha$ and dimension $d$. Thus they cannot be directly used to quantify the approximation error and guide algorithm design with fixed $\alpha$ in practice. By contrast, our work fills this gap by combining SA dynamics with derivative bounds for multivariate Gaussian Stein's method (Chen et al., 2011; Ross, 2011; Döbler et al., 2021). We obtain explicit approximation error bounds in Wasserstein distance with clear $\alpha$ and $d$ dependencies under standard smoothness that are common in SA theory, and standard moment assumptions comparable to those in classical CLT results (Petrov, 2012).

For the diagonal route, a classical line of work studies *decreasing* stepsizes $\{\alpha_k\}_{k \geq 0}$, where $\alpha_k \downarrow 0$ and the iterate converges to a stable root $x^\star$. Typically, one proves a similar central limit theorem (CLT) but under scaling $1/\sqrt{\alpha_k}$. (e.g., $\sqrt{k}(x_k - x^\star) \Rightarrow \mathcal{N}(0, \Sigma)$ when $\alpha_k \asymp 1/k$), with $\Sigma$ characterized via a similar Lyapunov-type equation involving the Jacobian and the noise covariance. Chung (1954) is among the first to establish a Gaussian fluctuation limit under diminishing step in one-dimensional setting; Sacks (1958) derives an asymptotic normal law (with an explicit variance characterization) for a broad class of stochastic approximation procedures; and Fabian (1968) shows a Lyapunov-type characterization of the limiting covariance. These CLT-type results and their variants are developed systematically in the monographs (Benveniste et al., 2012) and (Kushner & Yin, 2003), which provide unified martingale/Poisson-equation frameworks. Several extensions refine the diagonal-route CLT. Pelletier (1999) strengthens weak convergence to an *almost sure* central limit theorem for stochastic approximation, showing (under suitable conditions) that the empirical distribution of normalized errors converges almost surely to the Gaussian limit. Fort (2015) replaces the usual i.i.d. or martingale-difference noise assumption by Markovian noise and proves a CLT for the normalized error via a Poisson-equation approach, allowing even multiple locally attracting equilibria $\{x_j^\star\}$. Also for Markov chain CLT, Yu et al. (2020) studies nonsmooth, nonconvex settings under dissipativity-type conditions which ensures ergodicity. They establish a CLT for time averages at fixed $\alpha$. While much of the prior work focuses on asymptotic Gaussian limits, Wei et al. (2025) provides a pre-limit Gaussian approximation for finite-$k$ iterates of constants stepsize SGD (yet they still follow the diagonal route in Figure 1) by deriving a Berry–Esseen type bound taking supremum over convex set. Though their results can be adapted to our setting after taking $k \to \infty$ while keeping $\alpha$ fixed, our focus is different. We cover a broader class of stochastic approximation schemes beyond SGD, and we obtain quantitative approximation for steady state in a non-uniform version of Berry-Esseen type of bound, in contrast with the convex-set Berry–Esseen bounds in (Wei et al., 2025). Combined with the rate of convergence towards steady state in $\mathcal{W}_2$ metric, e.g., in (Huo et al., 2023), our Wasserstein bound in all theorem above can achieve finite time version, i.e., $d_W(Y_k^{(\alpha)}, Y)$, comparable with (Wei et al., 2025). Another contemporary work (Kong & Srikant, 2026) uses a sliding window argument to show pre-limit CLT error for diminishing stepsize. They group the noise and compares to discretized OU process. The key requirement is that the cumulative step size on each group still vanishes as the group size grows, which relies on $\alpha_k \to 0$. Thus, their technique is specific to the diminishing-step-size regime (**?**)Assumption 1]kong2026finitesamplewassersteinerrorbounds

Overall, the asymptotic normality of diagonal route is mainly driven by $k \to \infty$ rather than by the $\alpha \downarrow 0$, which can be viewed as extension to classical CLT since the noise averages out over time. The analytical tools and practical guidance are different from ours.

The lower-left dashed (time-rescaled) route often develops trajectory-level approximations of SA by continuous-time diffusions limits. A representative viewpoint is the stochastic modified equation, which builds an SDE whose flow matches the discrete algorithm and yields pathwise trajectory limit (Li et al., 2019; An et al., 2020). Such diffusion approximations have also been developed for nonconvex SGD (Hu et al., 2017). More recently, Wang et al. (2025) quantifies the approximation error between the SA path and its diffusion limit with explicit functional error bounds in probability metrics. Though these works are powerful for temporal scaling limits, they do not address the fixed-$\alpha$ *stationary* law and therefore are not directly applicable to our goal of quantitative Gaussian approximation for the stationary distribution.

**Convergence, stability and performance guarantees.** A broad line of work provides convergence analysis for SA/SGD-type recursions. One viewpoint models SGD as an Euler-type discretization of an underlying SDE with time step $\alpha$, and

studies the resulting discretization error from numerical SDEs (Talay, 1990; Mattingly et al., 2002). Complementarily, foundational SA theory analyzes the discrete recursion directly, establishing almost sure convergence and stability via martingale arguments and the functional analysis, with further refinements from dynamical-systems perspectives (Ljung, 2003; Benaïm, 2006). A central theme is stability under verifiable conditions and dependent (e.g., Markovian) data, often through Lyapunov/drift criteria for controlled Markov-chain driven SA (Andrieu et al., 2005) and general convergence theorems for stochastic algorithms (Delyon, 2002). Beyond asymptotics, modern nonasymptotic analyses give explicit finite-time moment/MSE bounds for SA under Markovian noise (notably for linear SA/TD learning) (Srikant & Ying, 2019), while averaging variants (time/iterate averaging and Polyak–Ruppert-type schemes) remain standard tools for improving robustness and statistical efficiency (Ruppert, 1988; Polyak & Juditsky, 1992; Lai & Robbins, 1979; Fabian, 1968).

## C. Proofs under iid Noise

In this section, we first sketch the proof of our main results, Theorem 3.5, 3.7, and 3.11. Then, we introduce the preliminaries of Stein's method, which is the main tool we use to prove our theorems. Finally, we give the detailed proofs of our theorems.

### C.1. Proof Sketch

We summarize the core steps used to prove Theorems 3.5–3.11, in the following points,

1. We establish the existence of stationary distribution and uniform moment bounds for the rescaled chain $Y^{(\alpha)}$. We then identify the limiting distribution as $\alpha \downarrow 0$. From (Chen et al., 2022), the limiting distribution of $Y^{(\alpha)}$ as $\alpha \downarrow 0$ should satisfy the implicit equation (18). We first identify the limit by resolving uniqueness conjecture in (Chen et al., 2022) and showing that the equation has a unique Gaussian solution with covariance solving the Lyapunov equation (see Proposition C.1).

2. Suggested by the limit characterization, we employ Stein's method to quantify the distance between $Y^{(\alpha)}$ and $Y$ in terms of Wasserstein-1 distance. We first develop preliminaries on the Stein's method, particularly the Stein equation for the OU process and regularity bounds on its solution (Lemma C.2 and Proposition C.3). We also introduce the one-step operator for the stationary discrete chain (Proposition C.5) as the Stein operator for the discrete chain. Therefore, we can adapt the generator coupling framework to compare the two Stein operators in order to bound the Wasserstein-1 distance between the two distributions.

3. Building upon the Stein's method framework, we establish a general purpose theorem that bounds the Wasserstein-1 distance between the stationary distribution of a discrete Markov chain and that of an OU process (Theorem C.7). We first reduce the Wasserstein-1 distance bound to bounding the expectation of the OU operator applied to the Stein solution $g_h$ for any 1-Lipschitz test function $h$. We add the one-step operator of the SA iterates and employ Taylor expansion to decompose the error into several terms. Finally, we bound each term using the regularity bounds of $g_h$, the assumptions on the SA dynamics, and the uniform moment bounds of the SA iterates.

4. With the general purpose theorem in place, we specialize it to our SA setting under iid noise, e.g., Theorem 3.5. We verify the assumptions of Theorem C.7 under different SA modeling, i.e., SGD, linear SA, and contractive SA. We then apply Theorem C.7 to obtain the Wasserstein-1 distance bounds.

5. Finally, we build upon the Wasserstein-1 distance bounds to derive the tail bounds via a concentration argument (Lemma C.8). We optimize the parameters in the tail bound to obtain the best possible rate.

6. We provide an alternative argument at the end of the whole section, which provides $\mathcal{O}(\sqrt{\alpha})$ Wasserstein bound for SGD model when the objective function is $f : \mathbb{R} \to \mathbb{R}, x \mapsto x^2/2$. The bound in this one dimension case relax the logarithmic factor $\log(1/\alpha)$ in Wasserstein bounds for high dimension.

### C.2. Solution to the Uniqueness Conjecture

Here we revisit the uniqueness conjecture in (Chen et al., 2022) on the solution to the following PDE characterizing the limiting distribution of the centered-scaled steady-state iterates $Y^{(\alpha)}$ as $\alpha \downarrow 0$. It is shown in (Chen et al., 2022) that any limit point $Y$ of $Y^{(\alpha)}$ as $\alpha \downarrow 0$ must satisfy the following equation,

$$\mathbb{E}\left[\left(t^\top \Sigma t + 2it^\top H_f Y\right) e^{it^\top Y}\right] = 0, \quad \forall t \in \mathbb{R}^d, \tag{18}$$

Thus, solving the uniqueness of the solution to the above equation would identify the limit distribution. In this section, we resolve this conjecture by directly characterizing the PDE solution using techniques similar to those in (Barbour, 1990).

**Proposition C.1.** *Suppose there is a random variable Y which satisfies the following equation*

$$\mathbb{E}\left[\left(t^\top \Sigma t + 2it^\top H_f Y\right) e^{it^\top Y}\right] = 0, \quad \forall t \in \mathbb{R}^d, \tag{19}$$

*then Y is gaussian with mean 0 and covariance $\Sigma_Y$ where*

$$H_f \Sigma_Y + \Sigma_Y H_f = \Sigma$$

*Proof.* Let $\phi(t) = \mathbb{E}[e^{it^\top Y}]$ for $t \in \mathbb{R}^d$. Then $\phi : \mathbb{R}^d \to \mathbb{C}$ is differentiable and $\nabla_t \phi(t) \in \mathbb{C}^d$. The identity

$$t^\top \Sigma t\, \phi(t) + 2\, t^\top H_f\, \nabla_t \phi(t) = 0$$

can be written as the first-order linear PDE

$$b(t)^\top \nabla_t \phi(t) + c(t)\phi(t) = 0, \qquad b(t) := 2H_f^\top t, \quad c(t) := t^\top \Sigma t.$$

We solve it by the method of characteristics (see (Evans, 2022, Chapter 3.2)). Fix $t_0 \in \mathbb{R}^d$ and let $t(\cdot)$ solve the characteristic ODE

$$\frac{dt(s)}{ds} = b(t(s)) = 2H_f^\top t(s), \qquad t(0) = t_0.$$

Since $b$ is linear, the unique solution is

$$t(s) = e^{2sH_f^\top} t_0.$$

Along this curve, by the chain rule,

$$\frac{d}{ds}\phi(t(s)) = \nabla_t \phi(t(s))^\top \frac{dt(s)}{ds} = \nabla_t \phi(t(s))^\top b(t(s)) = -c(t(s))\,\phi(t(s)),$$

where the last equality uses the PDE evaluated at $t = t(s)$. Hence $\phi(t(s))$ satisfies the scalar linear ODE

$$\frac{d}{ds}\phi(t(s)) = -c(t(s))\phi(t(s)),$$

whose unique solution is

$$\phi(t(s)) = \phi(t_0)\exp\left(-\int_0^s c(t(u))\,du\right), \qquad \text{equivalently} \qquad \phi(t_0) = \phi(t(s))\exp\left(\int_0^s c(t(u))\,du\right).$$

Since $H_f$ is positive definite, $e^{2sH_f^\top} t_0 \to 0$ as $s \to -\infty$ for any fixed $t_0$, so $t(s) \to 0$ as $s \to -\infty$. Also $\phi(0) = 1$. Letting $s \to -\infty$ gives

$$\phi(t_0) = \exp\left(-\int_{-\infty}^0 c(t(u))\,du\right) = \exp\left(-\int_{-\infty}^0 t(u)^\top \Sigma t(u)\,du\right).$$

Using $t(u) = e^{2uH_f^\top} t_0$, we obtain

$$\phi(t_0) = \exp\left(-\int_{-\infty}^0 (e^{2uH_f^\top} t_0)^\top \Sigma(e^{2uH_f^\top} t_0)\,du\right) = \exp\left(-t_0^\top\left[\int_{-\infty}^0 (e^{2uH_f})\Sigma(e^{2uH_f^\top})\,du\right]t_0\right).$$

With the change of variables $v = -2u$ (so $u = -v/2$, $du = -dv/2$),

$$\int_{-\infty}^0 (e^{2uH_f})\Sigma(e^{2uH_f^\top})\,du = \frac{1}{2}\int_0^\infty e^{-vH_f}\Sigma e^{-vH_f^\top}\,dv.$$

Define $\Sigma_Y$ as the unique positive definite solution of the Lyapunov equation

$$H_f \Sigma_Y + \Sigma_Y H_f^\top = \Sigma.$$

Then it is standard (and can be verified by differentiating $e^{-vH_f}\Sigma_Y e^{-vH_f^\top}$ and integrating from 0 to $\infty$) that

$$\Sigma_Y = \int_0^\infty e^{-vH_f}\,\Sigma\,e^{-vH_f^\top}\,dv,$$

and therefore

$$\int_{-\infty}^0 (e^{2uH_f})\,\Sigma\,(e^{2uH_f^\top})\,du = \frac{1}{2}\,\Sigma_Y.$$

Plugging back yields

$$\phi(t) = \exp\left(-\frac{1}{2}\,t^\top \Sigma_Y t\right),$$

which is the characteristic function of $\mathcal{N}(0, \Sigma_Y)$.

Finally, uniqueness: the characteristic ODE $t'(s) = 2H_f^\top t(s)$ has a unique solution for each initial condition $t(0) = t_0$. Hence characteristics cannot cross: if $t_1(\cdot)$ and $t_2(\cdot)$ satisfy $t_1(s_1) = t_2(s_2)$ for some $s_1, s_2$, then

$$t_1(0) = e^{-2s_1 H_f^\top} t_1(s_1) = e^{-2s_1 H_f^\top} t_2(s_2) = e^{2(s_2 - s_1)H_f^\top} t_2(0),$$

so the two points lie on the same trajectory. Therefore the PDE solution determined by the boundary condition $\phi(0) = 1$ is unique, and $Y \sim \mathcal{N}(0, \Sigma_Y)$. $\qquad\square$

## C.3. Preliminaries: Stein's Method and Technical Lemmas

In this section, we introduce the necessary preliminaries of Stein's method that will be used in our analysis. We will also state some technical lemmas regarding the generator comparison framework. We first introduce the high dimensional Ornstein-Uhlenbeck (OU) Process and its generator, stationary distribution. We will then present a key proposition regarding the regularity bounds on the solution to the Stein equation associated with the OU process.

**Lemma C.2.** *Let $J \in \mathbb{R}^{d\times d}$ be a Hurwitz matrix (all eigenvalues have strictly negative real parts), and $\Sigma \in \mathbb{R}^{d\times d}$ be a symmetric positive definite matrix. Consider the Ornstein–Uhlenbeck (OU) process defined by*

$$\mathrm{d}X_t = JX_t\,\mathrm{d}t + \Sigma^{1/2}\,\mathrm{d}W_t, \quad X_0 \in \mathbb{R}^d, \tag{20}$$

*where $W_t$ is a $d$-dimensional standard Brownian motion. Then the following hold:*

*(i)* *The infinitesimal generator of the process is given by*

$$\mathcal{L}f(x) = \langle Jx, \nabla f(x)\rangle + \frac{1}{2}\operatorname{Tr}(\Sigma\nabla^2 f(x)) \tag{21}$$

*for all sufficiently regular test functions $f : \mathbb{R}^d \to \mathbb{R}$.*

*(ii)* *The unique stationary distribution of the OU process is Gaussian:*

$$X_\infty \sim \mathcal{N}(0, \Sigma_Y), \tag{22}$$

*where $\Sigma_Y \in \mathbb{R}^{d\times d}$ is the unique symmetric positive definite solution to the Lyapunov equation*

$$J\Sigma_Y + \Sigma_Y J^\top = \Sigma. \tag{23}$$

*(iii)* *For a random vector $Z$ with law $\mathcal{L}(Z) = \mathcal{N}(0, \Sigma_Y)$, the Stein characterization holds:*

$$\mathbb{E}[\mathcal{L}f(Z)] = 0 \quad \text{for all } f : \mathbb{R}^d \to \mathbb{R}, \ f \in C^2. \tag{24}$$

*Conversely, if a random vector $X$ satisfies $\mathbb{E}[\mathcal{L}f(X)] = 0$ for all $f$ in an appropriately rich class, then $X \sim \mathcal{N}(0, \Sigma_Y)$.*

We now present the regularity bounds for the solution to the Stein equation associated with the OU process defined in Lemma C.2. These regularity bounds are crucial for our analysis using Stein's method.

**Proposition C.3.** *Consider the Stein equation associated with the OU process defined in Lemma C.2:*

$$\mathcal{L}f_h(y) = h(y) - \mathbb{E}[h(Z)], \tag{25}$$

*where $Z \sim \mathcal{N}(0, \Sigma_Y)$ and $\mathcal{L}$ is the generator of the OU process, i.e., $dX_t = -JX_t\,dt + \Sigma^{1/2}\,dW_t$. Let $\Sigma \in \mathbb{R}^{d \times d}$ be a symmetric positive definite matrix, and $J$ be a Hurwitz matrix, i.e., all the eigenvalues of $J$ have real parts bounded above by $-\lambda_{min} < 0$, and let $\Sigma_Y$ be the unique positive definite solution to the Lyapunov equation $J\Sigma_Y + \Sigma_Y J^\top = \Sigma$. Let $h : \mathbb{R}^d \to \mathbb{R}$ is a Lipschitz function with Lipschitz constant $1$. We first define the constant*

$$K_Y := \|\Sigma_Y^{1/2}\|_{\mathrm{op}} \cdot \|\Sigma_Y^{-1/2}\|_{\mathrm{op}}. \tag{26}$$

*Then, we define the set of solution to the Stein equation as*

$$G(J, \Sigma) := \{f_h : \mathbb{R}^d \to \mathbb{R} \mid \langle Jx, \nabla f_h(x)\rangle + \frac{1}{2}\operatorname{Tr}\big(\Sigma\nabla^2 f_h(x)\big) = h(y) - \mathbb{E}[h(Z)], \text{ for } h \in Lip_1 \cap C^1\}. \tag{27}$$

*Then, for any $f_h \in G(J, \Sigma)$, the following regularity bounds hold:*

1. *(First derivative bound) For any $y \in \mathbb{R}^d$,*

$$\|\nabla f_h(y)\|_2 \leq \underbrace{K_Y \cdot \frac{2}{\lambda}}_{:=g_{1,Y}}, \tag{28}$$

2. *(Second derivative bound) For any $y \in \mathbb{R}^d$,*

$$\|\nabla^2 f_h(y)\|_{\mathrm{op}} \leq \underbrace{\frac{2}{\lambda}\sqrt{\frac{2}{\pi}}\, K_Y^2\, \|\Sigma_Y^{-1/2}\|_{\mathrm{op}}}_{:=g_{2,Y}}, \tag{29}$$

*where $\|\cdot\|_{\mathrm{op}}$ denotes the operator norm induced by the Euclidean norm.*

3. *(Hessian Hölder bound) For any $\beta \in (0,1)$, there exists a constant $C_\beta > 0$ depending only on $\beta$ such that*

$$\|\nabla^2 f_h(y) - \nabla^2 f_h(x)\|_{\mathrm{op}} \leq g_{3,Y}\frac{1}{1-\beta} \cdot \|y - x\|_2^\beta, \tag{30}$$

*for all $x, y \in \mathbb{R}^d$, where $g_{3,Y}$ is defined in (37).*

*Proof.* The proof largely follows (Gorham & Mackey, 2015; Gallouët et al., 2018). We consider the following SDE and its associated semigroup.

$$dY_t = JY_t\,dt + \Sigma^{1/2}\,dW_t,$$

With $Y_0 = Y$ being a random vector independent of the Brownian motion $W_t$. And $J$ is a Hurwitz matrix, i.e., all eigenvalues of $J$ have strictly negative real parts such that the Lyapunov equation

$$J\Sigma_Y + \Sigma_Y J^\top + \Sigma = 0 \tag{31}$$

has a unique positive definite solution $\Sigma_Y$. The stationary distribution of the SDE is $\mathcal{N}(0, \Sigma_Y)$. And the associated semigroup is defined as

$$P_t h(y) := \mathbb{E}_{\eta_t}\big[h\big(e^{tJ}y + \eta_t\big)\big],$$

where $\eta_t \sim \mathcal{N}(0, Q_t)$ and $Q_t := \int_0^t e^{sJ}\Sigma\, e^{sJ^\top}\,ds$. Note that $\eta_t$ is independent of $Y$. Now we consider the Stein equation associated with the SDE:

$$\mathcal{L}f_h(y) = h(y) - \mathbb{E}[h(Z)], \tag{32}$$

where $Z \sim \mathcal{N}(0, \Sigma_Y)$ and $\mathcal{L}$ is the generator of the SDE defined as

$$\mathcal{L}f(y) = \langle Jy, \nabla f(y)\rangle + \frac{1}{2}\operatorname{Tr}(\Sigma\nabla^2 f(y)).$$

The solution to the Stein equation (32) is given in (Gallouët et al., 2018) by

$$f_h(y) := -\int_0^\infty \Big(P_t h(y) - \mathbb{E}[h(Z)]\Big)\, dt. \tag{33}$$

Our ultimate goal is to present the regularity bounds for the solution $f_h$ to the Stein equation (32). To proceed, we first present some useful properties regarding the SDE and the semigroup $P_t$, which combines the Lyapunov equation and the properties of the OU process.

**(a)** We first show that $Q_t = \Sigma_Y - e^{tJ}\Sigma_Y e^{tJ^\top}$. Define $M_t := e^{tJ}\Sigma_Y e^{tJ^\top}$ for $t \geq 0$. By differentiating with respect to $t$ and using the Lyapunov equation $J\Sigma_Y + \Sigma_Y J^\top = -\Sigma$, we obtain

$$\frac{d}{dt}M_t = e^{tJ}(J\Sigma_Y + \Sigma_Y J^\top)e^{tJ^\top} + e^{tJ}\Sigma_Y J^\top e^{tJ^\top}$$
$$= -e^{tJ}\Sigma e^{tJ^\top}.$$

Integrating both sides from $0$ to $t$ and using the boundary condition $M_0 = \Sigma_Y$, we obtain

$$M_t - \Sigma_Y = -\int_0^t e^{sJ}\Sigma e^{sJ^\top}\, ds = -Q_t,$$

which yields a desired relation $Q_t = \Sigma_Y - M_t = \Sigma_Y - e^{tJ}\Sigma_Y e^{tJ^\top}$.

**(b)** We establish decay bounds for the semigroup $e^{tJ}$ and upper bounds for $Q_t$ in terms of $\Sigma_Y$.

For any $u \in \mathbb{R}^d$, we compute:

$$-u^\top\big(\tfrac{d}{dt}M_t\big)u = u^\top e^{tJ}\Sigma e^{tJ^\top}u$$
$$= \big(\Sigma_Y^{-1/2}e^{tJ^\top}u\big)^\top\big(\Sigma_Y^{-1/2}\Sigma\Sigma_Y^{-1/2}\big)\big(\Sigma_Y^{-1/2}e^{tJ^\top}u\big)$$
$$\geq \lambda_{\min}\big(\Sigma_Y^{-1/2}\Sigma\Sigma_Y^{-1/2}\big)\cdot u^\top e^{tJ}\Sigma_Y e^{tJ^\top}u,$$

where we define

$$\lambda := \lambda_{\min}\big(\Sigma_Y^{-1/2}\Sigma\Sigma_Y^{-1/2}\big) > 0.$$

This yields the differential inequality

$$\frac{d}{dt}M_t \preceq -\lambda M_t.$$

By Grönwall's inequality applied componentwise, we obtain

$$M_t \preceq e^{-\lambda t}\Sigma_Y, \qquad \text{hence} \qquad Q_t = \Sigma_Y - M_t \preceq (1 - e^{-\lambda t})\Sigma_Y.$$

For the operator norm of $e^{tJ}$, we use the relation

$$\|e^{tJ}\|_{\mathrm{op}} \leq K_Y\, e^{-\lambda t/2},$$

where

$$K_Y := \|\Sigma_Y^{1/2}\|_{\mathrm{op}}\cdot\|\Sigma_Y^{-1/2}\|_{\mathrm{op}}.$$

These bounds control the decay of the semigroup and ensure the convergence of the integral representations used in subsequent estimates.

Having established these preliminary bounds, we now proceed to derive the regularity estimates for the Stein solution $f_h$. For the two key constants $K_Y$ and $\lambda$, we organize their definitions here for clarity:

$$K_Y := \|\Sigma_Y^{1/2}\|_{\mathrm{op}}\cdot\|\Sigma_Y^{-1/2}\|_{\mathrm{op}}, \quad \lambda := \lambda_{\min}\big(\Sigma_Y^{-1/2}\Sigma\Sigma_Y^{-1/2}\big) > 0.$$

**First-order gradient bound.** For the first-order derivative of the Stein solution $f_h$, we employ the integral representation in (33). By differentiating under the integral sign (which is justified by dominated convergence), we obtain:

$$\nabla f_h(y) = -\int_0^\infty \nabla P_t h(y)\, dt,$$

recalling the semigroup derivative is given by

$$\nabla P_t h(y) = \mathbb{E}\big[\nabla h\big(e^{tJ}y + \eta_t\big)\, e^{tJ}\big],$$

with $\eta_t \sim \mathcal{N}(0, Q_t)$ independent of $y$.

To bound the norm, we use the triangle inequality and properties of conditional expectations:

$$
\begin{aligned}
\|\nabla f_h(y)\|_2 &\le \int_0^\infty \big\|\mathbb{E}[\nabla h(e^{tJ}y + \eta_t)e^{tJ}]\big\|_2\, dt \\
&\le \int_0^\infty \mathbb{E}\big[\|\nabla h(e^{tJ}y + \eta_t)\|_2 \cdot \|e^{tJ}\|_{\mathrm{op}}\big]\, dt \\
&\le \int_0^\infty \mathrm{Lip}(h) \cdot \|e^{tJ}\|_{\mathrm{op}}\, dt.
\end{aligned}
$$

By the exponential decay bound from Section C.3, we have $\|e^{tJ}\|_{\mathrm{op}} \le K_Y e^{-\lambda t/2}$ for all $t \ge 0$, where $\lambda = \lambda_{\min}(\Sigma_Y^{-1/2}\Sigma\Sigma_Y^{-1/2}) > 0$. Therefore,

$$\|\nabla f_h\|_\infty := \sup_{y\in\mathbb{R}^d} \|\nabla f_h(y)\|_2 \le \mathrm{Lip}(h) \cdot \int_0^\infty K_Y e^{-\lambda t/2}\, dt \tag{34}$$

$$= \mathrm{Lip}(h) \cdot K_Y \cdot \frac{2}{\lambda}, \tag{35}$$

which is independent of the point $y$ and depends only on the Lipschitz constant of $h$, the Lyapunov solution $\Sigma_Y$, and the spectral gap $\lambda$ of the Lyapunov operator.

**Second-order derivative bound.** We now turn to the second-order derivative bound for the Stein solution $f_h$. For this, we first define
$$\Phi_t(m) := \mathbb{E}\big[h(m + \eta_t)\big],$$
which allows us to express
$$P_t h(y) = \Phi_t\big(e^{tJ}y\big).$$
Recalling that $h \in \mathrm{Lip}(1) \cap C^1$, we have $\nabla h$ is bounded. We will use Gaussian integration by parts to derive the second derivative bounds.

Gaussian Integration by Parts (IBP): We have the relation
$$\mathbb{E}\big[\partial_u g(m + \eta_t)\big] = \mathbb{E}\big[g(m + \eta_t)\,\langle Q_t^{-1}\eta_t, u\rangle\big],$$
for any $u \in \mathbb{R}^d$. Consequently, for unit vectors $\|u\|_2 = \|v\|_2 = 1$, it follows that

$$\partial_u\partial_v\Phi_t(m) = \mathbb{E}\Big[\partial_v h(m + \eta_t)\,\langle Q_t^{-1}\eta_t, u\rangle\Big].$$

Thus, we can derive the operator norm of the second derivative:

$$
\begin{aligned}
\|\nabla^2\Phi_t\|_{\mathrm{op},\infty} &\le \|\nabla h\|_\infty \cdot \mathbb{E}\big[|\langle Q_t^{-1}\eta_t, u\rangle|\big] \\
&\le \|\nabla h\|_\infty \cdot \|Q_t^{-1/2}u\|_2 \quad \Big(\text{since } \langle Q_t^{-1}\eta_t, u\rangle \sim \mathcal{N}\big(0, \|Q_t^{-1/2}u\|_2^2\big)\Big) \\
&\le \sqrt{\frac{2}{\pi}}\, \|\nabla h\|_\infty \cdot \|Q_t^{-1/2}\|_{\mathrm{op}},
\end{aligned}
$$

where the norm $\|\cdot\|_{\mathrm{op},\infty}$ denotes the supremum operator norm over all $m \in \mathbb{R}^d$. Therefore, we can bound the operator norm of the second derivative of $P_t h$:

$$\|\nabla^2 P_t h\|_{\mathrm{op},\infty} \leq \|e^{tJ}\|_{\mathrm{op}}^2 \|\nabla^2 \Phi_t\|_{\mathrm{op},\infty}$$

$$\leq \sqrt{\frac{2}{\pi}} \|\nabla h\|_\infty \|e^{tJ}\|_{\mathrm{op}}^2 \|Q_t^{-1/2}\|_{\mathrm{op}}$$

$$\leq \sqrt{\frac{2}{\pi}} \|\nabla h\|_\infty K_Y^2 e^{-\lambda t} (1 - e^{-\lambda t})^{-1/2} \quad \text{(by the paragraph (b) above).}$$

Consequently, we obtain the following bound for the operator norm of the second derivative of $f_h$:

$$\|\nabla^2 f_h\|_{\mathrm{op},\infty} \leq \int_0^\infty \|\nabla^2 P_t h\|_{\mathrm{op},\infty} \, dt$$

$$\leq \sqrt{\frac{2}{\pi}} \|\nabla h\|_\infty K_Y^2 \|\Sigma_Y^{-1/2}\|_{\mathrm{op}} \int_0^\infty e^{-\lambda t}(1 - e^{-\lambda t})^{-1/2} \, dt$$

$$= \sqrt{\frac{2}{\pi}} \|\nabla h\|_\infty K_Y^2 \|\Sigma_Y^{-1/2}\|_{\mathrm{op}} \cdot \frac{1}{\lambda} \int_0^1 (1 - u)^{-1/2} \, du$$

$$= \frac{2}{\lambda} \sqrt{\frac{2}{\pi}} \|\nabla h\|_\infty K_Y^2 \|\Sigma_Y^{-1/2}\|_{\mathrm{op}}.$$

**Hessian Hölder bound.** Finally, we derive the Hölder continuity bound for the Hessian of the Stein solution $f_h$. For any $x, y \in \mathbb{R}^d$, recall the integral representation of $f_h$:

$$f_h(x) - f_h(y) = -\int_0^\infty \big(P_t h(x) - P_t h(y)\big) \, dt.$$

By differentiating twice under the integral sign, we have

$$\nabla^2 f_h(x) - \nabla^2 f_h(y) = -\int_0^\infty \big(\nabla^2 P_t h(x) - \nabla^2 P_t h(y)\big) \, dt.$$

We split the integral into two parts at a cutoff time $t_r > 0$, with $r = \|x - y\|_2$ and $t_r := \min\{1, r^2\}$. Thus, we bound the two integrals separately.

**Part 1: Integral over $[0, t_r]$.** For the first integral over $[0, t_r]$, we use the second derivative bound derived earlier

$$\big\|\nabla^2 P_t h(x) - \nabla^2 P_t h(y)\big\|_{\mathrm{op}} \leq 2\|\nabla^2 P_t h\|_{\mathrm{op},\infty}$$

$$\leq 2\sqrt{\frac{2}{\pi}} \|\nabla h\|_\infty K_Y^2 e^{-\lambda t}(1 - e^{-\lambda t})^{-1/2}$$

$$\leq 2\sqrt{\frac{2}{\pi}} \|\nabla h\|_\infty K_Y^2 \cdot \frac{1}{\sqrt{\lambda t}}.$$

Integrating from $0$ to $t_r$, we obtain

$$\int_0^{t_r} \big\|\nabla^2 P_t h(x) - \nabla^2 P_t h(y)\big\|_{\mathrm{op}} \, dt \leq 2\sqrt{\frac{2}{\pi}} \|\nabla h\|_\infty K_Y^2 \int_0^{t_r} \frac{1}{\sqrt{\lambda t}} \, dt$$

$$= 4\sqrt{\frac{2}{\pi}} \|\nabla h\|_\infty K_Y^2 \cdot \frac{\sqrt{t_r}}{\sqrt{\lambda}}.$$

From the definition of $t_r$, if $r < 1$, then $\sqrt{t_r} \leq r \leq r^\beta$ for any $\beta \in (0, 1)$. If $r \geq 1$, then $\sqrt{t_r} \leq 1 \leq r^\beta$ for any $\beta \in (0, 1)$. Thus, we have

$$\int_0^{t_r} \big\|\nabla^2 P_t h(x) - \nabla^2 P_t h(y)\big\|_{\mathrm{op}} \, dt \leq 4\sqrt{\frac{2}{\pi}} \|\nabla h\|_\infty K_Y^2 \frac{1}{\sqrt{\lambda}} \cdot r^\beta.$$

For the second integral over $[t_r, \infty)$, we split further into two terms, one is $t \in [t_r, 1]$ and the other is $t \in [1, \infty)$. We first derive a bound for the difference of Hessians using integration by parts for Gaussian measures.

$$\nabla^2 \Phi_t(m) = \mathbb{E}\Big[\nabla h(m + \eta_t) \underbrace{(Q_t^{-1}\eta_t)(Q_t^{-1}\eta_t)^\top}_{:=A_t}\Big].$$

Thus, we have

$$
\begin{aligned}
&\left\|\nabla^2 P_t h(x) - \nabla^2 P_t h(y)\right\|_{\mathrm{op}} \\
&= \left\|e^{tJ}\big(\nabla^2 \Phi_t(e^{tJ}x) - \nabla^2 \Phi_t(e^{tJ}y)\big)e^{tJ^\top}\right\|_{\mathrm{op}} \\
&\leq \|e^{tJ}\|_{\mathrm{op}}^2 \cdot \left\|\nabla^2 \Phi_t(e^{tJ}x) - \nabla^2 \Phi_t(e^{tJ}y)\right\|_{\mathrm{op}} \\
&= \|e^{tJ}\|_{\mathrm{op}}^2 \left\|\mathbb{E}\Big[\big(\nabla h(e^{tJ}x + \eta_t) - \nabla h(e^{tJ}y + \eta_t)\big)A_t\Big]\right\|_{\mathrm{op}} \\
&\leq \|e^{tJ}\|_{\mathrm{op}}^2 \mathbb{E}\Big[\big\|\nabla h(e^{tJ}x + \eta_t) - \nabla h(e^{tJ}y + \eta_t)\big\|_2 \cdot \|A_t\|_{\mathrm{op}}\Big] \\
&\leq \|e^{tJ}\|_{\mathrm{op}}^3 \cdot \mathrm{Lip}(\nabla h) \cdot \|x - y\|_2 \cdot \mathbb{E}[\|A_t\|_{\mathrm{op}}] \\
&= \|e^{tJ}\|_{\mathrm{op}}^3 \cdot \mathrm{Lip}(\nabla h) \cdot \|x - y\|_2 \cdot \mathbb{E}[\|A_t\|_{\mathrm{op}}] \\
&\overset{(a)}{=} \|e^{tJ}\|_{\mathrm{op}}^2 \cdot \mathrm{Lip}(\nabla h) \cdot \|x - y\|_2 \cdot \mathbb{E}\big[\|Q_t^{-1/2}(WW^\top - I_d)Q_t^{-1/2}\|_{\mathrm{op}}\big] \\
&\leq \|e^{tJ}\|_{\mathrm{op}}^3 \cdot \mathrm{Lip}(\nabla h) \cdot \|x - y\|_2 \cdot \|Q_t^{-1/2}\|_{\mathrm{op}}^2 \cdot \mathbb{E}[\|WW^\top - I_d\|_{\mathrm{op}}] \\
&\leq \|e^{tJ}\|_{\mathrm{op}}^3 \cdot \|x - y\|_2 \cdot \|Q_t^{-1/2}\|_{\mathrm{op}}^2 \cdot \big(\mathbb{E}(\|W\|_2^2) + \|I_d\|_{\mathrm{op}}\big) \\
&= \|e^{tJ}\|_{\mathrm{op}}^3 \cdot \|x - y\|_2 \cdot \|Q_t^{-1/2}\|_{\mathrm{op}}^2 \cdot (d + 1),
\end{aligned}
\tag{36}
$$

Equality (a) follows by writing $\eta_t = Q_t^{1/2}W$ with $W \sim \mathcal{N}(0, I_d)$, and thus we can bound the second integral over $[t_r, \infty)$ by

$$
\begin{aligned}
\left\|\nabla^2 P_t h(x) - \nabla^2 P_t h(y)\right\|_{\mathrm{op}} &\leq (d + 1)\,\|x - y\|_2\,\|e^{tJ}\|_{\mathrm{op}}^3\,\|Q_t^{-1}\|_{\mathrm{op}} \\
&\leq (d + 1)\,r\,\big(K_Y e^{-\lambda t/2}\big)^3\,\|Q_t^{-1}\|_{\mathrm{op}}.
\end{aligned}
$$

We next bound $\|Q_t^{-1}\|_{\mathrm{op}}$ separately on $t \in [t_r, 1]$ and $t \geq 1$.

**Part 2-a: the moderate-time regime** $t \in [t_r, 1]$. Using the covariance lower bound $Q_t \succeq (1 - e^{-\lambda t})\Sigma_Y$, we have

$$\|Q_t^{-1}\|_{\mathrm{op}} \leq \frac{1}{(1 - e^{-\lambda t})\,\lambda_{\min}(\Sigma_Y)} \leq \frac{2}{\lambda\,\lambda_{\min}(\Sigma_Y)} \cdot \frac{1}{t}, \qquad t \in (0, 1],$$

where we used $1 - e^{-\lambda t} \geq \frac{\lambda t}{2}$ for $t \in (0, 1]$. Therefore, for $t \in [t_r, 1]$,

$$\left\|\nabla^2 P_t h(x) - \nabla^2 P_t h(y)\right\|_{\mathrm{op}} \leq (d + 1)\,r\,K_Y^3 e^{-3\lambda t/4} \cdot \frac{2}{\lambda\,\lambda_{\min}(\Sigma_Y)} \cdot \frac{1}{t}.$$

Since $e^{-3\lambda t/4} \leq 1$ on $[t_r, 1]$, we obtain

$$
\begin{aligned}
\int_{t_r}^1 \left\|\nabla^2 P_t h(x) - \nabla^2 P_t h(y)\right\|_{\mathrm{op}} dt &\leq (d + 1)\,r\,K_Y^3 \cdot \frac{2}{\lambda\,\lambda_{\min}(\Sigma_Y)} \int_{t_r}^1 \frac{dt}{t} \\
&= (d + 1)\,r\,K_Y^3 \cdot \frac{2}{\lambda\,\lambda_{\min}(\Sigma_Y)} \log\!\Big(\frac{1}{t_r}\Big).
\end{aligned}
$$

**Part 2-b: the large-time regime** $t \geq 1$. Again, $Q_t \succeq (1 - e^{-\lambda})\Sigma_Y$ for all $t \geq 1$, hence

$$\|Q_t^{-1}\|_{\mathrm{op}} \leq \frac{1}{(1 - e^{-\lambda})\,\lambda_{\min}(\Sigma_Y)}.$$

Therefore,

$$\int_1^\infty \left\|\nabla^2 P_t h(x) - \nabla^2 P_t h(y)\right\|_{\mathrm{op}} dt \le (d+1)\, r\, K_Y^3 \frac{1}{(1-e^{-\lambda})\,\lambda_{\min}(\Sigma_Y)} \int_1^\infty e^{-3\lambda t/4}\, dt$$

$$= (d+1)\, r\, K_Y^3 \frac{1}{(1-e^{-\lambda})\,\lambda_{\min}(\Sigma_Y)} \cdot \frac{4}{3\lambda}\, e^{-3\lambda/4}.$$

Combining the two sub-regimes yields

$$\int_{t_r}^\infty \left\|\nabla^2 P_t h(x) - \nabla^2 P_t h(y)\right\|_{\mathrm{op}} dt \le (d+1)\, r\, K_Y^3 \cdot \frac{2}{\lambda\,\lambda_{\min}(\Sigma_Y)} \log\!\Big(\frac{1}{t_r}\Big)$$

$$+ (d+1)\, r\, K_Y^3 \frac{1}{(1-e^{-\lambda})\,\lambda_{\min}(\Sigma_Y)} \cdot \frac{4}{3\lambda}\, e^{-3\lambda/4}.$$

Finally, since $t_r = \min\{1, r^2\}$, we have $\log(1/t_r) = 0$ if $r \ge 1$ and $\log(1/t_r) = 2\log(1/r)$ if $0 < r < 1$. Hence, for all $x, y \in \mathbb{R}^d$,

$$\int_{t_r}^\infty \left\|\nabla^2 P_t h(x) - \nabla^2 P_t h(y)\right\|_{\mathrm{op}} dt \le C_2\, r\Big(1 + \log^+ \frac{1}{r}\Big),$$

where $\log^+(u) := \max\{0, \log u\}$ and one may take

$$C_2 := (d+1)\, K_Y^3 \left[\frac{4}{\lambda\,\lambda_{\min}(\Sigma_Y)} + \frac{4}{3\lambda} \cdot \frac{e^{-3\lambda/4}}{(1-e^{-\lambda})\,\lambda_{\min}(\Sigma_Y)}\right].$$

Putting the bounds for the two integrals together, we conclude that

$$\left\|\nabla^2 f_h(x) - \nabla^2 f_h(y)\right\|_{\mathrm{op}} \le 4\sqrt{\frac{2}{\pi}}\, \|\nabla h\|_\infty\, K_Y^2 \frac{1}{\sqrt{\lambda}} \cdot r^\beta \; + \; C_2\, r\Big(1 + \log^+ \frac{1}{r}\Big).$$

In particular, for any $0 < \beta < 1$, using $r^1 \log^+(1/r) \le \frac{1}{1-\beta} r^\beta$ for $r \in (0,1]$, we finaly obtain the global $\beta$-Hölder estimate

$$\left\|\nabla^2 f_h(x) - \nabla^2 f_h(y)\right\|_{\mathrm{op}} \le g_{3,Y} \frac{1}{1-\beta} r^\beta,$$

for an explicit constant $g_{3,Y}$ defined as follows:

$$g_{3,Y} := 4\sqrt{\frac{2}{\pi}}\, \|\nabla h\|_\infty\, K_Y^2 \frac{1}{\sqrt{\lambda}} \; + \; C_2 \tag{37}$$

$$C_2 := (d+1)\, K_Y^3 \left[\frac{4}{\lambda\,\lambda_{\min}(\Sigma_Y)} + \frac{4}{3\lambda} \cdot \frac{e^{-3\lambda/4}}{(1-e^{-\lambda})\,\lambda_{\min}(\Sigma_Y)}\right]. \tag{38}$$

$\square$

Note that the function class $G(\cdot, \cdot)$ in (27), i.e., the solution to the Stein equation (32) is defined for $h \in \mathrm{Lip}_S \cap C^1(\mathbb{R}^d)$. The reason for choosing such a function class is that the Wasserstein 1 distance in dual form (3) requires $h$ to be Lipschitz continuous, and it can be further characterized by the following density lemma.

**Lemma C.4.** *For two random variable $X$ and $Y$, we have*

$$\sup_{h \in \mathrm{Lip}_S} \{\mathbb{E}[h(X) - h(Y)]\} = \sup_{h \in \mathrm{Lip}_S \cap C^1(\mathbb{R}^d)} \{\mathbb{E}[h(X) - h(Y)]$$

*Proof.* We know from (Luukkainen, 1979) that $\mathrm{Lip}_S$ is closed in the space of continuous functions $C(\mathbb{R}^d)$ and from (Stein & Shakarchi, 2005) that $C^1(\mathbb{R}^d)$ is dense in $C(\mathbb{R}^d)$. Which means for any $\epsilon > 0$ and any $h \in \mathrm{Lip}_S$, there exist $h_\epsilon \in \mathrm{Lip}_S \cap \mathbb{C}^1$, such that

$$\mathbb{E}[h(X) - h_\epsilon(X)] < \epsilon\,, \quad \mathbb{E}[h(Y) - h_\epsilon(Y)] < \epsilon.$$

After taking the supremum, we know

$$\sup_{h \in \text{Lip}_S} \{\mathbb{E}[h(X) - h(Y)]\} = \sup_{h \in \text{Lip}_S \cap C^1(\mathbb{R}^d)} \{\mathbb{E}[h(X) - h(Y)].$$

□

To compute the distance of $Y$ and $Y^{(\alpha)}$, the key is to compare the difference of the two corresponding Stein's generator. The following lemma enables us to attain a Stein generator for $Y^{(\alpha)}$.

**Lemma C.5.** *Let the random variable $X$ and $X'$ shares the same distribution. Consider the generator*

$$\mathcal{L}f(x) := \mathbb{E}[f(X') - f(X)|X = x].$$

*Then*

$$\mathbb{E}[\mathcal{L}f(X)] = 0$$

*for all $f$ integrable.*

*Proof.* Since $X$ and $X'$ shares the same distribution, we know

$$\mathbb{E}[\mathcal{L}f(X)] = \mathbb{E}_{x \sim X}[\mathbb{E}[f(X') - f(X)|X = x]] = \mathbb{E}[f(X')] - \mathbb{E}[f(X)] = 0$$

□

Having established the necessary preliminaries, we now proceed to a key lemma that serves as the foundation for proving our main results. Later on, we will demonstrate how Theorems 3.5, 3.7, and 3.11 can be derived as specific instances of this general lemma.

### C.4. A General Purpose Lemma under iid Noise

In this section, we present the general lemma that serves as the foundation for proving our main results, Theorems 3.5, 3.7, and 3.11. In order to propose this general purpose lemma, we will work with the constant-stepsize SA recursion

$$X_{k+1}^{(\alpha)} = X_k^{(\alpha)} + \alpha\big(F(X_k^{(\alpha)}) + \xi_k\big), \tag{39}$$

We first introduce the following assumption on the drift function $F$.

**Assumption C.6** (Drift Regularity). The mapping $F : \mathbb{R}^d \to \mathbb{R}^d$ is continuously differentiable with a unique root $x^\star \in \mathbb{R}^d$ satisfying $F(x^\star) = 0$. Additionally:

1. $F$ is globally Lipschitz: $\|F(x) - F(y)\|_2 \leq L\|x - y\|_2$ for all $x, y \in \mathbb{R}^d$.

2. $F \in C^3(\mathbb{R}^d; \mathbb{R}^d)$ with bounded derivatives: $\sup_{x \in \mathbb{R}^d} \max_{i,j} \left|\frac{\partial^2 f^{(i)}}{\partial x_j \partial x_k}(x)\right| \leq M$ for some $M < \infty$.

3. Let $J := DF(x^\star)$ be the Jacobian of $F$ at $x^\star$, and recall $\Sigma$ is the covariance matrix defined in Assumption 3.1. We assume that $J^\star$ is Hurwitz and the Lyapunov equation

$$J^\star \Sigma_Y + \Sigma_Y(J^\star)^\top = -\Sigma \tag{40}$$

admits a unique symmetric positive definite solution $\Sigma_Y \in \mathbb{R}^{d \times d}$.

**Lemma C.7** (i.i.d Gaussian approximation). *Consider the constant-stepsize SA recursion*

$$X_{k+1}^{(\alpha)} = X_k^{(\alpha)} + \alpha\big(F(X_k^{(\alpha)}) + \xi_k\big), \tag{41}$$

*where $X_0^{(\alpha)} \in \mathbb{R}^d$ and $\alpha > 0$ is fixed. We require the following:*

1. *The noise sequence $\{\xi_k\}_{k \geq 0}$ is i.i.d. with random vectors in $\mathbb{R}^d$ satisfying Assumption 3.1;*

2. *The drift function $F$ satisfies Assumption C.6.*

3. *The Markov chain $\{X_k^{(\alpha)}\}_{k \geq 0}$ under the above two conditions admits a unique stationary distribution $\pi_\alpha^X$. Let $X^{(\alpha)} \sim \pi_\alpha^X$ and define $Y^{(\alpha)} := (X^{(\alpha)} - x^\star)/\sqrt{\alpha}$, then we require $\mathbb{E}[\|Y^{(\alpha)}\|_2^3] < \infty$.*

*Define the target Gaussian distribution $Y \sim \mathcal{N}(0, \Sigma_Y)$. Then there exists $\alpha_0 \in (0, 1]$ and constants $U_1 \in (0, \infty)$ (depending on $d, L, M, A$, noise parameters, and spectral properties of $J^\star$) such that for all $\alpha \in (0, \alpha_0)$,*

$$d_W\big(\mathcal{L}(Y^{(\alpha)}), \mathcal{L}(Y)\big) \leq U \sqrt{\alpha} \log(1/\alpha), \tag{42}$$

*where $U := M\mathbb{E}[\|Y^{(\alpha)}\|_2^2]g_{1,Y} + \frac{L^2}{2}g_{2,Y}\mathbb{E}[\|Y^{(\alpha)}\|_2^2] + g_{3,Y}\big(1 + L^3\mathbb{E}[\|Y^{(\alpha)}\|_2^3] + \mathbb{E}[\|\xi\|_2^3]\big).$*

We now proceed to prove Lemma C.7 using Stein's method. The proof consists of two main steps: first, we establish the generator coupling between the discrete-time Markov chain $Y^{(\alpha)}$ and the continuous-time Ornstein-Uhlenbeck (OU) process $Y$. Next, we bound the difference between the two generators using Taylor expansion and moment bounds.

### C.4.1. GENERATOR COUPLING

Generator coupling is a powerful technique in Stein's method that allows us to compare two stochastic processes by analyzing their generators. In our case, we will compare the generator of the discrete-time Markov chain $Y^{(\alpha)}$ with that of the continuous-time OU process $Y$.

Under 3.3 and the extension in (Chen et al., 2022), we have that for $\alpha$ sufficiently small the chain admits a stationary law. Therefore, given a test function $g : \mathbb{R}^d \to \mathbb{R}$, we define the discrete Stein operator associated with the stationary rescaled chain by

$$\mathcal{L}^{(\alpha)}g(y, z) := \frac{1}{\alpha}\mathbb{E}\Big[g\big(Y_1^{(\alpha)}\big) - g\big(Y_0^{(\alpha)}\big) \Big| Y_0^{(\alpha)} = y, \Big]. \tag{43}$$

As the target distribution is the Gaussian, the generator is that of the OU process discussed in the previous subsection. That is, the generator is defined as

$$\mathcal{L}g(y) := (J^\star y)^\top \nabla g(y) + \frac{1}{2}\text{tr}\big(\Sigma \nabla^2 g(y)\big),$$

We can now obtain an upper bound on the Wasserstein distance between $Y^{(\alpha)}$ and $Y$ via Stein's method:

$$
\begin{aligned}
d_W(Y^{(\alpha)}, Y) &= \sup_{h \in \text{Lip}_1} \big\{\mathbb{E}[h(Y^{(\alpha)}) - h(Y)]\big\} \\
&= \sup_{h \in \text{Lip}_1} \big\{\mathbb{E}[\mathcal{L}g_h(Y^{(\alpha)})]\big\} \\
&\overset{(a)}{\leq} \sup_{g_h \in G(J^*, \Sigma)} \big\{\mathbb{E}[\mathcal{L}g_h(Y^{(\alpha)})]\big\} \\
&\overset{(b)}{=} \sup_{g_h \in G(J^*, \Sigma)} \big\{\mathbb{E}[\mathcal{L}g_h(Y^{(\alpha)}) - \mathcal{L}^{(\alpha)}g_h(Y^{(\alpha)})]\big\} \\
&\overset{\triangle}{=} D.
\end{aligned}
$$

where inequality (a) holds from the definition of $G(J^*, \Sigma)$ in Proposition C.3. Equality (b) holds since $\mathbb{E}[\mathcal{L}^{(\alpha)}g_h(Y^{(\alpha)})] = 0$ by Lemma C.2.

### C.4.2. BOUNDING THE GENERATOR DIFFERENCE

We will now Taylor expand $g_h(Y_{\infty+1}) - g_h(Y_\infty)$. If not stated otherwise, the norm $\|\cdot\|$ denotes the Euclidean norm for vectors and the operator norm for matrices induced by the Euclidean norm. Abstracting the higher-order remainder as $R_3(Y_{\infty+1}, Y_\infty)$, we have

$$D = \frac{1}{2}\text{tr}\big(\Sigma \nabla^2 g_h(y)\big) + \big\langle (J^\star)^\top y, \nabla g_h(y)\big\rangle$$

$$- \frac{1}{\alpha} \mathbb{E}\left[ \nabla g_h(Y_\infty)^\top (Y_{\infty+1} - Y_\infty) + \frac{1}{2} (Y_{\infty+1} - Y_\infty)^\top \nabla^2 g_h(Y_\infty) (Y_{\infty+1} - Y_\infty) + R_3(Y_{\infty+1}, Y_\infty) \,\middle|\, Y_\infty = y \right]$$

$$= W_1 + W_2 + R,$$

where

$$W_1 = \left\langle (J^\star)^\top y, \nabla g_h(y) \right\rangle - \frac{1}{\alpha} \mathbb{E}\left[ \nabla g_h(Y_\infty)^\top (Y_{\infty+1} - Y_\infty) \,\middle|\, Y_\infty = y \right],$$

$$W_2 = \frac{1}{2} \operatorname{tr}\!\left( \Sigma \nabla^2 g_h(y) \right) - \frac{1}{2\alpha} \mathbb{E}\left[ (Y_{\infty+1} - Y_\infty)^\top \nabla^2 g_h(Y_\infty) (Y_{\infty+1} - Y_\infty) \,\middle|\, Y_\infty = y \right],$$

$$R = -\frac{1}{\alpha} \mathbb{E}[R_3(Y_{\infty+1}, Y_\infty) \mid Y_\infty = y].$$

**Handling the Remainder Term** $R$. We bound the remainder term $R_3$ here before proceeding to bound $W_1$ and $W_2$. We first denote $y_0 := y$ and $y_1 := y + \alpha^{1/2}(F(x^* + \sqrt{\alpha}\, y) + \xi_0)$ a random variable with the same distribution as $Y_{\infty+1}$ conditioned on $Y_\infty = y$. Then, by Taylor's theorem with integral remainder, we have that

$$R = \frac{-1}{\alpha} \mathbb{E}_\pi[R_3]$$

$$\overset{(a)}{=} \frac{-1}{\alpha} \mathbb{E}_\pi\left[ \frac{1}{2} \int_0^1 (1-t)\, (y_1 - y_0)^\top \left( \nabla^2 g_h(y_0 + t(y_1 - y_0)) - \nabla^2 g_h(y_0) \right)(y_1 - y_0)\, dt \right]$$

$$\overset{(b)}{\leq} g_{3,Y} \frac{1}{1-\beta} \left( \frac{\alpha^{\frac{\beta}{2}}}{2} \int_0^1 (1-t)|t|^\beta \, dt \right) \mathbb{E}_\pi\left[ \left\| F(x^* + \sqrt{\alpha}\, y_0) + \xi_0 \right\|^{\beta+2} \right]$$

$$\overset{(c)}{\leq} g_{3,Y} \frac{1}{1-\beta} \frac{\alpha^{\beta/2}}{2(\beta+1)(\beta+2)} \mathbb{E}_\pi\left[ \left\| F(x^* + \sqrt{\alpha}\, y_0) + \xi_0 \right\|^{\beta+2} \right]$$

$$\overset{(d)}{\leq} g_{3,Y} \frac{1}{1-\beta} \frac{\alpha^{\beta/2}}{2(\beta+1)(\beta+2)} \mathbb{E}_\pi\left[ 2^{\beta+1} \left( 1 + \left\| F(x^* + \sqrt{\alpha}\, y_0) \right\|^3 + \|\xi_0\|^3 \right) \right]$$

$$\overset{(e)}{\leq} g_{3,Y} \frac{1}{1-\beta} \frac{\alpha^{\beta/2} 2^\beta}{(\beta+1)(\beta+2)} \left( 1 + \mathbb{E}_\pi\left[ (L\sqrt{\alpha}\, \|y_0\|)^3 \right] + \mathbb{E}_\pi\left[ \|\xi_0\|^3 \right] \right)$$

$$\overset{(f)}{\leq} g_{3,Y} \frac{1}{1-\beta} \alpha^{\beta/2} \cdot \left( 1 + L^3 C_{y,3} + C_{\xi,3} \right).$$

Equality $(a)$ follows from the integral form of the remainder in Taylor's theorem. Inequality $(b)$ uses the Hölder continuity of the Hessian of $g_h$ from Proposition C.3 , and inequality $(c)$ follows from evaluating the integral. Inequality $(d)$ uses the Minkowski inequality. Inequality $(e)$ follows from the Lipschitz continuity of $F$ and the fact that $F(x^*) = 0$. Finally, inequality $(f)$ uses the moment bounds on $y_0$ and $\xi$ from the assumptions of the lemma.

**Bounding** $W_1$ We now focus on $\mathbb{E}_{y \sim \mathrm{Law}(Y^{(\alpha)})}[W_1]$. Because the noise sequence $\{\xi_k\}_{k \geq 0}$ is i.i.d and independent of the current state $Y_k$, we have

$$\frac{1}{\alpha} \mathbb{E}\left[ \nabla g_h(Y_\infty)^\top (Y_{\infty+1} - Y_\infty) \mid Y_\infty = y \right] = \frac{1}{\sqrt{\alpha}} \left( F(x^* + \sqrt{\alpha}\, y) \right)^\top \nabla g_h(y).$$

This yields that

$$W_1 = \mathbb{E}\left[ \langle (J^\star)^\top y, \nabla g_h(y) \rangle \right] - \frac{1}{\sqrt{\alpha}} \mathbb{E}\left[ \nabla g_h(y)^\top F(x^* + \sqrt{\alpha}\, y) \right] \tag{44}$$

$$= \mathbb{E}\left[ \nabla g_h(y)^\top (J^\star)^\top y - \frac{1}{\sqrt{\alpha}} \nabla g_h(y)^\top F(x^* + \sqrt{\alpha}\, y) \right]$$

$$= \mathbb{E}\left[ \nabla g_h(y)^\top (J^\star)^\top y - \int_0^1 \nabla g_h(y)^\top DF(x^* + t\sqrt{\alpha} y)\, y \, dt \right]$$

$$= -\mathbb{E}\left[ \int_0^1 \nabla g_h(y)^\top \left( DF(x^* + t\sqrt{\alpha} y) - J^\star \right) y \, dt \right]$$

$$= -\mathbb{E}\left[\int_0^1 \int_0^t \sqrt{\alpha}\, \nabla g_h(y)^\top \big(D^2 F(x^* + u\sqrt{\alpha}y)[y]\big)\, y\, du\, dt\right].$$

By the definition of our assumptions on $F$, we know that $D^2 F$ is bounded uniformly over $\mathbb{R}^d$. Thus, we have, for all $u, t \in [0, 1]$,

$$\left|\sqrt{\alpha}\, \nabla g_h(y)^\top \big(D^2 F(x^* + u\sqrt{\alpha}y)[y]\big)\, y\right| \le \sqrt{\alpha}\, d^2 \|D^2 F\|_\infty \|\nabla g_h\|_\infty \|y\|^2.$$

Since $\mathbb{E}\|y\|^2 < \infty$, the integrand is integrable and we may apply Fubini-Tonelli theorem to interchange the expectation with the integrals. Hence

$$\mathbb{E}\left[\int_0^1 \int_0^t \sqrt{\alpha}\, \nabla g_h(y)^\top \big(D^2 F(x^* + u\sqrt{\alpha}y)[y]\big)\, y\, du\, dt\right]$$
$$= \int_0^1 \int_0^t \sqrt{\alpha}\, \mathbb{E}\big[\nabla g_h(y)^\top \big(D^2 F(x^* + u\sqrt{\alpha}y)[y]\big)\, y\big]\, du\, dt.$$

We now bound the expectation using the operator norm:

$$\left|\mathbb{E}\big[\nabla g_h(y)^\top \big(D^2 F(x^* + u\sqrt{\alpha}y)[y]\big)\, y\big]\right| \le \mathbb{E}\big[\|D^2 F(x^* + u\sqrt{\alpha}y)[y]\|\, \|y\|\, \|\nabla g_h(y)\|\big]$$
$$\le \|D^2 F\|_\infty \|\nabla g_h\|_\infty \mathbb{E}\big[\|y\|^2\big].$$

Therefore,

$$\left|\mathbb{E}\big[\langle (J^\star)^\top y, \nabla g_h(y)\rangle\big] - \frac{1}{\sqrt{\alpha}} \mathbb{E}\big[\nabla g_h(y)^\top F(x^* + \sqrt{\alpha}\, y)\big]\right| \le \int_0^1 \int_0^t \sqrt{\alpha}\, \|D^2 F\|_\infty \|\nabla g_h\|_\infty \mathbb{E}\big[\|y\|^2\big]\, du\, dt$$
$$= \sqrt{\alpha}\, \|D^2 F\|_\infty \|\nabla g_h\|_\infty \mathbb{E}\big[\|y\|^2\big] \int_0^1 \int_0^t du\, dt$$
$$= \frac{\sqrt{\alpha}}{2} \|D^2 F\|_\infty \|\nabla g_h\|_\infty \mathbb{E}\big[\|y\|^2\big]$$
$$= M\mathbb{E}[\|y\|^2]g_{1,Y}\, \sqrt{\alpha}. \tag{45}$$

**Bounding $W_2$.** We expand $W_2$ as follows:

$$W_2 = \frac{1}{2} \operatorname{tr}\big(\Sigma \nabla^2 g_h(y)\big) - \frac{1}{2\alpha} \mathbb{E}\Big[(Y_{\infty+1} - Y_\infty)^\top \nabla^2 g_h(Y_\infty)\, (Y_{\infty+1} - Y_\infty) \,\Big|\, Y_\infty = y\Big]$$
$$= \frac{1}{2} \operatorname{tr}\big(\Sigma \nabla^2 g_h(y)\big) - \frac{1}{2} \mathbb{E}\Big[\big(F(x^\star + \sqrt{\alpha}\, Y_\infty) + w_\infty\big)^\top \nabla^2 g_h(Y_\infty)\, \big(F(x^\star + \sqrt{\alpha}\, Y_\infty) + w_\infty\big) \,\Big|\, Y_\infty = y\Big]$$
$$= \frac{1}{2} \operatorname{tr}\big(\Sigma \nabla^2 g_h(y)\big) - \frac{1}{2} \mathbb{E}\Big[F(x^\star + \sqrt{\alpha}\, Y_\infty)^\top \nabla^2 g_h(Y_\infty)\, F(x^\star + \sqrt{\alpha}\, Y_\infty) \,\Big|\, Y_\infty = y\Big]$$
$$- \mathbb{E}\Big[w_\infty^\top \nabla^2 g_h(Y_\infty)\, F(x^\star + \sqrt{\alpha}\, Y_\infty) \,\Big|\, Y_\infty = y\Big] - \frac{1}{2} \mathbb{E}\Big[w_\infty^\top \nabla^2 g_h(Y_\infty)\, w_\infty \,\Big|\, Y_\infty = y\Big].$$

Hence, taking expectation under the stationary law $\pi$ of $Y^{(\alpha)}$,

$$\mathbb{E}_\pi[W_2] = \frac{1}{2} \mathbb{E}_\pi\big[\operatorname{tr}(\Sigma \nabla^2 g_h(y))\big] - \frac{1}{2} \mathbb{E}_\pi\Big[F(x^\star + \sqrt{\alpha}\, y)^\top \nabla^2 g_h(y)\, F(x^\star + \sqrt{\alpha}\, y)\Big]$$
$$- \mathbb{E}_\pi\Big[w^\top \nabla^2 g_h(y)\, F(x^\star + \sqrt{\alpha}\, y)\Big] - \frac{1}{2} \mathbb{E}_\pi\big[w^\top \nabla^2 g_h(y)\, w\big]$$
$$= -\frac{1}{2} \mathbb{E}_\pi\Big[F(x^\star + \sqrt{\alpha}\, y)^\top \nabla^2 g_h(y)\, F(x^\star + \sqrt{\alpha}\, y)\Big] \tag{46}$$
$$\le \frac{1}{2} \mathbb{E}_\pi\Big[\|\nabla^2 g_h(y)\|_{\operatorname{op}} \|F(x^\star + \sqrt{\alpha}\, y)\|^2\Big]$$
$$\le \frac{1}{2} \|\nabla^2 g_h\|_\infty \mathbb{E}_\pi\Big[\|F(x^\star + \sqrt{\alpha}\, y)\|^2\Big]$$

$$
= \frac{1}{2} \|\nabla^2 g_h\|_\infty \, \mathbb{E}_\pi \Big[ \|F(x^\star + \sqrt{\alpha}\, y) - F(x^\star)\|^2 \Big]
$$

$$
\leq \frac{1}{2} \|\nabla^2 g_h\|_\infty \, \mathbb{E}_\pi \Big[ (L\sqrt{\alpha} \, \|y\|)^2 \Big]
$$

$$
= \frac{1}{2} \|\nabla^2 g_h\|_\infty \, L^2 \, \alpha \, \mathbb{E}_\pi[\|y\|^2] \tag{47}
$$

$$
= \frac{L^2}{2} g_{2,Y} \mathbb{E}[\|y\|^2] \alpha. \tag{48}
$$

Therefore, combining (45), (48), and the bound on $R$, we have

$$
d_w(Y^{(\alpha)}, Y) \leq M \mathbb{E}[\|y\|^2] g_{1,Y} \sqrt{\alpha} + \frac{L^2}{2} g_{2,Y} \mathbb{E}[\|y\|^2] \alpha + g_{3,Y} \frac{1}{1-\beta} \alpha^{\beta/2} \cdot \left( 1 + L^3 C_{y,3} + C_{\xi,3} \right)
$$

$$
\overset{(a)}{\leq} U \sqrt{\alpha} \log(1/\alpha),
$$

where (a) holds from choosing $\beta = 1 - 1/\log(1/\alpha)$. Here, $U$ is a constant defined as

$$
U := M \mathbb{E}[\|Y^{(\alpha)}\|^2] g_{1,Y} + \frac{L^2}{2} g_{2,Y} \mathbb{E}[\|Y^{(\alpha)}\|^2] + g_{3,Y} \left( 1 + L^3 \mathbb{E}[\|Y^{(\alpha)}\|^3] + \mathbb{E}[\|\xi\|^3] \right).
$$

Now we have established the Wasserstein bound conditioning on the drift, noise, and moment bounds. To proceed, we provide an extra argument that connect Wasserstein bounds directly to the non-uniform Berry-Esseen type of bounds, presented in the second bullet points for all theorems in the main text. Then we will apply both two argument, i.e., Wasserstein bounds and tail bounds to different models.

## C.5. From Wasserstein Distance to Concentration

Building upon Wasserstein bound, we can establish non-uniform Berry-Esseen type bound as follows.

**Lemma C.8.** *Let $Y$ be a real-valued random vector in $\mathbb{R}^d$ and let $Z \sim \mathcal{N}(0, \Sigma_Z)$. If $d_W(Y, Z) \leq \delta$, where $d_W(\cdot, \cdot)$ denotes the Wasserstein-1 distance defined in (2), then for every $a > 0$, every $\rho \in [0, 1)$, and every unit vector $\zeta \in \mathbb{R}^d$ such that $\|\zeta\| = 1$, we have*

$$
\left| \mathbb{P}(\langle Y, \zeta \rangle > a) - \mathbb{P}(\langle Z, \zeta \rangle > a) \right| \; \leq \; \frac{(1-\rho)a}{\sqrt{\zeta^T \Sigma_Z \zeta}} \phi\left( \frac{\rho a}{\sqrt{\zeta^T \Sigma_Z \zeta}} \right) + \frac{d_W(Y, Z)}{(1-\rho)a} \tag{49}
$$

*where $\phi(x) = \frac{1}{\sqrt{2\pi}} e^{-x^2/2}$ is the standard normal density.*

*Proof.* Given the bounds on Wasserstein-1 distance we obtained in previous sections, we can further derive concentration inequalities using the Wasserstein-1 bound. The derivation below connects the Wasserstein-1 distance to concentration inequalities in the spirit of (Austern & Mackey, 2022; Fang & Koike, 2022; Wang & Maguluri, 2026, Inequality 5.2). Particularly, recall $\Phi(a)$ as cumulative distribution function (CDF) of standard normal distribution, $\Phi^c(a) = 1 - \Phi(a)$ as its complementary CDF, and $\phi(a) = e^{-\frac{x^2}{2}}$ as the probability density function (PDF) of standard normal distribution. Then for any $\rho \in [0, 1)$ and $a \geq 0$, we have

$$
\begin{aligned}
\mathbb{P}(\langle Y, \zeta \rangle > a) &\leq \mathbb{P}(\langle Y - Z, \zeta \rangle \geq (1-\rho)a) + \mathbb{P}(\langle Z, \zeta \rangle \geq \rho a) \\
&\leq \mathbb{P}(|\langle \zeta, Y - Z \rangle| \geq (1-\rho)a) + \mathbb{P}(\langle Z, \zeta \rangle \geq \rho a) \\
&\overset{(a)}{\leq} \frac{\|\zeta\| \mathbb{E}[\|Y - Z\|]}{(1-\rho)a} + \mathbb{P}(\langle Z, \zeta \rangle \geq \rho a) \\
&\overset{(b)}{=} \frac{\|\zeta\| d_W(Y, Z)}{(1-\rho)a} + \mathbb{P}(\langle Z, \zeta \rangle \geq \rho a) \\
&\overset{(c)}{=} \frac{d_W(Y, Z)}{(1-\rho)a} + \Phi^c\left( \frac{\rho a}{\sqrt{\zeta^T \Sigma_Z \zeta}} \right) \\
&\overset{(d)}{\leq} \frac{d_W(Y, Z)}{(1-\rho)a} + \frac{(1-\rho)a}{\sqrt{\zeta^T \Sigma_Z \zeta}} \phi\left( \frac{\rho a}{\sqrt{\zeta^T \Sigma_Z \zeta}} \right) + \Phi^c\left( \frac{a}{\sqrt{\zeta^T \Sigma_Z \zeta}} \right)
\end{aligned}
$$

Where inequality $(a)$ is from Markov inequality. Inequality $(b)$ follows from the definition of Wasserstein-1 distance in (2). Note that we can choose a coupling $(Y, Z)$ such that the $L^1$ distance between $Y$ and $Z$ is within error $\epsilon$ to the Wasserstein-1 distance $d_W(Y, Z)$. Such error $\epsilon$ can be arbitrary small since Wasserstein-1 distance is defined as infimum over all couplings. Thus, letting $\epsilon \downarrow 0$ we have the equality in $(b)$. Equality $(c)$ is from the fact $\|\zeta\| = 1$. Inequality $(d)$ follows from Taylor Expansion $\Phi^c(\rho b) \leq \Phi^c(b) + (b - \rho b) \sup_{\tilde{x} \in [\rho b, b]} \phi(\tilde{x})$. Meanwhile, similar argument shows the lower bound

$$\mathbb{P}(\langle Z, \zeta \rangle > a) \leq \mathbb{P}(\langle Y, \zeta \rangle > a) + \mathbb{P}(|\langle Y, \zeta \rangle - \langle Z, \zeta \rangle| \geq (1 - \rho)a) + \mathbb{P}(a \leq \langle Z, \zeta \rangle \leq (2 - \rho)a)$$

$$\leq \mathbb{P}(\langle Y, \zeta \rangle > a) + [(1 - \rho)a]^{-1} d_W(Y, Z) + \frac{(1 - \rho)a}{\sqrt{\zeta^T \Sigma_Z \zeta}} \phi(\frac{\rho a}{\sqrt{\zeta^T \Sigma_Z \zeta}})$$

In summary, we have for any $\rho \in [0, 1)$ and $a \geq 0$,

$$|\mathbb{P}(\langle Y, \zeta \rangle > a) - \mathbb{P}(\langle Z, \zeta \rangle > a)| \leq \frac{(1 - \rho)a}{\sqrt{\zeta^T \Sigma_Z \zeta}} \phi(\frac{\rho a}{\sqrt{\zeta^T \Sigma_Z \zeta}}) + \frac{d_W(Y, Z)}{(1 - \rho)a}. \tag{50}$$

$\square$

**Corollary C.9.** *Suppose for random vector $Y$ and standard normal vector $Z \sim \mathcal{N}(0, \Sigma_Z)$, we have $d_W(Y^{(\alpha)}, Z) \leq \delta \alpha^{1/2} \log^{1/2}(1/\alpha)$ for some constant $\delta > 0$. Then for every $a > 0$, let $\zeta \in \mathbb{R}^d$ be any unit vector, i.e., $\|\zeta\| = 1$. We have*

$$\left| \mathbb{P}(\langle Y^{(\alpha)}, \zeta \rangle > a) - \mathbb{P}(\langle Z, \zeta \rangle > a) \right| \leq U_{tail} \frac{\alpha^{\frac{1}{4}} \log^{\frac{1}{2}} \frac{1}{\alpha}}{a}, \tag{51}$$

*with constant $U_{tail} = (8\sqrt{\|\Sigma_Z\|_{op}} + 1)\delta^{1/2}$.*

*Proof.* From Lemma C.8 and the condition on $d_W(Y^{(\alpha)}, Z)$, we have

$$|\mathbb{P}(\langle Y^{(\alpha)}, \zeta \rangle > a) - \mathbb{P}(\langle Z, \zeta \rangle > a)| \leq \frac{(1 - \rho)a}{\sqrt{\zeta^T \Sigma_Z \zeta}} \phi(\frac{\rho a}{\sqrt{\zeta^T \Sigma_Z \zeta}}) + \frac{d_W(Y, Z)}{(1 - \rho)a}$$

for any $\rho \in [0, 1)$. Choose $\rho = 1 - \sqrt{\delta \alpha^{1/2} \log(1/\alpha)}$. For $\alpha$ sufficiently small such that $\delta \alpha^{1/2} \log(1/\alpha) < 1$, we have $\rho \in (0, 1)$. Then:

$$|\mathbb{P}(\langle Y^{(\alpha)}, \zeta \rangle > a) - \mathbb{P}(\langle Z, \zeta \rangle > a)| \leq \sqrt{\delta \alpha^{1/2} \log(1/\alpha)} \cdot \frac{a}{\sqrt{\zeta^T \Sigma_Z \zeta}} \phi\big((1 - \sqrt{\delta \alpha^{1/2} \log(1/\alpha)}) \frac{a}{\sqrt{\zeta^T \Sigma_Z \zeta}}\big)$$

$$+ \frac{\delta \alpha^{1/2} \log(1/\alpha)}{\sqrt{\delta \alpha^{1/2} \log(1/\alpha)} \cdot a}$$

$$= \frac{a}{\sqrt{\zeta^T \Sigma_Z \zeta}} \sqrt{\delta \alpha^{1/2} \log(1/\alpha)} \, \phi\big((1 - \sqrt{\delta \alpha^{1/2} \log(1/\alpha)}) \frac{a}{\sqrt{\zeta^T \Sigma_Z \zeta}}\big)$$

$$+ \frac{\sqrt{\delta \alpha^{1/2} \log(1/\alpha)}}{a}$$

For $\alpha$ sufficiently small, $(1 - \sqrt{\delta \alpha^{1/2} \log(1/\alpha)}) \geq 1/2$, so:

$$\phi\big((1 - \sqrt{\delta \alpha^{1/2} \log(1/\alpha)}) \frac{a}{\sqrt{\zeta^T \Sigma_Z \zeta}}\big) \leq \phi(\frac{a}{2\sqrt{\zeta^T \Sigma_Z \zeta}}) \leq \exp(-\frac{a^2}{8\zeta^T \Sigma_Z \zeta})$$

Therefore:

$$|\mathbb{P}(\langle Y^{(\alpha)}, \zeta \rangle > a) - \mathbb{P}(\langle Z, \zeta \rangle > a)| \leq \sqrt{\delta \alpha^{1/2} \log(1/\alpha)} \left( \frac{a}{\sqrt{\zeta^T \Sigma_Z \zeta}} \exp(-\frac{a^2}{8\zeta^T \Sigma_Z \zeta}) + \frac{1}{a} \right)$$

$$\overset{(a)}{\leq} (8\sqrt{\zeta^T \Sigma_Z \zeta} + 1)\delta^{1/2} \cdot \frac{\alpha^{1/4} \log^{1/2}(1/\alpha)}{a}$$

$$\overset{(b)}{\leq} \underbrace{(8\sqrt{\|\Sigma_Z\|_{op}} + 1)\delta^{1/2}}_{:=U_{tail}} \cdot \frac{\alpha^{1/4}\log^{1/2}(1/\alpha)}{a},$$

where the inequality $(a)$ holds for all $a > 0$ and sufficiently small $\alpha > 0$, since $be^{-b^2/8} \leq 8/b$ for all $b > 0$. Inequality $(b)$ follows from the fact that $\zeta^T \Sigma_Z \zeta \leq \|\Sigma_Z\|_{op}$ for any unit vector $\zeta$. $\qquad\square$

We are now ready to apply the general purpose Lemma C.7 and tail bounds, especially Corollary C.9 to specific algorithms in the following sections.

### C.6. Proof of Theorem 3.5: Constant-stepsize SGD

Consider constant-stepsize SGD for minimizing a differentiable objective $f : \mathbb{R}^d \to \mathbb{R}$:

$$X_{k+1} = X_k - \alpha\big(\nabla f(X_k) + \xi_k\big), \qquad k \geq 0, \tag{52}$$

where $\{\xi_k\}_{k \geq 0}$ are i.i.d. with $\mathbb{E}[\xi_k] = 0$ and $\mathbb{E}[\xi_k \xi_k^\top] = \Sigma$ (and satisfy Assumption 3.1). Define

$$F(x) := -\nabla f(x), \tag{53}$$

so that (88) can be written in the form,

$$X_{k+1} = X_k + \alpha\big(F(X_k) + \xi_k\big).$$

In the following, we will apply the general purpose Lemma C.7 to establish Theorem 3.5 by verifying Assumptions C.6, 3.1, and the existence of third moment for stationary distribution of iterates $\{X_k\}_{k \geq 0}$.

(i). Drfit Assumptions C.6 By Assumption 3.2, $f$ is $\sigma$-strongly convex and $L$-smooth, so $f$ has a unique minimizer $x^\star$ characterized by $\nabla f(x^\star) = 0$, hence $F(x^\star) = 0$ and this root is unique. Moreover, $L$-smoothness implies $\nabla f$ is globally $L$-Lipschitz, so for all $x, y$,

$$\|F(x) - F(y)\|_2 = \|\nabla f(y) - \nabla f(x)\|_2 \leq L\|x - y\|_2.$$

Meanwhile, since $F = -\nabla f$, we have $DF(x) = -\nabla^2 f(x)$ and $D^2 F(x) = -\nabla^3 f(x)$. Thus Assumption C.6(2) holds by Assumption 3.4 which gives $\sup_x \|\nabla^3 f(x)\|_{op} < \infty$. Finally, by $\sigma$-strong convexity, $\nabla^2 f(x^\star) \succeq \sigma I$, hence the eigenvalues of $J^\star$ satisfy $\Re(\lambda(J^\star)) \leq -\sigma < 0$, i.e., $J^\star$ is Hurwitz. Consequently, the continuous Lyapunov equation (40) admits a unique symmetric solution $\Sigma_Y$, and if $\Sigma \succ 0$ then $\Sigma_Y \succ 0$.

(ii). Noise Assumption 3.1 Since $\{\xi_k\}$ are i.i.d. with $\mathbb{E}[\xi_k] = 0$ and covariance $\Sigma$, it follows that $\{\xi_k\}$ are i.i.d. with $\mathbb{E}[\xi_k] = 0$ and $\mathbb{E}[\xi_k \xi_k^\top] = \Sigma$, and inherit any regularity conditions required by Assumption 3.1.

(iii). Stationarity We note that the existence of stationary distribution of $X^{(\alpha)}$ is already established in Lemma 3.3. We focus on proving the boundedness of third moment of the stationary distribution. In the following, we first establish a contractive drift condition for deterministic vector $x \in \mathbb{R}^d$ when $\alpha$ is sufficiently small.

$$\begin{aligned}
\|x - \alpha\nabla f(x)\|_2^2 &= \|x\|_2^2 - 2\alpha\langle x, \nabla f(x)\rangle + \alpha^2\|\nabla f(x)\|_2^2 \\
&\overset{(a)}{\leq} \|x\|_2^2 - 2\alpha\sigma\|x\|_2^2 + \alpha^2 L^2\|x\|_2^2 \\
&= (1 - 2\alpha\sigma + \alpha^2 L^2)\|x\|_2^2 \\
&\leq (1 - \alpha\sigma)\|x\|_2^2, \ \forall\alpha \in \left(0, \frac{\sigma}{L^2}\right), \forall x \in \mathbb{R}^d,
\end{aligned}$$

where (a) follows from the $\sigma$-strong convexity and $L$-smoothness of $f$. Note that we achieve the last inequality by choosing $\alpha$ such that $1 - 2\alpha\sigma + \alpha^2 L^2 \leq 1 - \alpha\sigma$, i.e., $\alpha \leq \frac{\sigma}{L^2}$. Thus, we can establish the following drift condition for $X_k$ when $\alpha$ is sufficiently small.

$$\begin{aligned}
\mathbb{E}[\|X_{k+1}\|_2^3]^{1/3} &\overset{(a)}{\leq} \mathbb{E}[\|X_k - \alpha\nabla f(X_k)\|_2^3]^{1/3} + \alpha\mathbb{E}[\|\xi_k\|_2^3]^{1/3} \\
&\leq \mathbb{E}[(1 - \alpha\sigma)^{3/2}\|X_k\|_2^3]^{1/3} + \alpha\mathbb{E}[\|\xi_k\|_2^3]^{1/3} \\
&\leq (1 - \alpha\sigma)^{1/2}\mathbb{E}[\|X_k\|_2^3]^{1/3} + \alpha\mathbb{E}[\|\xi_k\|_2^3]^{1/3}.
\end{aligned}$$

Inequality (a) follows from Minkowski inequality. We note that from Assumption 3.10, $\mathbb{E}[\|\xi_k\|_2^3]$ is bounded. Thus, by letting step $k$ tending to $\infty$, we have $X_\infty \overset{d.}{=} X_{\infty+1} \sim \pi_\alpha^X$ and thus rearranging the above inequality yields

$$\mathbb{E}[\|X^{(\alpha)}\|_2^3]^{1/3} \leq \frac{\alpha\mathbb{E}[\|\xi_k\|_2^3]^{1/3}}{1-(1-\alpha\sigma)^{1/2}} \quad \text{for } \alpha \in \left(0, \frac{\sigma}{L^2}\right)$$

From triangle inequality, we know that

$$\begin{aligned}
\mathbb{E}[\|Y^{(\alpha)}\|_2^3]^{1/3} &= \frac{\mathbb{E}[\|X^{(\alpha)} - x^\star\|_2^3]^{1/3}}{\sqrt{\alpha}} \\
&\leq \frac{\mathbb{E}[\|X^{(\alpha)}\|_2^3]^{1/3} + \|x^\star\|_2}{\sqrt{\alpha}} \\
&\leq \underbrace{\frac{\alpha\mathbb{E}[\|\xi_k\|_2^3]^{1/3}}{\sqrt{\alpha}(1-(1-\alpha\sigma)^{1/2})} + \frac{\|x^\star\|_2}{\sqrt{\alpha}}}_{:=A_{SGD}} \quad \text{for } \alpha \in \left(0, \frac{\sigma}{L^2}\right)
\end{aligned}$$

which is bounded for $\alpha \in (0, \alpha_0)$ where $\alpha_0 := \min\{1, \frac{\sigma}{L^2}\}$, with constant $A = A_{SGD}$ in condition Assumption C.6.

Thus all conditions are verified for Lemma C.7, so the result follows by a direct application of the Lemma. We directly present the final result here for completeness. Recall the definition of $Y^{(\alpha)}, Y$ in Theorem 3.5, and Assumptions 3.2, 3.4, and 3.10. Then we have the following Wasserstein-1 bound.

$$d_W(Y^{(\alpha)}, Y) \leq U_1\alpha^{1/2}\log^{1/2}(1/\alpha),$$
$$\text{where } U_1 := MA_{SGD}^2 g_{1,Y} + \frac{L^2}{2}g_{2,Y}A_{SGD}^2 + g_{3,Y}\left(1 + L^3 A_{SGD}^3 + \mathbb{E}[\|\xi\|_2^3]\right), \tag{54}$$

recalling $A_{SGD}$ is defined in the proof above and $g_{1,Y}, g_{2,Y}, g_{3,Y}$ are defined in Proposition C.3. Thus, using Corollary C.9, we have for every $a > 0$, and every unit vector $\zeta \in \mathbb{R}^d$ such that $\|\zeta\| = 1$,

$$\left|\mathbb{P}(\langle Y^{(\alpha)}, \zeta\rangle > a) - \mathbb{P}(\langle Y, \zeta\rangle > a)\right| \leq U_1'\frac{\alpha^{1/4}\log^{1/2}(1/\alpha)}{a},$$
$$\text{where } U_1' := (8\sqrt{\|\Sigma_Y\|_{op}} + 1)U_1^{1/2}. \tag{55}$$

### C.7. Proof of Theorem 3.7: Linear SA

Recall the linear SA recursion
$$X_{k+1} = X_k + \alpha\left(B(X_k - x^\star) + \xi_k\right), \qquad k \geq 0, \tag{56}$$
where $B \in \mathbb{R}^{d\times d}$ is fixed, $x^\star \in \mathbb{R}^d$ is the unique root such that $Bx = b$. And $\{\xi_k\}_{k\geq 0}$ are i.i.d. with $\mathbb{E}[\xi_k] = 0$ and $\mathbb{E}[\xi_k\xi_k^\top] = \Sigma$ (and satisfy Assumption 3.1). Define the drift

$$F(x) := B(x - x^\star). \tag{57}$$

Then (89) is exactly of the form:
$$X_{k+1} = X_k + \alpha\left(F(X_k) + \xi_k\right).$$

In the following, we will apply the general purpose Lemma C.7 to establish Theorem 3.7 by verifying Assumptions C.6, 3.1, and the existence of third moment for stationary distribution of iterates $\{X_k\}_{k\geq 0}$.

(i). Drift regularity We have $F(x^\star) = 0$, and since $B$ is non-singular then this root is unique since $F(x) = 0 \iff B(x - x^\star) = 0 \iff x = x^\star$. Moreover, $F$ is globally Lipschitz with constant $L = \|B\|_{\mathrm{op}}$:

$$\|F(x) - F(y)\|_2 = \|B(x-y)\|_2 \leq \|B\|_{\mathrm{op}}\|x-y\|_2.$$

Moreover, $F$ is linear, hence $F \in C^\infty$ and all third derivatives are identically zero, so Assumption C.6 holds. Finally, for the linear drift $F(x) = J(x - x^\star)$, we have $J^\star = DF(x^\star) = J$. If $J$ is Hurwitz, then the continuous Lyapunov equation

$$J\Sigma_Y + \Sigma_Y J^\top = -\Sigma$$

admits a unique symmetric solution $\Sigma_Y$, and if $\Sigma \succ 0$ then $\Sigma_Y \succ 0$.

(ii) Noise regularity. Exactly as in the proof of the previous Theorem, this assumption holds by 3.1. So, the noise sequence is i.i.d., mean 0, covariance $\Sigma$, and has bounded third moments.

(iii) Stationarity

We note that the existence of a stationary distribution for the linear SA recursion is standard under the Hurwitz assumption on $B$ (together with $\alpha$ sufficiently small) (Chen et al., 2022, Theorem 2.3); we focus on proving boundedness of the third moment of the stationary distribution.

Let $U_k := X_k - x^\star$. Then the recursion becomes

$$U_{k+1} \;=\; (I + \alpha B)U_k \;+\; \alpha\xi_k.$$

Since $B$ is Hurwitz, there exists a symmetric positive definite matrix $P \succ 0$ and a constant $m > 0$ (Horn & Johnson, 2013), such that

$$B^\top P + PB \;\preceq\; -2mP. \tag{58}$$

Define the $P$-norm $\|u\|_P := (u^\top Pu)^{1/2}$. We first establish a contractive drift condition in $\|\cdot\|_P$ for deterministic $u \in \mathbb{R}^d$ when $\alpha$ is sufficiently small:

$$
\begin{aligned}
\|(I + \alpha B)u\|_P^2 &= u^\top (I + \alpha B)^\top P(I + \alpha B)u \\
&= u^\top Pu \;+\; \alpha\, u^\top(B^\top P + PB)u \;+\; \alpha^2\, u^\top B^\top PB\, u \\
&\overset{(a)}{\leq} \|u\|_P^2 \;-\; 2m\alpha\|u\|_P^2 \;+\; \alpha^2\|P^{1/2}Bu\|_2^2 \\
&\leq \left(1 - 2m\alpha + \alpha^2\|P^{1/2}BP^{-1/2}\|_{\mathrm{op}}^2\right)\|u\|_P^2 \\
&\leq (1 - m\alpha)\,\|u\|_P^2, \qquad \forall \alpha \in \left(0, \frac{m}{\|P^{1/2}BP^{-1/2}\|_{\mathrm{op}}^2}\right), \; \forall u \in \mathbb{R}^d,
\end{aligned}
$$

where (a) uses (58) and $\|P^{1/2}Bu\|_2 \leq \|P^{1/2}BP^{-1/2}\|_{\mathrm{op}}\|u\|_P$. The last inequality follows by choosing $\alpha$ such that $1 - 2m\alpha + \alpha^2\|P^{1/2}BP^{-1/2}\|_{\mathrm{op}}^2 \leq 1 - m\alpha$, i.e., $\alpha \leq \frac{m}{\|P^{1/2}BP^{-1/2}\|_{\mathrm{op}}^2}$.

Thus, we can establish the following drift condition for $(U_k)$ in $L^3$ when $\alpha$ is sufficiently small:

$$
\begin{aligned}
\mathbb{E}\big[\|U_{k+1}\|_P^3\big]^{1/3} &= \mathbb{E}\big[\|(I + \alpha B)U_k + \alpha\xi_k\|_P^3\big]^{1/3} \\
&\overset{(a)}{\leq} \mathbb{E}\big[\|(I + \alpha B)U_k\|_P^3\big]^{1/3} \;+\; \alpha\, \mathbb{E}\big[\|\xi_k\|_P^3\big]^{1/3} \\
&\leq \mathbb{E}\big[(1 - m\alpha)^{3/2}\|U_k\|_P^3\big]^{1/3} \;+\; \alpha\, \mathbb{E}\big[\|\xi_k\|_P^3\big]^{1/3} \\
&\leq (1 - m\alpha)^{1/2}\, \mathbb{E}\big[\|U_k\|_P^3\big]^{1/3} \;+\; \alpha\, \mathbb{E}\big[\|\xi_k\|_P^3\big]^{1/3},
\end{aligned}
$$

where (a) follows from Minkowski inequality.

Assume $\sup_k \mathbb{E}[\|\xi_k\|_2^3] < \infty$ (hence $\sup_k \mathbb{E}[\|\xi_k\|_P^3] < \infty$ since $\|\cdot\|_P \leq \lambda_{\max}(P)^{1/2}\|\cdot\|_2$). Letting $k \to \infty$, we have $X_\infty \overset{d.}{=} X_{\infty+1} \sim \pi_\alpha^X$, and thus $U_\infty \overset{d}{=} U_{\infty+1}$ and therefore rearranging yields

$$
\mathbb{E}\big[\|X^{(\alpha)}\|_P^3\big]^{1/3} \;\leq\; \|x^\star\|_P + \frac{\alpha\, \mathbb{E}\big[\|\xi_k\|_P^3\big]^{1/3}}{1 - (1 - m\alpha)^{1/2}} \qquad \text{for } \alpha \in \left(0, \frac{m}{\|P^{1/2}BP^{-1/2}\|_{\mathrm{op}}^2}\right). \tag{59}
$$

Finally, define $Y^{(\alpha)} := \frac{X^{(\alpha)} - x^\star}{\sqrt{\alpha}} = \frac{U^{(\alpha)}}{\sqrt{\alpha}}$. Using $\|u\|_2 \leq \lambda_{\min}(P)^{-1/2}\|u\|_P$, we obtain

$$
\begin{aligned}
\mathbb{E}\big[\|Y^{(\alpha)}\|_2^3\big]^{1/3} = \frac{1}{\sqrt{\alpha}}\, \mathbb{E}\big[\|U^{(\alpha)}\|_2^3\big]^{1/3} &\leq \frac{1}{\sqrt{\alpha}}\, \lambda_{\min}(P)^{-1/2}\, \mathbb{E}\big[\|U^{(\alpha)}\|_P^3\big]^{1/3} \\
&\leq \underbrace{\frac{\sqrt{\alpha}\, \lambda_{\min}(P)^{-1/2}\, \mathbb{E}\big[\|\xi_k\|_P^3\big]^{1/3}}{1 - (1 - m\alpha)^{1/2}}}_{:=A_{\mathrm{LSA}}(\alpha)} \qquad \text{for } \alpha \in \left(0, \frac{m}{\|P^{1/2}BP^{-1/2}\|_{\mathrm{op}}^2}\right),
\end{aligned}
$$

where we used (59) in the last line. Equivalently, since $\mathbb{E}[\|\xi_k\|_P^3]^{1/3} \leq \lambda_{\max}(P)^{1/2}\mathbb{E}[\|\xi_k\|_2^3]^{1/3}$,

$$A_{\text{LSA}}(\alpha) \leq \frac{\sqrt{\alpha}\,\sqrt{\kappa(P)}\,\mathbb{E}[\|\xi_k\|_2^3]^{1/3}}{1 - (1 - m\alpha)^{1/2}}, \qquad \kappa(P) := \frac{\lambda_{\max}(P)}{\lambda_{\min}(P)}.$$

Hence $\sup_{\alpha \in (0,\alpha_0)} \mathbb{E}[\|Y^{(\alpha)}\|_2^3] < \infty$ for any fixed $\alpha_0 \leq \min\left\{1, \frac{m}{\|P^{1/2}BP^{-1/2}\|_{\text{op}}^2}\right\}$, with an explicit bound given by $A_{\text{LSA}}(\alpha)$, or the displayed upper bound involving $\kappa(P)$.

Since we justify the conditions (i)–(iii) of Lemma C.7, we can apply the Lemma to attain 3.7 as desired. We present the final result here for completeness. Recall the definition of $Y^{(\alpha)}, Y$ in Theorem 3.7, and Assumptions 3.6, we have

$$d_W(Y^{(\alpha)}, Y) \leq U_2\alpha^{1/2}\log(1/\alpha)$$

$$\text{where } U_2 := \frac{\|B\|_{\text{op}}^2}{2}g_{2,Y}A_{LSA}^2 + g_{3,Y}\left(1 + \|B\|_{\text{op}}^3 A_{LSA}^3 + \mathbb{E}[\|\xi\|^3]\right), \tag{60}$$

where constants $L_{LSA}$ is defined in above formula, and $g_{2,Y}, g_{3,Y}$ are defined in Lemma C.7. Moreover, recalling the tail bound connection from Wasserstein bound in Corollary C.9, we further achieve the following tail bounds.

$$\left|\mathbb{P}(\langle Y^{(\alpha)}, \zeta\rangle > a) - \mathbb{P}(\langle Y, \zeta\rangle > a)\right| \leq U_2'\frac{\alpha^{1/4}\log^{1/2}(1/\alpha)}{a},$$

$$\text{where } U_2' := (8\sqrt{\|\Sigma_Y\|_{op}} + 1)U_2^{1/2}. \tag{61}$$

### C.8. Proof of Theorem 3.11: Contractive SA

Consider the SA recursion driven by a contractive operator $\mathcal{T} : \mathbb{R}^d \to \mathbb{R}^d$:

$$X_{k+1} = X_k + \alpha\big(\mathcal{T}(X_k) - X_k + \xi_k\big), \qquad k \geq 0, \tag{62}$$

where $\{\xi_k\}_{k \geq 0}$ are i.i.d. with $\mathbb{E}[\xi_k] = 0$ and $\mathbb{E}[\xi_k\xi_k^\top] = \Sigma$ (and satisfy Assumption 3.1). Define the drift

$$F(x) := \mathcal{T}(x) - x. \tag{63}$$

Then (91) is exactly:

$$X_{k+1} = X_k + \alpha\big(F(X_k) + \xi_k\big).$$

Similarly, we will apply the general purpose Lemma C.7 to establish Theorem 3.11 by verifying Assumptions C.6, and other required conditions.

(i) Drift regularity. By Banach's fixed-point theorem, 3.8 implies $\mathcal{T}$ admits a unique fixed point $x^\star \in \mathbb{R}^d$ such that $\mathcal{T}(x^\star) = x^\star$. Hence $F(x^\star) = 0$ and this root is unique. Moreover, recall $F(x) := \mathcal{T}(x) - x$ and the weighted norm

$$\|x\|_\mu := \Big(\sum_{i=1}^d \mu_i x_i^2\Big)^{1/2} = \|D_\mu^{1/2}x\|_2, \qquad D_\mu := \text{diag}(\mu_1, \ldots, \mu_d),$$

under which $\mathcal{T}$ is a $\gamma$-contraction:

$$\|\mathcal{T}(x_1) - \mathcal{T}(x_2)\|_\mu \leq \gamma\|x_1 - x_2\|_\mu.$$

Then for all $x, y \in \mathbb{R}^d$,

$$\begin{aligned}\|F(x) - F(y)\|_\mu &= \|(\mathcal{T}(x) - \mathcal{T}(y)) - (x - y)\|_\mu \\ &\leq \|\mathcal{T}(x) - \mathcal{T}(y)\|_\mu + \|x - y\|_\mu \\ &\leq (\gamma + 1)\,\|x - y\|_\mu.\end{aligned}$$

Moreover, since $\|\cdot\|_\mu$ is induced by a diagonal matrix, it is equivalent to the Euclidean norm:

$$\sqrt{\mu_{\min}}\|x\|_2 \leq \|x\|_\mu \leq \sqrt{\mu_{\max}}\|x\|_2, \qquad \mu_{\min} := \min_i \mu_i, \; \mu_{\max} := \max_i \mu_i.$$

Therefore,

$$\|F(x) - F(y)\|_2 \leq \frac{1}{\sqrt{\mu_{\min}}}\|F(x) - F(y)\|_\mu \leq (1+\gamma)\sqrt{\frac{\mu_{\max}}{\mu_{\min}}}\,\|x - y\|_2.$$

Meanwhile, since $F(x) = \mathcal{T}(x) - x$, and from Assumption 3.10, we have that $F \in C^3$ and that $D^2F$ is bounded. Finally, according to Lemma 3.9, we have that all eigenvalues of $J^\star = DF(x^\star) = D\mathcal{T}(x^\star) - I$ have strictly negative real parts, i.e., $J^\star$ is Hurwitz. Consequently, the continuous Lyapunov equation (40) admits a unique symmetric solution $\Sigma_Y$.

(ii) Noise regularity (Assumption 3.1). This holds by assumption on $\{\xi_k\}$ (i.i.d., mean 0, covariance $\Sigma$, and any additional required moments).

(iii) Stationarity. We note that the existence of a stationary distribution of $X^{(\alpha)}$ is already established in (Chen et al., 2022, Theorem 2.6). We focus on proving boundedness of the third moment of the stationary distribution.

Recall $x^\star$ is the unique fixed point of $\mathcal{T}$, i.e., $\mathcal{T}(x^\star) = x^\star$, and define the centered iterate $U_k := X_k - x^\star$. Consider the contractive SA recursion

$$X_{k+1} = X_k + \alpha\big(\mathcal{T}(X_k) - X_k + \xi_k\big), \qquad \text{equivalently} \qquad U_{k+1} = (1-\alpha)U_k + \alpha\big(\mathcal{T}(X_k) - \mathcal{T}(x^\star)\big) + \alpha\xi_k. \quad (64)$$

By the assumed contraction,

$$\|\mathcal{T}(x_1) - \mathcal{T}(x_2)\|_\mu \leq \gamma\|x_1 - x_2\|_\mu, \qquad \forall x_1, x_2 \in \mathbb{R}^d, \quad (65)$$

for some $\gamma \in (0,1)$. We first establish a contractive drift condition for a deterministic vector $u \in \mathbb{R}^d$. Using (65) with $x_2 = x^\star$ and $\mathcal{T}(x^\star) = x^\star$, we obtain

$$\|\mathcal{T}(x^\star + u) - x^\star\|_\mu \leq \gamma\|u\|_\mu.$$

Therefore, for any $\alpha \in (0,1)$,

$$\begin{aligned}
\big\|(1-\alpha)u + \alpha(\mathcal{T}(x^\star + u) - x^\star)\big\|_\mu &\leq (1-\alpha)\|u\|_\mu + \alpha\|\mathcal{T}(x^\star + u) - x^\star\|_\mu \\
&\leq (1-\alpha)\|u\|_\mu + \alpha\gamma\|u\|_\mu \\
&= \big(1 - \alpha(1-\gamma)\big)\|u\|_\mu.
\end{aligned}$$

Plugging $u = U_k$ and applying Minkowski inequality yields the $L^3$ drift:

$$\begin{aligned}
\mathbb{E}\big[\|U_{k+1}\|_\mu^3\big]^{1/3} &\overset{(a)}{\leq} \mathbb{E}\Big[\big\|(1-\alpha)U_k + \alpha(\mathcal{T}(X_k) - \mathcal{T}(x^\star))\big\|_\mu^3\Big]^{1/3} + \alpha\,\mathbb{E}\big[\|\xi_k\|_\mu^3\big]^{1/3} \\
&\leq \big(1 - \alpha(1-\gamma)\big)\mathbb{E}\big[\|U_k\|_\mu^3\big]^{1/3} + \alpha\,\mathbb{E}\big[\|\xi_k\|_\mu^3\big]^{1/3},
\end{aligned}$$

where (a) follows from Minkowski inequality.

From assumption, we have $\mathbb{E}[\|\xi_k\|_2^3] < \infty$ (hence $\mathbb{E}[\|\xi_k\|_\mu^3] < \infty$ because $\|x\|_\mu \leq \sqrt{\mu_{\max}}\|x\|_2$ with $\mu_{\max} := \max_i \mu_i$). Letting $k \to \infty$, we have $U_\infty \overset{d}{=} U_{\infty+1}$, so rearranging gives

$$\mathbb{E}\big[\|U^{(\alpha)}\|_\mu^3\big]^{1/3} \leq \frac{\alpha\,\mathbb{E}\big[\|\xi_k\|_\mu^3\big]^{1/3}}{1 - \big(1 - \alpha(1-\gamma)\big)} = \frac{\mathbb{E}\big[\|\xi_k\|_\mu^3\big]^{1/3}}{1-\gamma}, \qquad \forall\alpha \in (0,1). \quad (66)$$

Now define $Y^{(\alpha)} := \frac{X^{(\alpha)} - x^\star}{\sqrt{\alpha}} = \frac{U^{(\alpha)}}{\sqrt{\alpha}}$. Then

$$\begin{aligned}
\mathbb{E}\big[\|Y^{(\alpha)}\|_2^3\big]^{1/3} &= \frac{1}{\sqrt{\alpha}}\mathbb{E}\big[\|U^{(\alpha)}\|_2^3\big]^{1/3} \leq \frac{1}{\sqrt{\alpha}}\frac{1}{\sqrt{\mu_{\min}}}\mathbb{E}\big[\|U^{(\alpha)}\|_\mu^3\big]^{1/3} \\
&\overset{(a)}{\leq} \underbrace{\frac{1}{\sqrt{\alpha}}\frac{1}{\sqrt{\mu_{\min}}} \cdot \frac{\mathbb{E}\big[\|\xi_k\|_\mu^3\big]^{1/3}}{1-\gamma}}_{:=A_{\mathrm{CSA}}(\alpha)},
\end{aligned}$$

where $\mu_{\min} := \min_i \mu_i > 0$, and inequality (a) uses (66). Equivalently, using $\mathbb{E}[\|\xi_k\|_\mu^3]^{1/3} \leq \sqrt{\mu_{\max}}\mathbb{E}[\|\xi_k\|_2^3]^{1/3}$,

$$A_{\mathrm{CSA}}(\alpha) \ \leq \ \frac{1}{\sqrt{\alpha}}\frac{\sqrt{\mu_{\max}/\mu_{\min}}}{1-\gamma}\,\mathbb{E}\big[\|\xi_k\|_2^3\big]^{1/3}.$$

This verifies the desired third-moment drift bound under the $\|\cdot\|_\mu$-contraction assumption.

Having verified conditions (i)–(iii) of Lemma C.7, we can apply the Lemma to attain Theorem 3.11 as desired. We present the final result here for completeness. Recall the definition of $Y^{(\alpha)}, Y$ in Theorem 3.11, Assumptions 3.8, and Assumption 3.10, we present the following Wasserstein bounds.

$d_W(Y^{(\alpha)}, Y) \leq U_3 \alpha^{1/2} \log(1/\alpha)$

where $U_3 := MA_{\mathrm{CSA}}^2 g_{1,Y} + \dfrac{(1+\gamma)^2\mu_{\max}/\mu_{\min}}{2}g_{2,Y}A_{\mathrm{CSA}}^2 + g_{3,Y}\big(1 + ((1+\gamma)\sqrt{\mu_{\max}/\mu_{\min}})^3 A_{\mathrm{CSA}}^3 + \mathbb{E}[\|\xi\|^3]\big),$

(67)

with $A_{\mathrm{CSA}}$ defined above, and $g_{1,Y}, g_{2,Y}, g_{3,Y}$ defined in Proposition C.3. Applying Corollary C.9, we achieve the following tail bounds.

$$\big|\mathbb{P}(\langle Y^{(\alpha)}, \zeta\rangle > a) - \mathbb{P}(\langle Y, \zeta\rangle > a)\big| \leq U_3'\frac{\alpha^{1/4}\log^{1/2}(1/\alpha)}{a},$$

$$\text{where } U_3' := (8\sqrt{\|\Sigma_Y\|_{op}} + 1)U_2^{1/2}.$$

(68)

## C.9. An alternative proof via Stein pair

In this section, we will show that one can achieve a Wasserstein distance bound of $O(\sqrt{\alpha})$ to a specialized case in 1-dimension. Though our general framework can recover such a bound by applying stronger Stein bounds in 1-dimension, we will demonstrate a technique introduced by Stein known as the method of exchangeable pairs.

Consider the SGD algorithm the case when $d = 1$, $f(x) = x^2/2$, and the noise sequence $\{\xi_k\}_{k\geq 0}$ is i.i.d. standard normal, then Algorithm (1) reduces to

$$X_{k+1}^{(\alpha)} \ = \ (1-\alpha)\,X_k^{(\alpha)} \ + \ \alpha\,\xi_k, \qquad k \geq 0.$$

(69)

To analyze the limiting distribution, we must analyze the *centered, scaled* iterate

$$Y_k^{(\alpha)} \ = \ \frac{X_k^{(\alpha)}}{\sqrt{\alpha}},$$

which yields the recursion

$$Y_k^{(\alpha)} = (1-\alpha)\,Y_{k-1}^{(\alpha)} + \sqrt{\alpha}\,w_{k-1}, \qquad k \geq 0$$

(70)

which is the result of dividing both sides of (69) by $\sqrt{\alpha}$. By recursion, we obtain

$$\begin{aligned}
Y_k^{(\alpha)} &= (1-\alpha)\,Y_{k-1}^{(\alpha)} + \sqrt{\alpha}\,w_{k-1} \\
&= (1-\alpha)^2\,Y_{k-2}^{(\alpha)} + (1-\alpha)\sqrt{\alpha}\,w_{k-2} + \sqrt{\alpha}\,w_{k-1} \\
&\ \ \vdots \\
&= (1-\alpha)^k\,Y_0^{(\alpha)} + \sum_{i=0}^{k-1}(1-\alpha)^{k-1-i}\sqrt{\alpha}\,w_i.
\end{aligned}$$

Now, we shall introduce our method of exchangeable pairs.

**Definition C.10.** The ordered pair of random variables $(W', W)$ is an exchangeable pair if $(W', W) \overset{d}{=} (W, W')$. If for some $0 < a \leq 1$

$$\mathbb{E}[W'|W] = (1-a)W$$

then $(W', W)$ is a a-Stein pair.

We define our W as

$$W = \frac{Y_k^{(\alpha)} - \mathbb{E}[Y_k^{(\alpha)}]}{\sqrt{\mathrm{Var}(Y^{(\alpha)})}}$$

$$= \frac{1}{\sigma} \sum_{i=0}^{k-1} (1-\alpha)^{k-1-i} \sqrt{\alpha}\, w_i$$

where

$$\mathrm{Var}\left[Y_k^{(\alpha)}\right] = \mathrm{Var}\left[(1-\alpha)^k Y_0^{(\alpha)} + \sum_{i=0}^{k-1}(1-\alpha)^{k-1-i}\sqrt{\alpha}W_i\right]$$

$$= \alpha \sum_{i=0}^{k-1}(1-\alpha)^{2i}$$

$$= \frac{1}{2-\alpha}\left(1 - (1-\alpha)^{2k}\right)$$

$$= \sigma^2$$

We construct an *exchangeable pair* by choosing an index uniformly at random and replacing it by an independent copy. Formally, let $I$ be uniform on $\{1, \ldots, n\}$, $(w_1', \ldots w_n')$ an independent copy of $(w_1, \ldots, w_n)$, and define

$$W' = W - \frac{1}{\sigma}(1-\alpha)^{k-1-i}\sqrt{\alpha}\, w_i + \frac{1}{\sigma}(1-\alpha)^{k-1-i}\sqrt{\alpha}\, w_i'.$$

It is straightforward to check that $(W, W')$ is exchangeable, and we now verify it is also a $(1/n)$-Stein pair. The calculation is direct; in the penultimate equality we use the independence of $w_i$ and $w_i'$ and the fact that $\mathbb{E}[w_i] = 0$:

$$\mathbb{E}[W' - W \mid (w_1, \ldots, w_n)] = \mathbb{E}[w_I' - w_I \mid (w_1, \ldots, w_n)]$$

$$= \frac{1}{n\sigma} \sum_{i=1}^{n}(1-\alpha)^{k-1-i}\sqrt{\alpha}\,\mathbb{E}[w_i' - w_i \mid (w_1, \ldots, w_n)]$$

$$= -\frac{1}{n\sigma} \sum_{i=1}^{n}(1-\alpha)^{k-1-i}\sqrt{\alpha}\, w_i = -\frac{W}{n}.$$

Now, with our construction of exchangeable pairs, we estimate the bound on the distance using Theorem 3.7 of (Ross, 2011).

**Proposition C.11.** *If $(W, W')$ is an $a$-Stein pair with $\mathbb{E}[W^2] = 1$ and $Z \sim \mathcal{N}(0, 1)$, then*

$$d_{\mathrm{W}}(W, Z) \leq \frac{\sqrt{\mathrm{Var}\big(\mathbb{E}[(W' - W)^2 \mid W]\big)}}{\sqrt{2\pi}\, a} + \frac{\mathbb{E}|W' - W|^3}{3a}.$$

Using Theorem 1, we first bound,

$$\mathbb{E}[|W' - W|^3] = \frac{1}{n} \sum_{i=1}^{n} \frac{1}{\sigma^3}(1-\alpha)^{3(k-1-i)} \alpha^{\frac{3}{2}}\, \mathbb{E}[|w_i' - w_i|^3]$$

$$\leq \frac{8}{n\sigma^3} \alpha^{\frac{3}{2}} \mathbb{E}[|w_i|^3] \sum_{i=1}^{n}(1-\alpha)^{3(k-1-i)}$$

using AM-GM inequality for the cross terms. Next, we compute

$$\mathbb{E}[(W' - W)^2 \mid W] = \frac{1}{n} \sum_{i=1}^{n} \frac{1}{\sigma}(1-\alpha)^{2(k-1-i)}\alpha\,(1 + w_i^2)$$

Hence,

$$\mathrm{Var}\big(\mathbb{E}[(W'-W)^2 \mid W]\big) \le \frac{1}{n^2\sigma^2}\alpha^2\,\mathbb{E}[|w_i|^4]\sum_{i=1}^{n}(1-\alpha)^{4(k-1-i)}$$

Combing both estimates above and plugging in $a = 1/n$, we have

$$d_{\mathrm{W}}(W_k, Z) \le \sqrt{\frac{2\pi n^2}{n^2\sigma^2}\alpha^2\,\mathbb{E}[|w_i|^4]\sum_{i=1}^{n}(1-\alpha)^{4(k-1-i)}} + \frac{8n}{3n\sigma^3}\alpha^{\frac{3}{2}}\mathbb{E}[|w_i|^3]\sum_{i=1}^{n}(1-\alpha)^{3(k-1-i)}$$

$$= \frac{\sqrt{2\pi}}{\sigma}\alpha\sqrt{\mathbb{E}[|w_i|^4]\sum_{i=1}^{n}(1-\alpha)^{4(k-1-i)}} + \frac{8}{3\sigma^3}\alpha^{\frac{3}{2}}\mathbb{E}[|w_i|^3]\sum_{i=1}^{n}(1-\alpha)^{3(k-1-i)}$$

Then, assuming $\alpha \in (0,1)$

$$d_{\mathrm{W}}(W, Z) = \lim_{n\to\infty}d_{\mathrm{W}}(W_k, Z) \le \sqrt{2\pi(2-\alpha)}\alpha\sqrt{\mathbb{E}[|w_i|^4]\frac{1}{1-(1-\alpha)^4}} + \frac{8(2-\alpha)^{\frac{3}{2}}}{3}\alpha^{\frac{3}{2}}\mathbb{E}[|w_i|^3]\frac{1}{1-(1-\alpha)^3}$$

Notice when $\alpha \in (0,1)$, we have that $\frac{1}{1-(1-\alpha)^4} = \frac{1}{\alpha(1+(1-\alpha)^2)(2-\alpha)} \le \frac{1}{\alpha}$ and $\frac{1}{1-(1-\alpha)^3} = \frac{1}{\alpha((1-\alpha)^2+(1-\alpha)+1)} \le \frac{1}{\alpha}$ That gives the final bound of:

$$d_{\mathrm{W}}(W, Z) \le \left(\sqrt{2\pi(2-\alpha)\mathbb{E}[|w_i|^4]} + \frac{8(2-\alpha)^{\frac{3}{2}}}{3}\mathbb{E}[|w_i|^3]\right)\alpha^{\frac{1}{2}} \tag{71}$$

## D. Proofs under Markovian Noise

In this section, we will presen the proof for Markovian noise model. We will use similar Stein's method framework as in i.i.d proof. Yet we highlight that under Markovian noise setting, the temporal correlation will cause main technical challenges to the analysis. We introduce the Poisson equation technique to handle the correlated noise. Similar to i.i.d case, we begin with a general purpose lemma to achieve Wasserstein bounds. Then we will justify the assumptions of this lemma in each models and apply the lemma with concentration lemma C.8 to achieve our target Theorem 4.3.

### D.1. A General Purpose Lemma under Markovian Noise

**Lemma D.1** (Markovian Gaussian approximation). *Consider the constant-stepsize SA recursion*

$$X_{k+1}^{(\alpha)} = X_k^{(\alpha)} + \alpha\big(F(X_k^{(\alpha)}) + \xi(Z_k)\big), \tag{72}$$

*where $X_0^{(\alpha)} \in \mathbb{R}^d$ and $\alpha > 0$ is fixed. Suppose the Markov chain $\{X_k^{(\alpha)}\}_{k\ge 0}$ admits a unique stationary distribution $\pi_\alpha^X$. Let $X^{(\alpha)} \sim \pi_\alpha^X$ and define $Y^{(\alpha)} := (X^{(\alpha)} - x^\star)/\sqrt{\alpha}$. Further suppose Assumptions C.6 and Assumption 4 hold, and that the stationary laws have a uniformly bounded fourth moment, i.e., there exist constants $\alpha_0 \in (0,1]$ and $A \in (0,\infty)$ such that*

$$\sup_{\alpha\in(0,\alpha_0)}\mathbb{E}[\|Y^{(\alpha)}\|_2^4] \le A, \tag{73}$$

*Let $J^\star := DF(x^\star) \in \mathbb{R}^{d\times d}$ be the Jacobian of $F$ at $x^\star$, and recall $\Sigma_M \in \mathbb{R}^{d\times d}$ is the long-run noise covariance defined in 4. Assume that the Lyapunov equation*

$$J^\star\Sigma_Y + \Sigma_Y(J^\star)^\top = -\Sigma_M \tag{74}$$

*admits a unique symmetric positive definite solution $\Sigma_Y \in \mathbb{R}^{d\times d}$. Define the target Gaussian distribution $Y \sim \mathcal{N}(0, \Sigma_Y)$.*

*Then there exists $\alpha_1 \in (0,1]$ and constants $\bar{U} \in (0,\infty)$ (depending on $d, L, M, A$, noise parameters, and spectral properties of $J^\star$) such that for all $\alpha \in (0,\alpha_0)$,*

$$d_W\big(\mathcal{L}(Y^{(\alpha)}), \mathcal{L}(Y)\big) \le \bar{U}\sqrt{\alpha}\log(1/\alpha). \tag{75}$$

*The constants $\bar{U}$ is defined explicitly in (87).*

The next two subsections are on proving this Lemma, in D.1.1, we set up Stein's Method, and in D.1.2 we show how to use the Poisson Equation to extend the computations in the iid noise case to Markovian noise.

D.1.1. GENERATOR COUPLING OF $Y$ AND $Y^{(\alpha)}$

As in the i.i.d. noise case, the core idea of this proof is to apply Stein's Method to the SA iterates. To implement this, we must correctly define the generators that we will compare. We highlight these below. We also note that the primary difference from i.i.d is the covariance matrix in the Lyapunov equation. In particular, the right hand side of the Lyapunov equation will correspond to the long-run covariance matrix.

The generator of the OU process is

$$\mathcal{L}g(y) \;=\; \frac{1}{2}\operatorname{Tr}\!\big(\Sigma_M \nabla^2 g(y)\big) + \big\langle J^\star y, \nabla g(y)\big\rangle. \tag{76}$$

We will compare the stationary law of the scaled iterates to this Gaussian distribution via Stein's method. With a slight abuse of notation, we will use $\pi$ to also denote the joint stationary law of the Markov chain $(Y_k^{(\alpha)}, Z_k)$, where $Y_k^{(\alpha)} = (X_k^{(\alpha)} - x^\star)/\sqrt{\alpha}$. And we let $\pi_Z$ denote the stationary law of the Markov chain $\{Z_k\}_{k\geq 0}$. We will also denote $Y^{(\alpha)} \sim \mathrm{Law}(Y^{(\alpha)})$, and we let $Y_0 = Y^{(\alpha)}$. We use $Z_0$ distributed as the stationary distribution of the Markov chain $\{Z_k\}_{k\geq 0}$. We use $Y_1$ and $Z_1$ to denote the next step of the chain. Recall that we use $Y \sim \mathcal{N}(0, \Sigma_Y)$ to denote the target Gaussian distribution, where $\Sigma_Y$ is the solution to the Lyapunov equation (74).

Note that under Assumptions C.6 and 4, we have that for $\alpha$ sufficiently small the joint chain admits a unique stationary law $\pi_\alpha$ and that $\mathbb{E}_{\pi_\alpha}\big[\|Y_0^{(\alpha)}\|^4\big] < \infty$. Therefore, given a test function $g : \mathbb{R}^d \to \mathbb{R}$, we define the discrete Stein operator associated with the stationary rescaled chain by

$$\mathcal{L}^{(\alpha)}g(y,z) \;:=\; \frac{1}{\alpha}\mathbb{E}\!\left[g\big(Y_1^{(\alpha)}\big) - g\big(Y_0^{(\alpha)}\big) \,\Big|\, Y_0^{(\alpha)} = y,\; Z_0 = z\right]. \tag{77}$$

The proof of this being a valid operator for the chain is exactly as in the i.i.d. case.

Now that we have defined the necessary generators, we will instantiate Stein's method. Denoting $J$ to be the lift operator, so that $Jg(y,z) = g(y)$. We have,

$$
\begin{aligned}
d_W\!\Big(Y^{(\alpha)}, Y\Big) &= \sup_{h\in\mathrm{Lip}_1(\mathbb{R}^d)} \mathbb{E}_{(Y_0,Z_0)\sim\mathrm{Law}(Y^{(\alpha)},\pi_Z)} h(Y_0) - \mathbb{E}_{Y\sim\mathrm{Law}(Y)}h(Y) \\
&= \sup_{h\in(\mathrm{Lip}_1(\mathbb{R}^d)\cap C^1)} \mathbb{E}_{(Y_0,Z_0)\sim\mathrm{Law}(Y^{(\alpha)},\pi_Z)}\big[h(Y_0) - \mathbb{E}_{Y\sim\mathrm{Law}(Y)}h(Y)\big] \\
&\leq \sup_{g_h\in G(J^*,\Sigma_M)} \mathbb{E}_{(Y_0,Z_0)\sim\mathrm{Law}(Y^{(\alpha)},\pi_Z)}\big[h(Y_0) - \mathbb{E}_{Y\sim\mathrm{Law}(Y)}h(Y)\big] \\
&\leq \sup_{g_h\in G(J^*,\Sigma_M)} \mathbb{E}_{(Y_0,Z_0)\sim\mathrm{Law}(Y^{(\alpha)},\pi_Z)}\big[\mathcal{L}g_h(Y_0)\big]
\end{aligned}
$$

giving us the classical "generator coupling" setup, recalling that $G(J^*, \Sigma_M)$ is the class of solutions to the Poisson equation defined in (27). In the following, we will use $e_1(\cdot)$ to denote the lifting function, i.e., $e_1(y,z) = y$ so as to handle the notation more easily.

D.1.2. BOUNDING THE GENERATOR DIFFERENCE VIA THE POISSON EQUATION

Plugging in the definition of $\mathcal{L}^{(\alpha)}$ and $\mathcal{L}$, we have

$$
\begin{aligned}
&\mathcal{L}g_h(y) - \mathcal{L}^{(\alpha)}(g_h \circ e_1)(y,z) \\
&= \frac{1}{2}\operatorname{tr}\!\big(\Sigma_M \nabla^2 g_h(y)\big) + \big\langle J^\star y, \nabla g_h(y)\big\rangle - \frac{1}{\alpha}\mathbb{E}\big[g_h(Y_1) - g_h(Y_0)\,\big|\,(Y_0,Z_0)=(y,z)\big].
\end{aligned} \tag{78}
$$

We will now Taylor expand $g_h(Y_1) - g_h(Y_0)$ in the above expression. Abstracting away the third-order remainder term as $R_3(Y_1, Y_0)$, we have that

$$
\begin{aligned}
(78) ={}& \frac{1}{2}\operatorname{tr}\!\big(\Sigma_M \nabla^2 g_h(y)\big) + \big\langle J^\star y, \nabla g_h(y)\big\rangle \\
&- \frac{1}{\alpha}\mathbb{E}\!\left[\nabla g_h(Y_0)^\top (Y_1 - Y_0) + \frac{1}{2}(Y_1 - Y_0)^\top \nabla^2 g_h(Y_0)(Y_1 - Y_0) + R_3(Y_1, Y_0)\,\big|\,(Y_0,Z_0)=(y,z)\right]
\end{aligned}
$$

$$:= W_1 + W_2 + R_3.$$

where

$$W_1 = \langle J^\star y, \nabla g_h(y) \rangle - \frac{1}{\alpha} \mathbb{E}\big[\nabla g_h(Y_0)^\top (Y_1 - Y_0) \,\big|\, (Y_0, Z_0) = (y, z)\big],$$

$$W_2 = \frac{1}{2} \operatorname{tr}\!\big(\Sigma_M \nabla^2 g_h(y)\big)$$
$$\qquad - \frac{1}{\alpha} \mathbb{E}\bigg[\frac{1}{2}(Y_1 - Y_0)^\top \nabla^2 g_h(Y_0)(Y_1 - Y_0) \,\bigg|\, (Y_0, Z_0) = (y, z)\bigg]$$

$$R = \frac{-1}{\alpha} \mathbb{E}\big[R_3(Y_1, Y_0) \,\big|\, (Y_0, Z_0) = (y, z)\big].$$

**Handling the Remainder Term.** We bound the remainder term $R$ here before proceeding to bound $W_1$ and $W_2$. By Taylor's theorem with integral remainder and the definition of conditional expectation, we have that

$$\mathbb{E}[R] = \frac{-1}{\alpha} \mathbb{E}\bigg[\frac{1}{2}\int_0^1 (1-t)(Y_1 - Y_0)^\top \big(\nabla^2 g_h(Y_0 + t(Y_1 - Y_0)) - \nabla^2 g_h(Y_0)\big)(Y_1 - Y_0)\, dt\bigg]$$

$$\overset{(a)}{\le} g_{3,Y} \frac{1}{1-\beta} \cdot \bigg(\frac{\alpha^{\frac{\beta}{2}}}{2}\int_0^1 (1-t)|t|^\beta\, dt\bigg) \mathbb{E}\Big[\big\| F(x^* + \sqrt{\alpha}\,Y_0) + \xi(Z_0)\big\|^{\beta+2}\Big]$$

$$\le g_{3,Y} \frac{1}{1-\beta} \cdot \frac{\alpha^{\beta/2}}{2(\beta+1)(\beta+2)} \mathbb{E}\Big[\big\| F(x^* + \sqrt{\alpha}\,Y_0) + \xi(Z_0)\big\|^{\beta+2}\Big]$$

$$\le g_{3,Y} \frac{1}{1-\beta} \cdot \frac{\alpha^{\beta/2}}{2(\beta+1)(\beta+2)} \mathbb{E}\Big[2^{\beta+1}\big(1 + \| F(x^* + \sqrt{\alpha}\,Y_0)\|^3 + \|\xi(Z_0)\|^3\big)\Big]$$

$$\le g_{3,Y} \frac{1}{1-\beta} \cdot \frac{\alpha^{\beta/2}2^\beta}{(\beta+1)(\beta+2)}\Big(1 + \mathbb{E}\big[(L\sqrt{\alpha}\,\|Y_0\|)^3\big] + \mathbb{E}\big[\|\xi(Z_0)\|^3\big]\Big) \tag{79}$$

Here inequality $(a)$ follows from the $\beta$-Hölder continuity of the Hessian of $g_h$ in Proposition C.3.

**Decomposing $W_1$ via the First Poisson Equation.** Recall that in the i.i.d. case, $W_1$ and $W_2$ are handled separately. With Markovian noise, we must handle them crucially together. Intuitively this is because the long run covariance has both linear and quadratic pieces – will show this explicitly later in the section. Prior to this, we will massage $W_1$ into a form easier to handle. We first focus on $\mathbb{E}_{(Y_0, Z_0)\sim\text{Law}(Y^{(\alpha)}, Z)}[W_1]$.

$$\frac{1}{\alpha} \mathbb{E}\big[\nabla g_h(Y_0)^\top (Y_1 - Y_0) \,\big|\, (Y_0, Z_0) = (y, z)\big]$$

$$= \frac{1}{\sqrt{\alpha}}\big(F(x^* + \sqrt{\alpha}\,y) + \xi(z)\big)^\top \nabla g_h(y)$$

$$= \underbrace{\frac{1}{\sqrt{\alpha}} \nabla g_h(y)^\top F(x^* + \sqrt{\alpha}\,y)}_{:=T_1} + \underbrace{\frac{1}{\sqrt{\alpha}} \nabla g_h(y)^\top \xi(z)}_{:=T_2}.$$

Note that in the i.i.d. case, $T_2$ vanishes, so we are done. In the Markovian case, we must apply the Poisson Equation to $T_2$:

$$\mathbb{E}[T_2] = \mathbb{E}\bigg[\frac{1}{\sqrt{\alpha}} \nabla g_h(Y_0)^\top \xi(Z_0)\bigg]$$

$$= \frac{1}{\sqrt{\alpha}}\Big(\mathbb{E}\big[V(Z_0)^\top \nabla g_h(Y_0)\big] - \mathbb{E}\big[\mathbb{E}[V(Z_1)\mid Z_0]^\top \nabla g_h(Y_0)\big]\Big)$$

$$\overset{(a)}{=} \frac{1}{\sqrt{\alpha}} \mathbb{E}\big[V(Z_1)^\top\big(\nabla g_h(Y_1) - \nabla g_h(Y_0)\big)\big], \tag{80}$$

where the equality $(a)$ is valid as $\mathbb{E}[V(Z_0)^\top \nabla g_h(Y_0)] \leq \sup_{z \in \mathsf{Z}} \|V(Z_0)\|_2 \cdot \sup_{z \in \mathsf{Z}} \|\nabla g_h(Y_0)\|_2 < \infty$ by the boundedness of the Stein Solution and $V$, where the latter is assumed in Assumption 4.2. We will now express the above in integral form, and plug in $Y_1 - Y_0 = \sqrt{\alpha}\big(F(x^* + \sqrt{\alpha}\, Y_0) + \xi(Z_0)\big)$.

$$(80) = \frac{1}{\sqrt{\alpha}} \mathbb{E}\left[V(Z_1)^\top \int_0^1 \nabla^2 g_h\big(Y_0 + t(Y_1 - Y_0)\big)\,(Y_1 - Y_0)\,dt\right]$$

$$= \mathbb{E}\left[V(Z_1)^\top \int_0^1 \nabla^2 g_h\big(Y_0 + t(Y_1 - Y_0)\big)\big(F(x^* + \sqrt{\alpha}\, Y_0) + \xi(Z_0)\big)\,dt\right]$$

$$= \mathbb{E}\big[V(Z_1)^\top \nabla^2 g_h(Y_0)\,F(x^* + \sqrt{\alpha}\, Y_0)\big] + \mathbb{E}\big[V(Z_1)^\top \nabla^2 g_h(Y_0)\,\xi(Z_0)\big]$$

$$+ \mathbb{E}\left[V(Z_1)^\top \int_0^1 \big(\nabla^2 g_h\big(Y_0 + t(Y_1 - Y_0)\big) - \nabla^2 g_h(Y_0)\big)\big(F(x^* + \sqrt{\alpha}\, Y_0) + \xi(Z_0)\big)\,dt\right]. \qquad (81)$$

We now bound the magnitude of the third term of (81) Using the $\beta$-Hölder bound again from Proposition C.3, we have

$$\left|\mathbb{E}\left[V(Z_1)^\top \int_0^1 \big(\nabla^2 g_h\big(Y_0 + t(Y_1 - Y_0)\big) - \nabla^2 g_h(Y_0)\big)\big(F(x^* + \sqrt{\alpha}\, Y_0) + \xi(Z_0)\big)\,dt\right]\right|$$

$$\leq \mathbb{E}\left[\|V(Z_1)\| \int_0^1 \big\|\nabla^2 g_h\big(Y_0 + t(Y_1 - Y_0)\big) - \nabla^2 g_h(Y_0)\big\|_{op}\big\|F(x^* + \sqrt{\alpha}\, Y_0) + \xi(Z_0)\big\|\,dt\right]$$

$$\overset{(a)}{\leq} \mathbb{E}\left[\|V(Z_1)\| \int_0^1 \frac{g_{3,Y}}{1-\beta}\|t(Y_1 - Y_0)\|_2^\beta \big\|F(x^* + \sqrt{\alpha}\, Y_0) + \xi(Z_0)\big\|\,dt\right]$$

$$\leq \mathbb{E}\left[\|V(Z_1)\| \frac{g_{3,Y}}{1-\beta}\big\|\sqrt{\alpha}\big(F(x^* + \sqrt{\alpha}\, Y_0) + \xi(Z_0)\big)\big\|^\beta \big\|F(x^* + \sqrt{\alpha}\, Y_0) + \xi(Z_0)\big\| \int_0^1 |t|^\beta dt\right]$$

$$= \alpha^{\frac{\beta}{2}}\frac{g_{3,Y}/(1-\beta)}{\beta+1}\mathbb{E}\left[\|V(Z_1)\|\,\big\|F(x^* + \sqrt{\alpha}\, Y_0) + \xi(Z_0)\big\|^{\beta+1}\right]$$

$$= \alpha^{\frac{\beta}{2}}\frac{g_{3,Y}/(1-\beta)}{\beta+1}\mathbb{E}\left[\|V(Z_1)\|\,\big\|F(x^* + \sqrt{\alpha}\, Y_0) - F(x^*) + \xi(Z_0)\big\|^{\beta+1}\right]$$

$$\leq \alpha^{\frac{\beta}{2}}\frac{g_{3,Y}/(1-\beta)}{\beta+1}\mathbb{E}\left[\|V(Z_1)\|\, 2^{\frac{(\beta+1)}{2}}\big(1 + \|F(x^* + \sqrt{\alpha}Y_0) - F(x^*)\|^2 + 2\|\xi(Z_0)\|^2\big)\right]$$

$$\leq \alpha^{\frac{\beta}{2}}\frac{g_{3,Y}/(1-\beta)}{\beta+1}\mathbb{E}\left[\|V(Z_1)\|\, 2^{\frac{(\beta+1)}{2}}\big(1 + L^2\alpha\|Y_0\|^2 + \|\xi(Z_0)\|^2\big)\right]$$

$$\leq 2\alpha^{\frac{\beta}{2}}\frac{g_{3,Y}/(1-\beta)}{\beta+1}\Big(L^2\alpha\,(\mathbb{E}\|V(Z_1)\|^2)^{1/2}(\mathbb{E}\|Y_0\|^4)^{1/2} + (\mathbb{E}\|V(Z_1)\|^2)^{1/2}(\mathbb{E}\|\xi(Z_0)\|^4)^{1/2} + (\mathbb{E}\|V(Z_1)\|^2)^{1/2}\Big)$$

$$= 2\alpha^{\frac{\beta}{2}}\frac{g_{3,Y}/(1-\beta)}{\beta+1}(\mathbb{E}\|V(Z_1)\|^2)^{1/2}\Big(L^2\alpha\,(\mathbb{E}\|Y_0\|^4)^{1/2} + (\mathbb{E}\|\xi(Z_0)\|^4)^{1/2} + 1\Big) \qquad (82)$$

where the inequality $(a)$ follows from the $\beta$-Hölder continuity of the Hessian of $g_h$ in Proposition C.3. We note that since we have bounded moments of $V$, $\|Y_0\|$ and $\|\xi(Z_0)\|$. The final expression is therefore of orde $O(\alpha^{\beta/2})$. This yields that

$$\mathbb{E}[W_1] = \mathbb{E}\big[\langle J^\star Y_0, \nabla g_h(Y_0)\rangle\big] - \frac{1}{\sqrt{\alpha}}\mathbb{E}\big[\nabla g_h(Y_0)^\top F(x^* + \sqrt{\alpha}\, Y_0)\big]$$

$$- \mathbb{E}\big[V(Z_1)^\top \nabla^2 g_h(Y_0)\,F(x^* + \sqrt{\alpha}\, Y_0)\big] - \mathbb{E}\big[V(Z_1)^\top \nabla^2 g_h(Y_0)\,\xi(Z_0)\big] + O(\alpha^{\beta/2}) \qquad (83)$$

For the right-hand side of inequality (83), we handle the first two terms together, and the last two terms together. For the first two terms on the first line, we compute (almost exactly as in the i.i.d. case):

$$\mathbb{E}\big[\langle J^\star Y_0, \nabla g_h(Y_0)\rangle\big] - \frac{1}{\sqrt{\alpha}}\mathbb{E}\big[\nabla g_h(Y_0)^\top F(x^* + \sqrt{\alpha}\, Y_0)\big]$$

$$= \mathbb{E}\left[\nabla g_h(Y_0)^\top J^\star Y_0 - \nabla g_h(Y_0)^\top \frac{1}{\sqrt{\alpha}}F(x^* + \sqrt{\alpha}\, Y_0)\right]$$

$$= \mathbb{E}\left[\nabla g_h(Y_0)^\top J^\star Y_0 - \int_0^1 \nabla g_h(Y_0)^\top DF(x^* + t\sqrt{\alpha}Y_0)\,Y_0\,dt\right]$$

$$= \mathbb{E}\left[\int_0^1 \nabla g_h(Y_0)^\top \big(DF(x^*) - DF(x^* + t\sqrt{\alpha}Y_0)\big) Y_0 \, dt\right]$$

$$= -\mathbb{E}\left[\int_0^1 \int_0^t \sqrt{\alpha} \, \nabla g_h(Y_0)^\top \big(D^2 F(x^* + u\sqrt{\alpha}Y_0)[Y_0]\big) Y_0 \, du \, dt\right].$$

By definition of the tensor norm and the operator norm, we have, for all $u, t \in [0, 1]$,

$$\left|\sqrt{\alpha} \, \nabla g_h(y)^\top \big(D^2 F(x^* + u\sqrt{\alpha}y)[y]\big) y\right| \le \sqrt{\alpha} \, \|D^2 F\|_\infty \, \|\nabla g_h\|_\infty \, \|Y_0\|^2.$$

Since $\mathbb{E}\|Y_0\|^2 < \infty$, the integrand is integrable and we may apply Fubini-Tonelli theorem to interchange the expectation with the integrals. Hence

$$\mathbb{E}\left[\int_0^1 \int_0^t \sqrt{\alpha} \, \nabla g_h(Y_0)^\top \big(D^2 F(x^* + u\sqrt{\alpha}Y_0)[Y_0]\big) Y_0 \, du \, dt\right]$$

$$= \int_0^1 \int_0^t \sqrt{\alpha} \, \mathbb{E}\big[\nabla g_h(Y_0)^\top \big(D^2 F(x^* + u\sqrt{\alpha}Y_0)[Y_0]\big) Y_0\big] \, du \, dt.$$

We now bound the expectation using the operator norm:

$$\left|\mathbb{E}\big[\nabla g_h(Y_0)^\top \big(D^2 F(x^* + u\sqrt{\alpha}Y_0)[Y_0]\big) Y_0\big]\right| \le \mathbb{E}\big[\|D^2 F(x^* + u\sqrt{\alpha}Y_0)[Y_0]\| \, \|Y_0\| \, \|\nabla g_h(Y_0)\|\big]$$

$$\le \|D^2 F\|_\infty \, \|\nabla g_h\|_\infty \, \mathbb{E}\big[\|Y_0\|^2\big].$$

Therefore,

$$\left|\mathbb{E}\big[\langle J^\star Y_0, \nabla g_h(Y_0)\rangle\big] - \frac{1}{\sqrt{\alpha}} \mathbb{E}\big[\nabla g_h(Y_0)^\top F(x^* + \sqrt{\alpha}Y_0)\big]\right|$$

$$\le \int_0^1 \int_0^t \sqrt{\alpha} \, \|D^2 F\|_\infty \, \|\nabla g_h\|_\infty \, \mathbb{E}\big[\|Y_0\|^2\big] \, du \, dt$$

$$= \sqrt{\alpha} \, \|D^2 F\|_\infty \, \|\nabla g_h\|_\infty \, \mathbb{E}\big[\|Y_0\|^2\big] \int_0^1 \int_0^t du \, dt.$$

and we have

$$\left|\mathbb{E}\big[\langle J^\star Y_0, \nabla g_h(Y_0)\rangle\big] - \frac{1}{\sqrt{\alpha}} \mathbb{E}\big[\nabla g_h(Y_0)^\top F(x^* + \sqrt{\alpha}Y_0)\big]\right| \le \frac{\sqrt{\alpha}}{2} \, \|D^2 F\|_\infty \, \|\nabla g_h\|_\infty \, \mathbb{E}\big[\|Y_0\|^2\big] \tag{84}$$

Combining (83) and (84), we have the following order analysis. So, plugging into (44), we obtain

$$W_1 = -\mathbb{E}\big[V(Z_1)^\top \nabla^2 g_h(Y_0) \, F(x^* + \sqrt{\alpha}Y_0)\big] - \mathbb{E}\big[V(Z_1)^\top \nabla^2 g_h(Y_0) \, \xi(Z_0)\big] + O(\sqrt{\alpha} + \alpha^{\beta/2}).$$

We delegate handling the last terms in above after we handle $W_2$, cause we will use some cancellations regarding the second Poisson equation.

**Decomposing $W_2$.**   We now expand $W_2$:

$$W_2 = \frac{1}{2} \text{tr}\big(\Sigma_M \nabla^2 g_h(y)\big) - \frac{1}{\alpha} \mathbb{E}\left[\frac{1}{2}(Y_1 - Y_0)^\top \nabla^2 g_h(Y_0)(Y_1 - Y_0) \,\Big|\, (Y_0, Z_0) = (y, z)\right]$$

$$= \frac{1}{2} \text{tr}\big(\Sigma_M \nabla^2 g_h(y)\big)$$

$$\quad - \mathbb{E}\left[\frac{1}{2}\big(F(x^* + \sqrt{\alpha}Y_0) + \xi(Z_0)\big)^\top \nabla^2 g_h(Y_0)\big(F(x^* + \sqrt{\alpha}Y_0) + \xi(Z_0)\big) \,\Big|\, (Y_0, Z_0) = (y, z)\right]$$

$$= \frac{1}{2} \text{tr}\big(\Sigma_M \nabla^2 g_h(y)\big) - \frac{1}{2} \mathbb{E}\big[F(x^* + \sqrt{\alpha}Y_0)^\top \nabla^2 g_h(Y_0) F(x^* + \sqrt{\alpha}Y_0) \,\big|\, (Y_0, Z_0) = (y, z)\big]$$

$$- \mathbb{E}\big[\xi(Z_0)^\top \nabla^2 g_h(Y_0) F(x^* + \sqrt{\alpha}\,Y_0)\,\big|\,(Y_0, Z_0) = (y, z)\big]$$

$$- \frac{1}{2}\,\mathbb{E}\big[\xi(Z_0)^\top \nabla^2 g_h(Y_0) \xi(Z_0)\,\big|\,(Y_0, Z_0) = (y, z)\big].$$

Hence, taking expectation under the stationary law $\pi$ of $(Y^{(\alpha)}, Z)$,

$$\mathbb{E}[W_2] = \frac{1}{2}\,\mathbb{E}\big[\mathrm{tr}(\Sigma_M \nabla^2 g_h(Y_0))\big] - \frac{1}{2}\,\mathbb{E}\big[F(x^* + \sqrt{\alpha}\,Y_0)^\top \nabla^2 g_h(Y_0)\,F(x^* + \sqrt{\alpha}\,Y_0)\big]$$

$$- \mathbb{E}\big[\xi(z)^\top \nabla^2 g_h(Y_0)\,F(x^* + \sqrt{\alpha}\,Y_0)\big] - \frac{1}{2}\,\mathbb{E}\big[\xi(Z_0)^\top \nabla^2 g_h(Y_0)\,\xi(Z_0)\big]$$

$$= \frac{1}{2}\,\mathbb{E}\big[\mathrm{tr}(\Sigma_M \nabla^2 g_h(Y_0))\big] - \frac{1}{2}\,\mathbb{E}\big[F(x^* + \sqrt{\alpha}\,Y_0)^\top \nabla^2 g_h(Y_0)\,F(x^* + \sqrt{\alpha}\,Y_0)\big]$$

$$- \mathbb{E}\big[V(Z_0)^\top \nabla^2 g_h(Y_0)\,F(x^* + \sqrt{\alpha}\,Y_0)\big] + \mathbb{E}\big[V(Z_1)^\top \nabla^2 g_h(Y_0)\,F(x^* + \sqrt{\alpha}\,Y_0)\big]$$

$$- \frac{1}{2}\,\mathbb{E}\big[\xi(Z_0)^\top \nabla^2 g_h(Y_0)\,\xi(Z_0)\big].$$

**Putting things together via a Second Poisson Equation** Putting $W_1$ and $W_2$ together yields:

$$\mathbb{E}[W_1 + W_2] = -\mathbb{E}\big[V(Z_1)^\top \nabla^2 g_h(Y_0)\,\xi(Z_0)\big] - \mathbb{E}\big[V(Z_0)^\top \nabla^2 g_h(Y_0)\,F(x^* + \sqrt{\alpha}Y_0)\big]$$

$$+ \tfrac{1}{2}\mathbb{E}\big[\mathrm{tr}(\Sigma_M \nabla^2 g_h(Y_0))\big] - \tfrac{1}{2}\mathbb{E}\big[\xi(Z_0)^\top \nabla^2 g_h(Y_0)\,\xi(Z_0)\big]$$

$$+ O(\sqrt{\alpha} + \alpha^{\beta/2})$$

$$= -\mathbb{E}\big[\xi(Z_0)^\top \nabla^2 g_h(Y_0)\,V(Z_0)\big] + \tfrac{1}{2}\mathbb{E}\big[\mathrm{tr}(\Sigma_M \nabla^2 g_h(Y_0))\big] + \tfrac{1}{2}\mathbb{E}\big[\xi(Z_0)^\top \nabla^2 g_h(Y_0)\,\xi(Z_0)\big]$$

$$- \mathbb{E}\big[V(Z_0)^\top \nabla^2 g_h(Y_0)\,F(x^* + \sqrt{\alpha}Y_0)\big] + O(\sqrt{\alpha} + \alpha^{\beta/2}).$$

We emphasize again that if $(Y_0, Z_0) \sim \pi$ were independent, we would be done, since

$$-\mathbb{E}\big[V(Z_0)\,\xi(Z_0)^\top\big] \;+\; \tfrac{1}{2}\Sigma_M \;+\; \tfrac{1}{2}\,\mathbb{E}\big[\xi(Z_0)\,\xi(Z_0)^\top\big] \;=\; 0,$$

by the definition of long run covariance. However, in the Markovian case, $Y_0$ and $Z_0$ are dependent. Thus, we apply the Poisson equation. Recalling the definition of $\Phi$ and $W$ in 4.2, we have the following.

$$\Phi(Z_0) := -V(Z_0)\,\xi(Z_0)^\top \;+\; \tfrac{1}{2}\Sigma_M \;+\; \tfrac{1}{2}\,\xi(Z_0)\,\xi(Z_0)^\top,$$

Then we have the following identity:

$$\mathbb{E}[W_1 + W_2] = \mathbb{E}[\langle \nabla^2 g_h(Y_0), \Phi(Z_0)\rangle] - \mathbb{E}\big[V(Z_0)^\top \nabla^2 g_h(Y_0)\,F(x^* + \sqrt{\alpha}\,Y_0)\big] + O(\sqrt{\alpha} + \alpha^{\beta/2}).$$

We now bound the two terms above. For the first term, we have

$$\big|\mathbb{E}[\langle \nabla^2 g_h(Y_0), \Phi(Z_0)\rangle]\big| = \big|\mathbb{E}\big[\mathrm{Tr}\big((\nabla^2 g_h(Y_0))^\top \Phi(Z_0)\big)\big]\big|$$

$$= \big|\mathbb{E}\big[\mathrm{Tr}\big((\nabla^2 g_h(Y_0))^\top (W(Z_0) - W(Z_1))\big)\big]\big|$$

$$= \big|\mathbb{E}\big[\mathrm{Tr}\big(W(Z_1)^\top (\nabla^2 g_h(Y_1) - \nabla^2 g_h(Y_0))\big)\big]\big|$$

$$\leq \mathbb{E}\big[\big|\mathrm{Tr}\big(W(Z_1)^\top (\nabla^2 g_h(Y_1) - \nabla^2 g_h(Y_0))\big)\big|\big]$$

$$\leq \mathbb{E}\big[\|W(Z_1)\|_F\,\|\nabla^2 g_h(Y_1) - \nabla^2 g_h(Y_0)\|_F\big]$$

$$\leq \mathbb{E}\big[(\sqrt{d}\,\|W(Z_1)\|_{op})\,(\sqrt{d}\,\|\nabla^2 g_h(Y_1) - \nabla^2 g_h(Y_0)\|_{op})\big]$$

$$= d\,\mathbb{E}\big[\|W(Z_1)\|_{op}\,\|\nabla^2 g_h(Y_1) - \nabla^2 g_h(Y_0)\|_{op}\big]$$

$$\leq d\,\mathbb{E}\big[\|W(Z_1)\|_{op}\,\frac{g_{3,Y}}{1-\beta}\|Y_1 - Y_0\|_2^\beta\big]$$

$$= d\,\frac{g_{3,Y}}{1-\beta}\alpha^{\frac{\beta}{2}}\,\mathbb{E}\big[\|W(Z_1)\|_{op}\,\|F(x^* + \sqrt{\alpha}Y_0) + \xi(Z_0)\|_2^\beta\big]$$

$$\leq d\,\frac{g_{3,Y}}{1-\beta}\alpha^{\frac{\beta}{2}}\,\big(\mathbb{E}[\|W(Z_1)\|_{op}^2]\big)^{1/2}\big(\mathbb{E}[\|F(x^* + \sqrt{\alpha}Y_0) + \xi(Z_0)\|_2^2]\big)^{\beta/2}$$

$$\leq d \frac{g_{3,Y}}{1-\beta} \alpha^{\frac{\beta}{2}} \sup_{z \in \mathsf{Z}} \|W(z)\|_{op} \big(\mathbb{E}[\|F(x^* + \sqrt{\alpha}Y_0) + \xi(Z_0)\|_2^2]\big)^{\beta/2}$$

$$\leq d \frac{g_{3,Y}}{1-\beta} \alpha^{\frac{\beta}{2}} \sup_{z \in \mathsf{Z}} \|W(z)\|_{op} \big(2\,\mathbb{E}[\|F(x^* + \sqrt{\alpha}Y_0)\|_2^2] + 2\,\mathbb{E}[\|\xi(Z_0)\|_2^2]\big)^{\beta/2}$$

$$\leq d \frac{g_{3,Y}}{1-\beta} \alpha^{\frac{\beta}{2}} \sup_{z \in \mathsf{Z}} \|W(z)\|_{op} \big(2L^2\alpha\,\mathbb{E}[\|Y_0\|_2^2] + 2\,\mathbb{E}[\|\xi(Z_0)\|_2^2]\big)^{\beta/2} \tag{85}$$

where all the above expectations are well defined as $\langle W(Z_0), \nabla^2 g_h(Y_0)\rangle \leq \|W\|\|\nabla^2 g_h\| < \infty$. Finally, we handle the rest of $\mathbb{E}[W_1 + W_2]$. We have

$$\left|\mathbb{E}\big[V(Z_0)^\top \nabla^2 g_h(Y_0)\,F(x^* + \sqrt{\alpha}\,Y_0)\big]\right| \leq \mathbb{E}\big[\|V(Z_0)\|\,\|\nabla^2 g_h(Y_0)\|_{op}\,\|F(x^* + \sqrt{\alpha}\,Y_0)\|\big]$$

$$\leq \|\nabla^2 g_h\|_\infty\,\mathbb{E}\big[\|V(Z_0)\|\,\|F(x^* + \sqrt{\alpha}\,Y_0)\|\big]$$

$$\leq \|\nabla^2 g_h\|_\infty\,\mathbb{E}\big[\|V(Z_0)\|\,L\sqrt{\alpha}\,\|Y_0\|\big]$$

$$= L\sqrt{\alpha}\,\|\nabla^2 g_h\|_\infty\,\mathbb{E}\big[\|V(Z_0)\|\,\|Y_0\|\big]$$

$$\leq L\sqrt{\alpha}\,\|\nabla^2 g_h\|_\infty\,\big(\mathbb{E}[\|V(Z_0)\|^2]\big)^{1/2}\big(\mathbb{E}[\|Y_0\|^2]\big)^{1/2} \tag{86}$$

Combining inequality (85), inequality (86), inequality (84), inequality (82), and inequality (79), we obtain the final bound.

$$d_W(Y^{(\alpha)}, Y) \leq \bar{U}\sqrt{\alpha}\log(1/\alpha)$$

$$\bar{U} := \left\{ g_{3,Y} \cdot 2\Big(1 + \mathbb{E}\big[(L\,\|Y_0\|)^3\big] + \mathbb{E}\big[\|\xi(Z_0)\|^3\big]\Big) + \frac{1}{2}\,\|D^2 F\|_\infty\,g_{1,Y}\,\mathbb{E}\big[\|Y_0\|^2\big]\right.$$

$$+ 2g_{3,Y}(\mathbb{E}\|V(Z_0)\|^2)^{1/2}\Big(L^2\alpha\,(\mathbb{E}\|Y_0\|^4)^{1/2} + (\mathbb{E}\|\xi(Z_0)\|^4)^{1/2} + 1\Big)$$

$$+ d\,g_{3,Y} \sup_{z \in \mathsf{Z}}\|W(z)\|_{op}\,\big(2L^2\alpha\,\mathbb{E}[\|Y_0\|_2^2] + 2\,\mathbb{E}[\|\xi(Z_0)\|_2^2]\big)^{1/2}$$

$$+ \left. L\sqrt{\alpha}\,g_{2,Y}\,\big(\mathbb{E}[\|V(Z_0)\|^2]\big)^{1/2}\big(\mathbb{E}[\|Y_0\|^2]\big)^{1/2}\right\} \tag{87}$$

where we use $\beta = 1 - \log(1/\alpha)$ to optimize the above bounds. All the constants are independent of $\alpha$ and are finite due to the assumptions in Lemma D.1. They are from the regularity of the Stein solution, the moments of $Y_0$ and the moments of $V(Z_0)$ and $\xi(Z_0)$.

## D.2. Proof of Theorem 4.3

With the general purpose lemma, D.1 in hand, proving 4.3 is exactly equivalent to verifying that the conditions of the Lemma hold for (1) SGD with strongly convex objective, (2) Linear SA, and (3) Contractive SA with Markovian noise. We do this below. Note that the following verification steps are highly similar to the i.i.d. setting. For completeness, we present the statements again, keeping the identical parts and only emphasizing the differences.

### D.2.1. CONSTANT-STEPSIZE SGD (MARKOVIAN NOISE)

Consider constant-stepsize SGD for minimizing a differentiable objective $f : \mathbb{R}^d \to \mathbb{R}$:

$$X_{k+1} = X_k - \alpha\big(\nabla f(X_k) + \xi(Z_k)\big), \qquad k \geq 0, \tag{88}$$

where $\{Z_k\}_{k \geq 0}$ is a Markov chain and the noise is of the form $\xi(Z_k)$ satisfying 4.2 Define $F(x) := -\nabla f(x)$, and $\xi(Z_k) = -\xi(\bar{Z}_k)$ so that (88) can be written in the form

$$X_{k+1} = X_k + \alpha\big(F(X_k) + \xi(Z_k)\big).$$

(i) Drift regularity. By Assumption 3.2, $f$ is $\sigma$-strongly convex and $L$-smooth, so $f$ has a unique minimizer $x^\star$ characterized by $\nabla f(x^\star) = 0$, hence $F(x^\star) = 0$ and this root is unique. Moreover, $L$-smoothness implies $\nabla f$ is globally $L$-Lipschitz, so for all $x, y$,

$$\|F(x) - F(y)\|_2 = \|\nabla f(y) - \nabla f(x)\|_2 \leq L\|x - y\|_2.$$

Finally, since $F = -\nabla f$, we have $DF(x) = -\nabla^2 f(x)$ and $D^2 F(x) = -\nabla^3 f(x)$. Thus Assumption C.6(2) holds by Assumption 3.4, as $\sup_x \|\nabla^3 f(x)\|_{\text{op}} < \infty$.

(ii) Noise regularity. Since the noise is of the form $\xi(Z_k)$ with the noise satisfying 4.2, the required regularity constraints follow.

(iii) Lyapunov equation and the target covariance. For SGD, $J^\star = DF(x^\star) = -\nabla^2 f(x^\star)$. By $\sigma$-strong convexity, $\nabla^2 f(x^\star) \succeq \sigma I$, hence the eigenvalues of $J^\star$ satisfy $\Re(\lambda(J^\star)) \leq -\sigma < 0$, i.e., $J^\star$ is Hurwitz. Consequently, the continuous Lyapunov equation (40) admits a unique symmetric solution $\Sigma_Y$, and if $\Sigma_M \succ 0$ then $\Sigma_Y \succ 0$.

(iv) Bounded Fourth Moment. The fourth moment of the SGD iterates being bounded follows by D.3.1.

By 4.2, the chain admits a stationary distribution. Then items (i)–(iv) verify all remaining hypotheses of Lemma D.1, so the result follows by a direct application of the lemma.

### D.2.2. LINEAR SA (MARKOVIAN NOISE)

Consider the linear SA recursion

$$X_{k+1} = X_k + \alpha\big(B(X_k - x^\star) + \xi(Z_k)\big), \qquad k \geq 0, \tag{89}$$

where $B \in \mathbb{R}^{d \times d}$ is fixed, $x^\star \in \mathbb{R}^d$ is fixed, and $\{Z_k\}_{k \geq 0}$ is a Markov chain with noise $\xi(Z_k)$ satisfying 4.2. Define the drift

$$F(x) := B(x - x^\star). \tag{90}$$

Then (89) is exactly of the form

$$X_{k+1} = X_k + \alpha\big(F(X_k) + \xi(Z_k)\big).$$

(i) Drift regularity. We have $F(x^\star) = 0$, and since $B$ is non-singular then this root is unique since $F(x) = 0 \iff B(x - x^\star) = 0 \iff x = x^\star$. Moreover, $F$ is globally Lipschitz with constant $L = \|B\|_{\text{op}}$:

$$\|F(x) - F(y)\|_2 = \|B(x - y)\|_2 \leq \|B\|_{\text{op}} \|x - y\|_2.$$

Finally, $F$ is linear, hence $F \in C^\infty$ and all third derivatives are identically zero, so Assumption C.6 holds.

(ii) Noise regularity. This holds by the Markov-noise regularity assumption 4.2 on $\xi(Z_k)$.

(iii) Lyapunov equation for the Gaussian limit (40). For the linear drift $F(x) = B(x - x^\star)$, we have $B^\star = DF(x^\star) = B$. If $B$ is Hurwitz, then the continuous Lyapunov equation

$$B\Sigma_Y + \Sigma_Y B^\top = -\Sigma_M$$

admits a unique symmetric solution $\Sigma_Y$, and if $\Sigma \succ 0$ then $\Sigma_Y \succ 0$.

(iv) Fourth moment bound. This follows from D.3.2

As items (i)–(iv) verify all hypotheses of Lemma C.7, we can apply the lemma to attain Theorem 3.7 as desired.

### D.2.3. CONTRACTIVE SA (MARKOVIAN NOISE)

Consider the SA recursion driven by a contractive operator $\mathcal{T} : \mathbb{R}^d \to \mathbb{R}^d$:

$$X_{k+1} = X_k + \alpha\big(\mathcal{T}(X_k) - X_k + \xi(Z_k)\big), \qquad k \geq 0, \tag{91}$$

where $\{Z_k\}_{k \geq 0}$ is a Markov chain and the noise satisfies 4.2. Define the drift

$$F(x) := \mathcal{T}(x) - x. \tag{92}$$

Then (91) is exactly:

$$X_{k+1} = X_k + \alpha\big(F(X_k) + \xi(Z_k)\big).$$

(i) Drift regularity. By Banach's fixed-point theorem, Assumption 3.8 yields a unique fixed point $x^\star$ with $\mathcal{T}(x^\star) = x^\star$, hence $F(x^\star) = 0$ for $F(x) := \mathcal{T}(x) - x$. Moreover, the $\|\cdot\|_\mu$-contraction implies, for all $x, y$,

$$\|F(x) - F(y)\|_\mu \leq \|\mathcal{T}(x) - \mathcal{T}(y)\|_\mu + \|x - y\|_\mu \leq (1 + \gamma)\|x - y\|_\mu,$$

so $L_\mu := 1 + \gamma$. Using $\sqrt{\mu_{\min}}\|v\|_2 \leq \|v\|_\mu \leq \sqrt{\mu_{\max}}\|v\|_2$, we also have

$$\|F(x) - F(y)\|_2 \leq L\,\|x - y\|_2, \qquad L := (1 + \gamma)\sqrt{\mu_{\max}/\mu_{\min}}.$$

Finally, since $\mathcal{T} \in C^3$ with bounded second derivatives (Assumption 3.10), the same holds for $F$.

(ii) Noise regularity. This follows from Assumption 4.2 on $\xi(Z_k)$.

(iii) Lyapunov equation. We have $J^\star = DF(x^\star) = D\mathcal{T}(x^\star) - I$. By Lemma 3.9, $J^\star$ is Hurwitz, hence the Lyapunov equation $J^\star \Sigma_Y + \Sigma_Y(J^\star)^\top = -\Sigma$ admits a unique symmetric solution $\Sigma_Y$ (and $\Sigma_Y \succ 0$ if $\Sigma \succ 0$).

(iv) Fourth moment bound. This follows from D.3.3

As items (i)–(iv) verify all hypotheses of Lemma C.7, we attain the Wasserstein bound for $Y^{(\alpha)} = (X^{(\alpha)} - x^\star)/\sqrt{\alpha}$ for the contractive recursion (91).

Having verified all the conditions needed for Lemma D.1, we now directly applly the lemma and yield the final results. Recall the bound in Lemma as follows.

$$d_W\big(\mathcal{L}(Y^{(\alpha)}), \mathcal{L}(Y)\big) \leq \bar{U}\,\sqrt{\alpha}\log\big(1/\alpha\big),$$

with constant $\bar{U}$ defined in (87). The result follows as we plug in the moment bounds, and different constants $L$, defined for different models in above, into (87). We further achieve tail bounds by applying this Wasserstein bound along with Lemma C.9. We omit the details but directly refer to the proof of the i.i.d. case for more clarity. We show the final tail bounds below for completeness.

$$|\mathbb{P}(\langle Y^{(\alpha)}, \zeta\rangle > a) - \mathbb{P}(Z_\zeta > a)| \leq U_4' \frac{\alpha^{1/4}\log^{1/2}(1/\alpha)}{a},$$

$$\text{where } U_4' := (8\sqrt{\|\Sigma_Y\|_{op}} + 2) \cdot \bar{U}^{1/2}. \tag{93}$$

## D.3. Deriving Moment bounds for Markovian Noise

### D.3.1. BOUNDEDNESS OF THE FOURTH MOMENT IN SGD

The following lemma is a direct application in strong convexity and $L$-smoothness. This lemma establishes that the SGD with Markov noise satisfies Assumptions 2.1 and 2.2 of (Yu et al., 2020).

**Lemma D.2.** *Let $f : \mathbb{R}^d \to \mathbb{R}$ be twice differentiable and satisfy Assumption 3.2 (i.e. $f$ is $L$-smooth and $\sigma$-strongly convex). Then $f$ satisfies:*

1. *(Linear growth) There exists a constant $C_L > 0$ such that*

$$\|\nabla f(x)\| \leq C_L\big(1 + \|x\|\big), \qquad \forall\, x \in \mathbb{R}^d.$$

2. *(Dissipativity) There exist constants $\mu > 0$ and $\gamma \geq 0$ such that*

$$\langle x, \nabla f(x)\rangle \geq \mu\|x\|^2 - \gamma, \qquad \forall\, x \in \mathbb{R}^d.$$

*Proof.* By L-smoothness, we have

$$\|\nabla f(x)\| = \|\nabla f(x) - \nabla f(x^*)\| \leq L(\|x\| + \|x^*\|) \leq C_L(1 + \|x\|) \tag{94}$$

verifying the first part. For the second part, note that strong convexity implies strong monotonicity of the gradient:

$$\langle \nabla f(x) - \nabla f(x^\star), x - x^\star\rangle \geq \sigma\|x - x^\star\|^2,$$

hence $\langle \nabla f(x), x - x^\star \rangle \geq \sigma \|x - x^\star\|^2$. Writing

$$\langle x, \nabla f(x) \rangle = \langle x - x^\star, \nabla f(x) \rangle + \langle x^\star, \nabla f(x) \rangle$$
$$\geq \sigma \|x - x^\star\|^2 - \|x^\star\| \, \|\nabla f(x)\| \langle x, \nabla f(x) \rangle$$
$$\geq \sigma \|x - x^\star\|^2 - L\|x^\star\| \, \|x - x^\star\|.$$

by $L$-smoothness. Applying Young's inequality $ab \leq \frac{\sigma}{2}a^2 + \frac{1}{2\sigma}b^2$ with $a = \|x - x^\star\|$ and $b = L\|x^\star\|$ yields

$$\langle x, \nabla f(x) \rangle \geq \frac{\sigma}{2}\|x - x^\star\|^2 - \frac{L^2}{2\sigma}\|x^\star\|^2.$$

Finally, since $\|x - x^\star\|^2 = \|x\|^2 + \|x^\star\|^2 - 2\langle x, x^\star \rangle \geq \frac{1}{2}\|x\|^2 - \|x^\star\|^2$, we conclude

$$\langle x, \nabla f(x) \rangle \geq \frac{\sigma}{4}\|x\|^2 - \left(\frac{\sigma}{2} + \frac{L^2}{2\sigma}\right)\|x^\star\|^2.$$

as desired. $\qquad\square$

Additionally, we will need the following two simple lemmas on relating powers of the $l_2$-norm.

**Lemma D.3.** *For any $\alpha > 0$ and any $\varepsilon > 0$, for all $x \in \mathbb{R}^d$,*

$$\alpha\|x\|^3 \leq \alpha^{1+\varepsilon}\|x\|^4 + \alpha^{1-3\varepsilon}.$$

*Proof.* Fix $x \in \mathbb{R}^d$ and set $r := \|x\| \geq 0$. We consider two cases. For the first case, suppose $r \geq \alpha^{-\varepsilon}$. Then $\alpha^\varepsilon r \geq 1$, so

$$\alpha r^3 \leq \alpha r^3(\alpha^\varepsilon r) = \alpha^{1+\varepsilon} r^4 = \alpha^{1+\varepsilon}\|x\|^4.$$

Therefore $\alpha\|x\|^3 \leq \alpha^{1+\varepsilon}\|x\|^4 \leq \alpha^{1+\varepsilon}\|x\|^4 + \alpha^{1-3\varepsilon}$. In the second case, we have $r < \alpha^{-\varepsilon}$. Then $r^3 < \alpha^{-3\varepsilon}$, hence

$$\alpha r^3 < \alpha \cdot \alpha^{-3\varepsilon} = \alpha^{1-3\varepsilon}.$$

Thus $\alpha\|x\|^3 \leq \alpha^{1-3\varepsilon} \leq \alpha^{1+\varepsilon}\|x\|^4 + \alpha^{1-3\varepsilon}$. Combining the two cases yields the claim. $\qquad\square$

**Lemma D.4.** *For any $\alpha > 0$ and any $\varepsilon > 0$, for all $x \in \mathbb{R}^d$,*

$$\alpha\|x\|^2 \leq \alpha^{1+\varepsilon}\|x\|^4 + \alpha^{1-\varepsilon}.$$

*Proof.* Fix $x \in \mathbb{R}^d$ and set $r := \|x\| \geq 0$. If $r \geq \alpha^{-\varepsilon/2}$, then $\alpha^\varepsilon r^2 \geq 1$, so

$$\alpha r^2 \leq \alpha r^2(\alpha^\varepsilon r^2) = \alpha^{1+\varepsilon} r^4 = \alpha^{1+\varepsilon}\|x\|^4.$$

Therefore $\alpha\|x\|^2 \leq \alpha^{1+\varepsilon}\|x\|^4 \leq \alpha^{1+\varepsilon}\|x\|^4 + \alpha^{1-\varepsilon}$. On the other hand, if $r < \alpha^{-\varepsilon/2}$, then $r^2 < \alpha^{-\varepsilon}$, hence

$$\alpha r^2 < \alpha \cdot \alpha^{-\varepsilon} = \alpha^{1-\varepsilon}.$$

Thus $\alpha\|x\|^2 \leq \alpha^{1-\varepsilon} \leq \alpha^{1+\varepsilon}\|x\|^4 + \alpha^{1-\varepsilon}$ and the result follows. $\qquad\square$

We will now proceed to the proof of bounded fourth moment. We will establish a Lyapunov drift inequality for the quartic Lyapunov function $L_4(x) := 1 + \|x\|^4$, and then apply it at stationarity in order to attain boundedness. Throughout this subsection, we write $X_k := X_k^{(\alpha)}$ for brevity.

**Lemma D.5** (Fourth-moment Lyapunov drift). *Suppose $f$ satisfies Assumption 3.2, and the noise satisfies Assumption 4.2. Consider the SGD recursion*

$$X_{k+1} = X_k - \alpha\big(\nabla f(X_k) + \xi(Z_k)\big), \qquad k \geq 0.$$

*There exist constants $\alpha_4^\dagger > 0$, $\kappa_4 \in (0,1)$ and $\beta_4 < \infty$ such that for all $\alpha \in (0, \alpha_4^\dagger)$,*

$$\mathbb{E}\big[1 + \|X_{k+1}\|^4 \,\big|\, \mathcal{F}_k\big] \leq \kappa_4\big(1 + \|X_k\|^4\big) + \beta_4, \qquad k \geq 0. \tag{95}$$

*Proof.* Define
$$G_k \; := \; \nabla f(X_k) + \xi(Z_k),$$
so that the recursion may be written
$$X_{k+1} = X_k - \alpha G_k.$$
We first expand $\|X_{k+1}\|^4$ in terms of $X_k$ and $G_k$. Observe that
$$\|X_{k+1}\|^2 = \|X_k - \alpha G_k\|^2 = \|X_k\|^2 - 2\alpha\langle X_k, G_k\rangle + \alpha^2\|G_k\|^2. \tag{96}$$
Squaring (96) yields
$$\begin{aligned}
\|X_{k+1}\|^4 &= \Big(\|X_k\|^2 - 2\alpha\langle X_k, G_k\rangle + \alpha^2\|G_k\|^2\Big)^2 \\
&= \|X_k\|^4 - 4\alpha\|X_k\|^2\langle X_k, G_k\rangle + 4\alpha^2\langle X_k, G_k\rangle^2 \\
&\quad + 2\alpha^2\|X_k\|^2\|G_k\|^2 - 4\alpha^3\langle X_k, G_k\rangle\|G_k\|^2 + \alpha^4\|G_k\|^4.
\end{aligned} \tag{97}$$
Taking conditional expectations, we obtain
$$\mathbb{E}\big[\|X_{k+1}\|^4 \mid \mathcal{F}_k\big] = T_0 + T_1 + T_2 + T_3 + T_4 + T_5, \tag{98}$$
where
$$\begin{aligned}
T_0 &= \|X_k\|^4, \\
T_1 &= -4\alpha\|X_k\|^2\,\mathbb{E}[\langle X_k, G_k\rangle \mid \mathcal{F}_k], \\
T_2 &= 4\alpha^2\,\mathbb{E}\big[\langle X_k, G_k\rangle^2 \mid \mathcal{F}_k\big], \\
T_3 &= 2\alpha^2\|X_k\|^2\,\mathbb{E}\big[\|G_k\|^2 \mid \mathcal{F}_k\big], \\
T_4 &= -4\alpha^3\,\mathbb{E}\big[\langle X_k, G_k\rangle\|G_k\|^2 \mid \mathcal{F}_k\big], \\
T_5 &= \alpha^4\,\mathbb{E}\big[\|G_k\|^4 \mid \mathcal{F}_k\big].
\end{aligned}$$

**Controlling $T_1$ via dissipativity.** We first control $T_1$. Plugging in $G_k = \nabla f(X_k) + \xi(Z_k)$ gives
$$T_1 = -4\alpha\|X_k\|^2\langle X_k, \nabla f(X_k)\rangle \; - \; 4\alpha\|X_k\|^2\mathbb{E}[\langle X_k, \xi(Z_k)\rangle \mid \mathcal{F}_k]. \tag{99}$$
By dissipativity (Lemma D.2(2)),
$$-4\alpha\|X_k\|^2\langle X_k, \nabla f(X_k)\rangle \leq -4\alpha\mu\|X_k\|^4 + 4\alpha\gamma\|X_k\|^2.$$
For the noise term, by Cauchy–Schwarz,
$$\Big|\mathbb{E}[\langle X_k, \xi(Z_k)\rangle \mid \mathcal{F}_k]\Big| \leq \|X_k\|\,\mathbb{E}[\|\xi(Z_k)\| \mid \mathcal{F}_k].$$
Hence
$$-4\alpha\|X_k\|^2\mathbb{E}[\langle X_k, \xi(Z_k)\rangle \mid \mathcal{F}_k] \leq 4\alpha\|X_k\|^3\,\mathbb{E}[\|\xi(Z_k)\| \mid \mathcal{F}_k]. \tag{100}$$
By 3.1, there exists a constant $C_{\xi,1} < \infty$ such that
$$\mathbb{E}[\|\xi(Z_k)\| \mid \mathcal{F}_k] \leq C_{\xi,1}\big(1 + \|X_k\|\big).$$
Substituting into (100) yields
$$4\alpha\|X_k\|^3\,\mathbb{E}[\|\xi(Z_k)\| \mid \mathcal{F}_k] \leq 4C_{\xi,1}\alpha\big(\|X_k\|^3 + \|X_k\|^4\big).$$
Now apply Lemmas D.3 and D.4 with some fixed $\varepsilon \in (0, 1/3)$:
$$\alpha\|X_k\|^3 \leq \alpha^{1+\varepsilon}\|X_k\|^4 + \alpha^{1-3\varepsilon}, \qquad \alpha\|X_k\|^2 \leq \alpha^{1+\varepsilon}\|X_k\|^4 + \alpha^{1-\varepsilon}.$$
Combining the above displays with (99), we obtain the bound
$$T_1 \leq -4\alpha\mu\|X_k\|^4 + C_1\alpha^{1+\varepsilon}\|X_k\|^4 + C_1\big(\alpha^{1-3\varepsilon} + \alpha^{1-\varepsilon} + \alpha\big), \tag{101}$$
for some finite constant $C_1 < \infty$ depending only on $\mu, \gamma, C_{\xi,1}$ and $\varepsilon$.

**Bounding the higher-order terms.** We now bound $T_2, \ldots, T_5$ using the $L$-smoothness of $f$ and the finite second and fourth moments of $\xi(Z_k)$. First, by Lemma D.2(1) there exists $C_\nabla < \infty$ such that

$$\|\nabla f(X_k)\| \le C_\nabla (1 + \|X_k\|).$$

By the noise moment assumptions, there exist constants $C_{\xi,2}, C_{\xi,4} < \infty$ such that

$$\mathbb{E}\big[\|\xi(Z_k)\|^2 \mid \mathcal{F}_k\big] \le C_{\xi,2}\big(1 + \|X_k\|^2\big), \qquad \mathbb{E}\big[\|\xi(Z_k)\|^4 \mid \mathcal{F}_k\big] \le C_{\xi,4}\big(1 + \|X_k\|^4\big).$$

Consequently, there exists $C_G < \infty$ such that

$$\mathbb{E}\big[\|G_k\|^2 \mid \mathcal{F}_k\big] \le C_G\big(1 + \|X_k\|^2\big), \qquad \mathbb{E}\big[\|G_k\|^4 \mid \mathcal{F}_k\big] \le C_G\big(1 + \|X_k\|^4\big). \tag{102}$$

Moreover, by Cauchy–Schwarz,

$$\mathbb{E}\big[\|G_k\|^3 \mid \mathcal{F}_k\big] \le \Big(\mathbb{E}[\|G_k\|^4 \mid \mathcal{F}_k]\Big)^{3/4} \le C_G\big(1 + \|X_k\|^3\big) \le C_G\big(1 + \|X_k\|^4\big),$$

where in the last step we used $\|x\|^3 \le 1 + \|x\|^4$.

Using Cauchy–Schwarz and (102), we obtain

$$T_2 = 4\alpha^2 \mathbb{E}\big[\langle X_k, G_k\rangle^2 \mid \mathcal{F}_k\big] \le 4\alpha^2 \|X_k\|^2 \, \mathbb{E}\big[\|G_k\|^2 \mid \mathcal{F}_k\big] \le C_2\alpha^2\big(1 + \|X_k\|^4\big),$$

$$T_3 = 2\alpha^2 \|X_k\|^2 \, \mathbb{E}\big[\|G_k\|^2 \mid \mathcal{F}_k\big] \le C_3\alpha^2\big(1 + \|X_k\|^4\big),$$

$$T_4 = -4\alpha^3 \, \mathbb{E}\big[\langle X_k, G_k\rangle\|G_k\|^2 \mid \mathcal{F}_k\big] \le 4\alpha^3 \|X_k\| \, \mathbb{E}\big[\|G_k\|^3 \mid \mathcal{F}_k\big] \le C_4\alpha^3\big(1 + \|X_k\|^4\big),$$

$$T_5 = \alpha^4 \, \mathbb{E}\big[\|G_k\|^4 \mid \mathcal{F}_k\big] \le C_5\alpha^4\big(1 + \|X_k\|^4\big),$$

for some finite constants $C_2, \ldots, C_5$. Therefore, for $\alpha \in (0, 1]$,

$$T_2 + T_3 + T_4 + T_5 \le C_{hi}\,\alpha^2\big(1 + \|X_k\|^4\big) \tag{103}$$

for some $C_{hi} < \infty$ (absorbing the $\alpha^3$ and $\alpha^4$ terms into $\alpha^2$).

**Drift inequality.** Substituting (101) and (103) into (98), and using $T_0 = \|X_k\|^4$, yields

$$\mathbb{E}\big[\|X_{k+1}\|^4 \mid \mathcal{F}_k\big] \le \|X_k\|^4 - 4\alpha\mu\|X_k\|^4 + C_1\alpha^{1+\varepsilon}\|X_k\|^4 + C_{hi}\alpha^2\big(1 + \|X_k\|^4\big)$$
$$+ C_1\big(\alpha^{1-3\varepsilon} + \alpha^{1-\varepsilon} + \alpha\big).$$

Adding 1 to both sides and recalling $L_4(x) := 1 + \|x\|^4$, we obtain

$$\mathbb{E}[L_4(X_{k+1}) \mid \mathcal{F}_k] \le \Big(1 - 4\mu\alpha + C_1\alpha^{1+\varepsilon} + C_{hi}\alpha^2\Big) L_4(X_k) + C\big(\alpha^{1-3\varepsilon} + \alpha^{1-\varepsilon} + \alpha + \alpha^2\big),$$

for a finite constant $C < \infty$.

Finally, choose $\alpha_4^\dagger \in (0, 1]$ sufficiently small so that for all $\alpha \in (0, \alpha_4^\dagger)$,

$$1 - 4\mu\alpha + C_1\alpha^{1+\varepsilon} + C_{hi}\alpha^2 \le: \kappa_4 < 1,$$

and define

$$\beta_4 := C\big(\alpha^{1-3\varepsilon} + \alpha^{1-\varepsilon} + \alpha + \alpha^2\big).$$

This yields (95) and completes the proof. $\square$

**Corollary D.6** (Bounded fourth moment). *Under Assumptions 3.2 and 4.2, let $\alpha \in (0, \alpha^\dagger)$ and let $\pi_\alpha$ denote the stationary distribution of the Markov chain $\{X_k^{(\alpha)}\}_{k\ge 0}$ generated by the iterates of SGD under Markovian noise. Then*

$$\int \|x\|^4\, \pi_\alpha(dx) < \infty.$$

*Proof.* Start the chain in stationarity: $(X_0, Z_0) \sim \pi_\alpha$. Taking expectations in (95) under $\pi_\alpha$ and using stationarity yields

$$\mathbb{E}_{\pi_\alpha}[L_4(X_1)] = \mathbb{E}_{\pi_\alpha}[L_4(X_0)] \leq \kappa_4 \mathbb{E}_{\pi_\alpha}[L_4(X_0)] + \beta_4.$$

Rearranging,

$$(1 - \kappa_4)\, \mathbb{E}_{\pi_\alpha}[L_4(X_0)] \leq \beta_4, \qquad \text{so} \qquad \mathbb{E}_{\pi_\alpha}[L_4(X_0)] \leq \frac{\beta_4}{1 - \kappa_4} < \infty.$$

Since $L_4(x) = 1 + \|x\|^4$, this gives $\mathbb{E}_{\pi_\alpha}\|X_0\|^4 < \infty$. Finally, the bound $\|x\|^2 \leq 1 + \|x\|^4$ implies that $\mathbb{E}_{\pi_\alpha}\|X_0\|^2 < \infty$ as well. $\qquad\square$

### D.3.2. BOUNDEDNESS OF THE FOURTH MOMENT IN LINEAR SA

We now rewrite the SGD fourth-moment drift proof in the setting of *linear* stochastic approximation with Markovian noise. The only substantive change is that we work with a weighted norm induced by a Lyapunov matrix. Recall the Linear SA recursion given by

$$X_{k+1} = X_k + \alpha\Big(J(X_k - x^\star) + \xi(Z_k)\Big), \qquad k \geq 0, \tag{104}$$

where $\{Z_k\}$ is a Markov chain and $\xi(Z_k)$ is the noise. We will work with the centered error $E_k := X_k - x^\star$, so that

$$E_{k+1} = E_k + \alpha\Big(JE_k + \xi(Z_k)\Big). \tag{105}$$

By 3.6, $J$ is Hurwitz. There exists a symmetric positive definite matrix $P \succ 0$ and a constant $\mu_P > 0$ such that

$$J^\top P + PJ \preceq -\mu_P P. \tag{106}$$

Define the weighted norm $\|e\|_P^2 := e^\top P e$ and the quartic Lyapunov function

$$L_{4,P}(e) := 1 + \|e\|_P^4.$$

Let $\lambda_{\min}(P)$ and $\lambda_{\max}(P)$ denote the extreme eigenvalues of $P$. Then for all $v \in \mathbb{R}^d$,

$$\lambda_{\min}(P)\|v\|^2 \leq \|v\|_P^2 \leq \lambda_{\max}(P)\|v\|^2. \tag{107}$$

Consequently, because $\xi(Z_k)$ is uniformly bounded along the chain (4.2), we have the following $P$-norm bounds: there exist constants $C_{\xi,1}^P, C_{\xi,2}^P, C_{\xi,4}^P < \infty$ such that

$$\mathbb{E}[\|\xi(Z_k)\|_P \mid \mathcal{F}_k] \leq C_{\xi,1}^P\big(1 + \|E_k\|_P\big), \tag{108}$$

$$\mathbb{E}[\|\xi(Z_k)\|_P^2 \mid \mathcal{F}_k] \leq C_{\xi,2}^P\big(1 + \|E_k\|_P^2\big), \tag{109}$$

$$\mathbb{E}[\|\xi(Z_k)\|_P^4 \mid \mathcal{F}_k] \leq C_{\xi,4}^P\big(1 + \|E_k\|_P^4\big). \tag{110}$$

We will now prove the boundedness of fourth moment via the contraction of an appropriate Lyapunov funciton.

**Lemma D.7** (Fourth-moment Lyapunov drift for Hurwitz linear SA). *Consider* (105). *Suppose $J$ is Hurwitz and let $P \succ 0$ satisfy* (106). *Assume 4.2. Then there exist constants $\alpha_4^\dagger > 0$, $\kappa_4 \in (0,1)$ and $\beta_4 < \infty$ such that for all $\alpha \in (0, \alpha_4^\dagger)$,*

$$\mathbb{E}\big[1 + \|E_{k+1}\|_P^4 \,\big|\, \mathcal{F}_k\big] \leq \kappa_4\big(1 + \|E_k\|_P^4\big) + \beta_4, \qquad k \geq 0. \tag{111}$$

*Proof.* Define

$$G_k := JE_k + \xi(Z_k),$$

so that $E_{k+1} = E_k + \alpha G_k$.

We have

$$\|E_{k+1}\|_P^2 = \|E_k + \alpha G_k\|_P^2 = \|E_k\|_P^2 + 2\alpha\langle E_k, PG_k\rangle + \alpha^2\|G_k\|_P^2. \tag{112}$$

Squaring (112) gives

$$\|E_{k+1}\|_P^4 = \|E_k\|_P^4 + 4\alpha\|E_k\|_P^2\langle E_k, PG_k\rangle + 4\alpha^2\langle E_k, PG_k\rangle^2$$
$$+ 2\alpha^2\|E_k\|_P^2\|G_k\|_P^2 + 4\alpha^3\langle E_k, PG_k\rangle\|G_k\|_P^2 + \alpha^4\|G_k\|_P^4. \tag{113}$$

Taking conditional expectation and splitting terms as before,

$$\mathbb{E}[\|E_{k+1}\|_P^4 \mid \mathcal{F}_k] = T_0 + T_1 + T_2 + T_3 + T_4 + T_5, \tag{114}$$

where

$$T_0 = \|E_k\|_P^4,$$
$$T_1 = 4\alpha\|E_k\|_P^2 \mathbb{E}[\langle E_k, PG_k\rangle \mid \mathcal{F}_k],$$
$$T_2 = 4\alpha^2 \mathbb{E}[\langle E_k, PG_k\rangle^2 \mid \mathcal{F}_k],$$
$$T_3 = 2\alpha^2\|E_k\|_P^2 \mathbb{E}[\|G_k\|_P^2 \mid \mathcal{F}_k],$$
$$T_4 = 4\alpha^3 \mathbb{E}[\langle E_k, PG_k\rangle\|G_k\|_P^2 \mid \mathcal{F}_k],$$
$$T_5 = \alpha^4 \mathbb{E}[\|G_k\|_P^4 \mid \mathcal{F}_k].$$

**Controlling $T_1$ via Lyapunov stability.** Using $G_k = JE_k + \xi(Z_k)$,

$$T_1 = 4\alpha\|E_k\|_P^2\langle E_k, PJE_k\rangle + 4\alpha\|E_k\|_P^2 \mathbb{E}[\langle E_k, P\xi(Z_k)\rangle \mid \mathcal{F}_k]. \tag{115}$$

By (106), for any $e \in \mathbb{R}^d$,

$$2\langle e, PJe\rangle = e^\top(PJ + J^\top P)e \leq -\mu_P\, e^\top Pe = -\mu_P\|e\|_P^2,$$

hence

$$\langle e, PJe\rangle \leq -\frac{\mu_P}{2}\|e\|_P^2. \tag{116}$$

Therefore,

$$4\alpha\|E_k\|_P^2\langle E_k, PJE_k\rangle \leq -2\mu_P\alpha\|E_k\|_P^4.$$

For the noise contribution, by Cauchy–Schwarz in the $P$–inner product,

$$\left|\langle E_k, P\xi(Z_k)\rangle\right| \leq \|E_k\|_P \|\xi(Z_k)\|_P.$$

Thus, using (108),

$$\left|\mathbb{E}[\langle E_k, P\xi(Z_k)\rangle \mid \mathcal{F}_k]\right| \leq \|E_k\|_P \mathbb{E}[\|\xi(Z_k)\|_P \mid \mathcal{F}_k] \leq C_{\xi,1}^P\big(\|E_k\|_P + \|E_k\|_P^2\big).$$

Hence

$$4\alpha\|E_k\|_P^2 \mathbb{E}[\langle E_k, P\xi(Z_k)\rangle \mid \mathcal{F}_k] \leq 4C_{\xi,1}^P\alpha\big(\|E_k\|_P^3 + \|E_k\|_P^4\big).$$

Applying Lemmas D.3 and D.4 (with any fixed $\varepsilon \in (0, 1/3)$, and with $\|\cdot\|$ replaced by $\|\cdot\|_P$) yields

$$T_1 \leq -2\mu_P\alpha\|E_k\|_P^4 + C_1\alpha^{1+\varepsilon}\|E_k\|_P^4 + C_1\big(\alpha^{1-3\varepsilon} + \alpha^{1-\varepsilon} + \alpha\big), \tag{117}$$

for some finite $C_1 < \infty$ depending only on $\mu_P, C_{\xi,1}^P, \varepsilon$.

**Bounding the higher-order terms** Since $\|JE_k\|_P \leq L_P\|E_k\|_P$ where $L_P := \|P^{1/2}JP^{-1/2}\|_{\mathrm{op}}$, and using (109)–(110) together with $\|a + b\|_P^p \leq 2^{p-1}(\|a\|_P^p + \|b\|_P^p)$ for $p \in \{2, 4\}$, there exists $C_G < \infty$ such that

$$\mathbb{E}[\|G_k\|_P^2 \mid \mathcal{F}_k] \leq C_G(1 + \|E_k\|_P^2), \qquad \mathbb{E}[\|G_k\|_P^4 \mid \mathcal{F}_k] \leq C_G(1 + \|E_k\|_P^4). \tag{118}$$

Moreover, by Cauchy–Schwarz,

$$\mathbb{E}[\|G_k\|_P^3 \mid \mathcal{F}_k] \leq \big(\mathbb{E}[\|G_k\|_P^4 \mid \mathcal{F}_k]\big)^{3/4} \leq C_G(1 + \|E_k\|_P^4).$$

Using $\langle E_k, PG_k \rangle \leq \|E_k\|_P \|G_k\|_P$ and Cauchy–Schwarz as in the SGD proof,

$$T_2 \leq 4\alpha^2 \|E_k\|_P^2 \, \mathbb{E}[\|G_k\|_P^2 \mid \mathcal{F}_k] \leq C_2 \alpha^2 (1 + \|E_k\|_P^4),$$
$$T_3 \leq 2\alpha^2 \|E_k\|_P^2 \, \mathbb{E}[\|G_k\|_P^2 \mid \mathcal{F}_k] \leq C_3 \alpha^2 (1 + \|E_k\|_P^4),$$
$$T_4 \leq 4\alpha^3 \|E_k\|_P \, \mathbb{E}[\|G_k\|_P^3 \mid \mathcal{F}_k] \leq C_4 \alpha^3 (1 + \|E_k\|_P^4),$$
$$T_5 \leq \alpha^4 \, \mathbb{E}[\|G_k\|_P^4 \mid \mathcal{F}_k] \leq C_5 \alpha^4 (1 + \|E_k\|_P^4),$$

for finite constants $C_2, \ldots, C_5$. Hence for $\alpha \in (0, 1]$,

$$T_2 + T_3 + T_4 + T_5 \leq C_{hi} \alpha^2 (1 + \|E_k\|_P^4), \tag{119}$$

absorbing $\alpha^3, \alpha^4$ into $\alpha^2$.

**Drift inequality.** Substituting (117) and (119) into (114) (and using $T_0 = \|E_k\|_P^4$) gives

$$\mathbb{E}[\|E_{k+1}\|_P^4 \mid \mathcal{F}_k] \leq \left(1 - 2\mu_P \alpha + C_1 \alpha^{1+\varepsilon} + C_{hi} \alpha^2\right) \|E_k\|_P^4 + C\left(\alpha^{1-3\varepsilon} + \alpha^{1-\varepsilon} + \alpha + \alpha^2\right),$$

for a finite $C < \infty$. Adding 1 and recalling $L_{4,P}(e) := 1 + \|e\|_P^4$ yields

$$\mathbb{E}[L_{4,P}(E_{k+1}) \mid \mathcal{F}_k] \leq \left(1 - 2\mu_P \alpha + C_1 \alpha^{1+\varepsilon} + C_{hi} \alpha^2\right) L_{4,P}(E_k) + C\left(\alpha^{1-3\varepsilon} + \alpha^{1-\varepsilon} + \alpha + \alpha^2\right).$$

Choose $\alpha_4^\dagger \in (0, 1]$ small enough that for all $\alpha \in (0, \alpha_4^\dagger)$,

$$1 - 2\mu_P \alpha + C_1 \alpha^{1+\varepsilon} + C_{hi} \alpha^2 \leq: \kappa_4 < 1,$$

and set

$$\beta_4 := C\left(\alpha^{1-3\varepsilon} + \alpha^{1-\varepsilon} + \alpha + \alpha^2\right).$$

This proves (111). □

**Corollary D.8** (Bounded fourth moment for Hurwitz linear SA). *Assume $J$ is Hurwitz and let $P \succ 0$ satisfy* (106). *If the Markov chain $\{(E_k, Z_k)\}$ induced by* (105) *admits a stationary distribution $\pi_\alpha$, then for any $\alpha \in (0, \alpha_4^\dagger)$,*

$$\int \|e\|^4 \, \pi_\alpha(de, dz) < \infty.$$

*Proof.* Start in stationarity: $(E_0, Z_0) \sim \pi_\alpha$. Taking expectations in (111) under $\pi_\alpha$ and using stationarity gives

$$\mathbb{E}_{\pi_\alpha}[L_{4,P}(E_1)] = \mathbb{E}_{\pi_\alpha}[L_{4,P}(E_0)] \leq \kappa_4 \mathbb{E}_{\pi_\alpha}[L_{4,P}(E_0)] + \beta_4.$$

Rearranging,

$$(1 - \kappa_4) \mathbb{E}_{\pi_\alpha}[L_{4,P}(E_0)] \leq \beta_4, \qquad \text{so} \qquad \mathbb{E}_{\pi_\alpha} \|E_0\|_P^4 < \infty.$$

Finally, by norm equivalence (107), $\|e\|^4 \leq \lambda_{\min}(P)^{-2} \|e\|_P^4$, hence $\mathbb{E}_{\pi_\alpha} \|E_0\|^4 < \infty$. Since $X_0 = E_0 + x^\star$ and $\|e + x^\star\|^4 \leq 8(\|e\|^4 + \|x^\star\|^4)$, we also have $\mathbb{E}_{\pi_\alpha} \|X_0\|^4 < \infty$. □

### D.3.3. BOUNDEDNESS OF THE FOURTH MOMENT IN CONTRACTIVE SA

We now establish a fourth-moment Lyapunov drift inequality for stochastic approximation driven by a *contractive* operator in a weighted norm. This is the nonlinear analogue of the Hurwitz linear-SA argument: contraction in the $\| \cdot \|_\mu$ norm plays the role of dissipativity, and we control a quartic Lyapunov function in the same norm.

Recall the drift term can be written as a residual of a nonlinear vector field, namely $F(x) = \mathcal{T}(x) - x$ with $\mathcal{T} : \mathbb{R}^d \to \mathbb{R}^d$. Thus, the recursion

$$X_{k+1} = X_k + \alpha\big(F(X_k) + \xi(Z_k)\big) = X_k + \alpha\Big(\mathcal{T}(X_k) - X_k + \xi(Z_k)\Big), \qquad k \geq 0, \tag{120}$$

Recall $x^\star$ is the unique fixed point of $\mathcal{T}$, $\mathcal{T}(x^\star) = x^\star$, and define the centered error $E_k := X_k - x^\star$. The, we have

$$E_{k+1} = E_k + \alpha\big(F(X_k) + \xi(Z_k)\big). \tag{121}$$

Furthermore, we rewrite the contractive assumption by letting $M := \mathrm{diag}(\mu_1, \ldots, \mu_d)$, so that $\|x\|_\mu^2 = x^\top M x$. By the contractivity assumption (3.8) $x^\star$ is unique and, moreover, for all $x \in \mathbb{R}^d$,

$$\langle x - x^\star, MF(x) \rangle = \langle x - x^\star, M(\mathcal{T}(x) - x) \rangle \leq -(1-\gamma)\|x - x^\star\|_\mu^2, \tag{122}$$

and

$$\|F(x)\|_\mu = \|\mathcal{T}(x) - x\|_\mu \leq \|\mathcal{T}(x) - \mathcal{T}(x^\star)\|_\mu + \|x - x^\star\|_\mu \leq (1+\gamma)\|x - x^\star\|_\mu. \tag{123}$$

where (122) follows from the identity $\langle u, M(v-u) \rangle = \frac{1}{2}(\|v\|_\mu^2 - \|u\|_\mu^2 - \|v-u\|_\mu^2)$ applied with $u = x - x^\star$ and $v = \mathcal{T}(x) - \mathcal{T}(x^\star)$ and applying contractive-ness. We now proceed to the core of this subsection.

**Lemma D.9** (Fourth-moment Lyapunov drift for contractive SA). *Consider* (120). *Suppose 3.8 and 4.2 hold. Then there exist constants* $\alpha_4^\dagger > 0$, $\kappa_4 \in (0,1)$ *and* $\beta_4 < \infty$ *such that for all* $\alpha \in (0, \alpha_4^\dagger)$,

$$\mathbb{E}\big[1 + \|E_{k+1}\|_\mu^4 \mid \mathcal{F}_k\big] \leq \kappa_4\big(1 + \|E_k\|_\mu^4\big) + \beta_4, \qquad k \geq 0. \tag{124}$$

*Proof.* Define

$$G_k := F(X_k) + \xi(Z_k),$$

so that $E_{k+1} = E_k + \alpha G_k$.

Since $\|e\|_\mu^2 = e^\top M e$, we have

$$\mathbb{E}[\|E_{k+1}\|_\mu^4 \mid \mathcal{F}_k] = T_0 + T_1 + T_2 + T_3 + T_4 + T_5,$$

where

$$\begin{aligned}
T_0 &= \|E_k\|_\mu^4, \\
T_1 &= 4\alpha\|E_k\|_\mu^2 \, \mathbb{E}[\langle E_k, MG_k \rangle \mid \mathcal{F}_k], \\
T_2 &= 4\alpha^2 \, \mathbb{E}[\langle E_k, MG_k \rangle^2 \mid \mathcal{F}_k], \\
T_3 &= 2\alpha^2\|E_k\|_\mu^2 \, \mathbb{E}[\|G_k\|_\mu^2 \mid \mathcal{F}_k], \\
T_4 &= 4\alpha^3 \, \mathbb{E}[\langle E_k, MG_k \rangle \|G_k\|_\mu^2 \mid \mathcal{F}_k], \\
T_5 &= \alpha^4 \, \mathbb{E}[\|G_k\|_\mu^4 \mid \mathcal{F}_k].
\end{aligned}$$

**Controlling $T_1$ via contraction.** Using $G_k = F(X_k) + \xi(Z_k)$,

$$T_1 = 4\alpha\|E_k\|_\mu^2\langle E_k, MF(X_k) \rangle + 4\alpha\|E_k\|_\mu^2 \, \mathbb{E}[\langle E_k, M\xi(Z_k) \rangle \mid \mathcal{F}_k]. \tag{125}$$

By (122),

$$4\alpha\|E_k\|_\mu^2\langle E_k, MF(X_k) \rangle \leq -4(1-\gamma)\alpha\|E_k\|_\mu^4.$$

For the noise cross term, by Cauchy–Schwarz in the $\mu$–inner product,

$$|\langle E_k, M\xi(Z_k) \rangle| \leq \|E_k\|_\mu \, \|\xi(Z_k)\|_\mu,$$

so,

$$\left| \mathbb{E}[\langle E_k, M\xi(Z_k) \rangle \mid \mathcal{F}_k] \right| \leq \|E_k\|_\mu \, \mathbb{E}[\|\xi(Z_k)\|_\mu \mid \mathcal{F}_k] \leq C_{\xi,1}\big(\|E_k\|_\mu + \|E_k\|_\mu^2\big).$$

Hence

$$4\alpha\|E_k\|_\mu^2 \, \mathbb{E}[\langle E_k, M\xi(Z_k) \rangle \mid \mathcal{F}_k] \leq 4C_{\xi,1}\alpha\big(\|E_k\|_\mu^3 + \|E_k\|_\mu^4\big).$$

Applying Lemmas D.3 and D.4 (with any fixed $\varepsilon \in (0, 1/3)$, and with $\|\cdot\|$ replaced by $\|\cdot\|_\mu$) yields

$$T_1 \leq -4(1-\gamma)\alpha\|E_k\|_\mu^4 + C_1\alpha^{1+\varepsilon}\|E_k\|_\mu^4 + C_1\big(\alpha^{1-3\varepsilon} + \alpha^{1-\varepsilon} + \alpha\big), \tag{126}$$

for some finite $C_1 < \infty$ depending only on $\gamma, C_{\xi,1}, \varepsilon$.

**Bounding the higher order terms**  By (123) we have $\|F(X_k)\|_\mu \leq (1+\gamma)\|E_k\|_\mu$, and by 4.2 there exists $C_G < \infty$ such that

$$\mathbb{E}[\|G_k\|_\mu^2 \mid \mathcal{F}_k] \leq C_G(1 + \|E_k\|_\mu^2), \qquad \mathbb{E}[\|G_k\|_\mu^4 \mid \mathcal{F}_k] \leq C_G(1 + \|E_k\|_\mu^4). \tag{127}$$

Moreover, by Cauchy–Schwarz,

$$\mathbb{E}[\|G_k\|_\mu^3 \mid \mathcal{F}_k] \leq \left(\mathbb{E}[\|G_k\|_\mu^4 \mid \mathcal{F}_k]\right)^{3/4} \leq C_G(1 + \|E_k\|_\mu^4).$$

Using $\langle E_k, MG_k \rangle \leq \|E_k\|_\mu \|G_k\|_\mu$ and Cauchy–Schwarz exactly as in the SGD proof,

$$\begin{aligned}
T_2 &\leq 4\alpha^2 \|E_k\|_\mu^2 \, \mathbb{E}[\|G_k\|_\mu^2 \mid \mathcal{F}_k] \leq C_2\alpha^2(1 + \|E_k\|_\mu^4), \\
T_3 &\leq 2\alpha^2 \|E_k\|_\mu^2 \, \mathbb{E}[\|G_k\|_\mu^2 \mid \mathcal{F}_k] \leq C_3\alpha^2(1 + \|E_k\|_\mu^4), \\
T_4 &\leq 4\alpha^3 \|E_k\|_\mu \, \mathbb{E}[\|G_k\|_\mu^3 \mid \mathcal{F}_k] \leq C_4\alpha^3(1 + \|E_k\|_\mu^4), \\
T_5 &\leq \alpha^4 \, \mathbb{E}[\|G_k\|_\mu^4 \mid \mathcal{F}_k] \leq C_5\alpha^4(1 + \|E_k\|_\mu^4),
\end{aligned}$$

for finite constants $C_2, \ldots, C_5$. Hence for $\alpha \in (0, 1]$,

$$T_2 + T_3 + T_4 + T_5 \leq C_{hi}\alpha^2(1 + \|E_k\|_\mu^4), \tag{128}$$

absorbing $\alpha^3, \alpha^4$ into $\alpha^2$.

**Drift inequality.**  Substituting (126) and (128) into the split and using $T_0 = \|E_k\|_\mu^4$ yields

$$\mathbb{E}[\|E_{k+1}\|_\mu^4 \mid \mathcal{F}_k] \leq \left(1 - 4(1-\gamma)\alpha + C_1\alpha^{1+\varepsilon} + C_{hi}\alpha^2\right)\|E_k\|_\mu^4 + C\left(\alpha^{1-3\varepsilon} + \alpha^{1-\varepsilon} + \alpha + \alpha^2\right),$$

for a finite $C < \infty$. Adding 1 and recalling $L_{4,\mu}(e) := 1 + \|e\|_\mu^4$ gives

$$\mathbb{E}[L_{4,\mu}(E_{k+1}) \mid \mathcal{F}_k] \leq \left(1 - 4(1-\gamma)\alpha + C_1\alpha^{1+\varepsilon} + C_{hi}\alpha^2\right)L_{4,\mu}(E_k) + C\left(\alpha^{1-3\varepsilon} + \alpha^{1-\varepsilon} + \alpha + \alpha^2\right),$$

where $L_{4,\mu}(e) := 1 + \|e\|_\mu^4$. Choose $\alpha_4^\dagger \in (0, 1]$ small enough that for all $\alpha \in (0, \alpha_4^\dagger)$,

$$1 - 4(1-\gamma)\alpha + C_1\alpha^{1+\varepsilon} + C_{hi}\alpha^2 \leq: \kappa_4 < 1,$$

and set

$$\beta_4 := C\left(\alpha^{1-3\varepsilon} + \alpha^{1-\varepsilon} + \alpha + \alpha^2\right).$$

This proves (124). $\qquad\qquad\qquad\qquad\qquad\qquad\qquad\qquad\qquad\qquad\qquad\qquad\qquad\qquad\qquad\square$

**Corollary D.10** (Bounded fourth moment for contractive SA).  *Suppose the Markov chain $\{(X_k, Z_k)\}$ induced by (120) admits a stationary distribution $\pi_\alpha$. Then for any $\alpha \in (0, \alpha_4^\dagger)$,*

$$\int \|x - x^\star\|_\mu^4 \, \pi_\alpha(dx, dz) < \infty.$$

*In particular, the stationary law of $X_k$ has finite fourth moment.*

*Proof.*  Start in stationarity: $(X_0, Z_0) \sim \pi_\alpha$, and let $E_0 = X_0 - x^\star$. Taking expectations in (124) under $\pi_\alpha$ and using stationarity yields

$$\mathbb{E}_{\pi_\alpha}[L_{4,\mu}(E_1)] = \mathbb{E}_{\pi_\alpha}[L_{4,\mu}(E_0)] \leq \kappa_4 \mathbb{E}_{\pi_\alpha}[L_{4,\mu}(E_0)] + \beta_4.$$

Rearranging gives $\mathbb{E}_{\pi_\alpha}\|E_0\|_\mu^4 < \infty$. Finally, $\|X_0\|^4 \leq 8(\|E_0\|^4 + \|x^\star\|^4)$ implies $\mathbb{E}_{\pi_\alpha}\|X_0\|^4 < \infty$. $\qquad\square$

### D.4. Establishing Uniqueness via a Minorization

In this section, we show a method to derive the uniqueness and existence of the stationary distribution in SGD with Markovian noise under a minorization assumption, instead of directly assuming the uniqueness in the main text. Note that this section is not necessary for the proof of 4.3, because we assumed that the joint chain admits a unique stationary distribution in 4.2. Furthermore, one can derive uniqueness via a myriad of ways under different assumptions; we simply present one such example. The minorization assumption we will make (solely for this subsection) is as follows:

**Assumption D.11.** For each $z \in \mathsf{Z}$, the conditional law of $\xi_{k+1}$ given $Z_k = z$ admits a decomposition

$$\mathcal{L}(\xi_{k+1} \mid Z_k = z) = \mu_{1,z} + \mu_{2,z},$$

where $\mu_{1,z}$ and $\mu_{2,z}$ are sub–probability measures on $(\mathbb{R}^d, \mathcal{B}(\mathbb{R}^d))$. Moreover, $\mu_{1,z}$ is absolutely continuous with respect to Lebesgue measure, with density $p_z$:

$$\mu_{1,z}(A) = \int_A p_z(u) \, du, \qquad A \in \mathcal{B}(\mathbb{R}^d).$$

Finally, for every bounded set $C \subset \mathsf{Z}$ and every bounded set $B \subset \mathbb{R}^d$,

$$\inf_{z \in C} \inf_{u \in B} p_z(u) \; > \; 0.$$

This assumption is the Markovian analogue of the minorization assumption in (Yu et al., 2020).Our methodology will roughly follow the same structure as theirs. The lemma we will prove is as follows.

**Lemma D.12.** *Suppose $f$ satisfies Assumption 3.2, and the noise satisfies Assumption 4.2 and D.11. Consider the SGD recursion*

$$X_{k+1} = X_k - \alpha\big(\nabla f(X_k) + \xi(Z_k)\big), \qquad k \geq 0. \tag{129}$$

*Then, there exist a constant $\alpha_4^\dagger > 0$, such that for all $\alpha \in (0, \alpha_4^\dagger)$, the joint chain $\big(X_k^{(\alpha)}, Z_k\big) \in \mathbb{R}^d \times \mathsf{Z}$ given by (129) admits a unique invariant probability distribution.*

*Proof.* Our strategy will be to show that the chain is $\psi$-irreducible, aperiodic, and Harris Recurrent. Because we already showed the contractiveness of the Lyapunov function $1 + \|x\|^4$ in 95, the proof will follow by Theorem 13.0.1 of (Meyn & Tweedie, 2009). The details are as follows:

**Step 1: $\psi$–irreducibility of the joint chain.** Define the $\sigma$–finite measure

$$\psi(A) := \int_{\mathsf{Z}} \mathrm{Leb}\big(\{x \in \mathbb{R}^d : (x, z) \in A\}\big) \, \pi_Z(dz), \qquad A \in \mathcal{B}(\mathbb{R}^d \times \mathsf{Z}),$$

i.e. Lebesgue measure on $\mathbb{R}^d$ multiplied by the invariant distribution $\pi_Z$ of the chain $(Z_k)$. Let $B \subset \mathbb{R}^d \times \mathsf{Z}$ be measurable with $\psi(B) > 0$, and write

$$B_z := \{x \in \mathbb{R}^d : (x, z) \in B\}$$

for its vertical sections. Then

$$\psi(B) = \int_{\mathsf{Z}} \mathrm{Leb}(B_z) \, \pi_Z(dz) > 0,$$

so the set

$$\mathsf{Z}_B := \{z \in \mathsf{Z} : \mathrm{Leb}(B_z) > 0\}$$

has strictly positive $\pi_Z$–measure. Pick $z^\star \in \mathsf{Z}_B$ with $\mathrm{Leb}(B_{z^\star}) > 0$. Since $(Z_k)$ is $\pi_Z$–irreducible, for every initial $z \in \mathsf{Z}$ there exists $n_0 \geq 1$ such that

$$\mathbb{P}_z\{Z_{n_0} = z^\star\} > 0.$$

Fix an arbitrary initial state $(x, z) \in \mathbb{R}^d \times \mathsf{Z}$. Then

$$P^{n_0+1}\big((x, z), B\big) = \mathbb{P}_{(x,z)}\big\{(X_{n_0+1}^{(\alpha)}, Z_{n_0+1}) \in B\big\}$$

$$\geq \mathbb{P}_{(x,z)}\big\{X_{n_0+1}^{(\alpha)} \in B_{z^\star}, \ Z_{n_0} = z^\star\big\}$$

$$= \mathbb{E}_{(x,z)}\Big[\mathbf{1}_{\{Z_{n_0}=z^\star\}}\mathbb{P}\Big(X_{n_0+1}^{(\alpha)} \in B_{z^\star} \,\Big|\, X_{n_0}^{(\alpha)}, Z_{n_0}\Big)\Big].$$

On the event $\{Z_{n_0} = z^\star\}$ we have the update

$$X_{n_0+1}^{(\alpha)} = X_{n_0}^{(\alpha)} - \alpha\nabla f\Big(X_{n_0}^{(\alpha)}\Big) + \alpha\,\xi(Z_{n_0}),$$

and by Assumption D.11 the conditional law of $\xi(Z_{n_0})$ given $Z_{n_0} = z^\star$ has a density that is positive Lebesgue-a.e. on $\mathbb{R}^d$. Since $\mathrm{Leb}(B_{z^\star}) > 0$, it follows that for every realised value of $X_{n_0}^{(\alpha)}$,

$$\mathbb{P}\Big(X_{n_0+1}^{(\alpha)} \in B_{z^\star} \,\Big|\, X_{n_0}^{(\alpha)}, Z_{n_0} = z^\star\Big) > 0.$$

Consequently, the random variable

$$\mathbf{1}_{\{Z_{n_0}=z^\star\}}\mathbb{P}\Big(X_{n_0+1}^{(\alpha)} \in B_{z^\star} \,\Big|\, X_{n_0}^{(\alpha)}, Z_{n_0}\Big)$$

is strictly positive on the event $\{Z_{n_0} = z^\star\}$, which has positive probability $\mathbb{P}_z\{Z_{n_0} = z^\star\} > 0$. Therefore

$$P^{n_0+1}\big((x,z), B\big) > 0.$$

Since $(x,z)$ and $B$ were arbitrary with $\psi(B) > 0$, the joint chain $\{(X_k^{(\alpha)}, Z_k)\}_{k\geq 0}$ is $\psi$–irreducible.

**Step 2: Aperiodicity of the joint chain.** We now show that the joint chain $\{(X_k^{(\alpha)}, Z_k)\}_{k\geq 0}$ is aperiodic. Fix a bounded nonempty Borel set $A \subset \mathbb{R}^d$ with $\mathrm{Leb}(A) > 0$, and set

$$B := A \times \mathsf{Z}.$$

By Assumption D.11(ii), for each bounded $C_X \subset \mathbb{R}^d$ there exists a constant $\varepsilon_{C_X} > 0$ such that, writing $C := C_X \times \mathsf{Z}$, we have the minorisation

$$P\big((x,z), A \times \mathsf{Z}\big) = \mathbb{P}\{X_1^{(\alpha)} \in A \mid X_0^{(\alpha)} = x, Z_0 = z\} \ \geq\ \varepsilon_{C_X}\,\mathrm{Leb}(A), \qquad \forall(x,z) \in C.$$

Choose $C_X$ bounded with $A \subset C_X$, so that $B \subset C$, and define

$$\varepsilon_B := \varepsilon_{C_X}\,\mathrm{Leb}(A) > 0.$$

Then for every $(x,z) \in B$,

$$P\big((x,z), B\big) = \mathbb{P}\{X_1^{(\alpha)} \in A \mid X_0^{(\alpha)} = x, Z_0 = z\} \ \geq\ \varepsilon_B \ > \ 0.$$

In particular, every state $(x,z) \in B$ returns to $B$ in one step with positive probability, so the period of each state in $B$ is equal to 1.

It remains to show that the whole chain is aperiodic. For a state $y = (x,z)$, define its period

$$d(y) := \gcd\{n \geq 1 : P^n(y, B) > 0\},$$

with the convention that the gcd of an empty set is 0. For any $y \in B$ we have $P(y, B) > 0$, so $1 \in \{n \geq 1 : P^n(y, B) > 0\}$ and hence $d(y) = 1$.

Now let $y_0$ be any state with $d(y_0) > 0$. By $\psi$–irreducibility (Step 1), there exist integers $m, r \geq 0$ and a state $y_1 \in B$ such that

$$P^m(y_0, y_1) > 0, \qquad P^r(y_1, y_0) > 0.$$

Then for every $n$ with $P^n(y_1, B) > 0$ we also have

$$P^{m+n+r}(y_0, B) \ \geq\ P^m(y_0, y_1)\,P^n(y_1, B)\,P^r(y_1, y_0) \ > \ 0.$$

This shows that every such $m + n + r$ is a multiple of $d(y_0)$, so $d(y_0)$ divides

$$\gcd\{n \geq 1 : P^n(y_1, B) > 0\} = d(y_1) = 1.$$

Thus $d(y_0) = 1$. Since $y_0$ was arbitrary among states with $d(y_0) > 0$, we conclude that $d(y) = 1$ for all states with finite period, i.e. the chain is aperiodic.

**Step 3: Harris recurrence of the joint chain.** We now verify the assumptions of Proposition 10.2.4 of (Douc et al., 2018) for the joint Markov chain

$$X_k := (X_k^{(\alpha)}, Z_k) \in \mathbb{R}^d \times \mathsf{Z}.$$

*Lyapunov function and drift condition.* Recall that by Lemma D.5, for all $\alpha \in (0, \alpha_4^\dagger)$ there exist constants $\kappa_4 \in (0, 1)$ and $\beta_4 < \infty$ such that

$$\mathbb{E}\left[1 + \|X_{k+1}^{(\alpha)}\|^4 \,\middle|\, X_k^{(\alpha)} = x\right] \leq \kappa_4 \left(1 + \|x\|^4\right) + \beta_4. \tag{130}$$

Define the Lyapunov function

$$V(x) := 1 + \|x\|^4, \qquad \tilde{V}(x, z) := V(x),$$

the latter being the extension of $V$ to $\mathbb{R}^d \times \mathsf{Z}$. Let $R > 0$ be chosen sufficiently large that

$$R \geq \frac{2\beta_4}{1 - \kappa_4}, \qquad \text{and define} \qquad C_X := \{x \in \mathbb{R}^d : V(x) \leq R\}, \quad C := C_X \times \mathsf{Z}.$$

Then $C_X$ is bounded and therefore $C$ is bounded in the $X$–coordinate.

If $(x, z) \notin C$, then $V(x) > R$ and $\beta_4 \leq \frac{1 - \kappa_4}{2} V(x)$, so from (130) we obtain

$$\mathbb{E}[\tilde{V}(X_{k+1}) \mid X_k = (x, z)] \leq \left(\kappa_4 + \frac{1 - \kappa_4}{2}\right) V(x) =: \gamma \, \tilde{V}(x, z),$$

where

$$\gamma := \frac{1 + \kappa_4}{2} \in (\kappa_4, 1).$$

On the other hand, since $C_X$ is bounded we have

$$b := \sup_{x \in C_X} \mathbb{E}\left[V(X_{k+1}^{(\alpha)}) \,\middle|\, X_k^{(\alpha)} = x\right] < \infty.$$

Combining the two displays yields the geometric drift condition

$$\mathbb{E}[\tilde{V}(X_{k+1}) \mid X_k = (x, z)] \leq \gamma \, \tilde{V}(x, z) + b \, \mathbf{1}_C(x, z), \qquad (x, z) \in \mathbb{R}^d \times \mathsf{Z}. \tag{131}$$

*Return time to $C$.* Let

$$\sigma_C := \inf\{n > 0 : X_n \in C\}$$

denote the hitting time of $C$ by the joint chain. Applying (131) (see, e.g., Lemma A.3 and Corollary A.4 of (Mattingly et al., 2002)) we obtain the existence of constants $\kappa < \infty$ and $\rho \in (0, 1)$ such that, for every initial state $(x, z) \in \mathbb{R}^d \times \mathsf{Z}$ and every $n \geq 0$,

$$\mathbb{P}_{(x,z)}(\sigma_C > n) \leq \kappa \rho^n [\tilde{V}(x, z) + 1], \tag{132}$$

and in particular

$$\mathbb{P}_{(x,z)}(\sigma_C < \infty) = 1. \tag{133}$$

*Petite set.* By Assumption D.11(ii), the conditional law of $\xi(Z_k)$ given $Z_k = z$ has an absolutely continuous component with density uniformly bounded below on bounded subsets of $\mathbb{R}^d$. Since $C_X$ is bounded, this yields a one–step minorisation on $C$, and hence $C$ is a small (and therefore petite) set for the joint chain.

By Step 1 the joint chain is $\psi$–irreducible. By the previous paragraph, $C$ is petite, and by (133) we have $\mathbb{P}_{(x,z)}(\sigma_C < \infty) = 1$ for all $(x, z) \notin C$. Therefore the hypotheses of Proposition 10.2.4 of (Douc et al., 2018) are satisfied, and the joint chain $\{(X_k^{(\alpha)}, Z_k)\}_{k \geq 0}$ is *Harris recurrent*.

Because the joint chain is Harris recurrent, and the function $V$ with petite set $C$ used in the previous step satisfies Condition (4) of Theorem 13.0.1 of (Meyn & Tweedie, 2009), we have that $\pi$ is the unique invariant measure of the joint chain, as desired. $\qquad\square$

# E. SGD with general convex objective $f$

### E.1. Proof of Proposition 5.5 assuming 5

We will show that under our conjecture (regularity of Stein solutions for the Gibbs distribution in 1-dimension), and necessary regularity requirements,

### E.1.1. GENERATOR COUPLING OF $Y$ AND $Y^{(\alpha)}$

Because we are in the setting of constant step-size, we can derive the constant step-size generator with the same framework. That is, with the conjectured scaling of $g(\alpha) = \alpha^{\frac{1}{h}}$, the discrete generator is given by

$$\mathcal{L}^{(\alpha)} g(y) = \frac{1}{\alpha^{2-\frac{2}{h}}} \mathbb{E}\left[ g(Y_{k+1}^{(\alpha)}) - g(Y_k^{(\alpha)}) \,\Big|\, Y_k^{(\alpha)} = y \right] \tag{134}$$

$$= \frac{1}{\alpha^{2-\frac{2}{h}}} \mathbb{E}\left[ g\left( y + \alpha^{1-\frac{1}{h}} \left( -f'(x^\star + \alpha^{1/h} y) + \xi \right) \right) - g(y) \right], \tag{135}$$

where $\xi \overset{d}{=} \xi_0$ is independent of $y$ and $Y_k^{(\alpha)}$ denotes the Markov chain induced by SGD under the centered and scaled iterates $Y_k^{(\alpha)} = \frac{(X_k^{(\alpha)} - x^\star)}{\alpha^{1/h}}$.

Because we are aiming to show convergence in distribution to the Gibbs distribution, we write its generator below. For notation, let $Y$ denote the one-dimensional Gibbs distribution with density

$$p(y) \propto \exp\left( -\frac{2}{\sigma^2} \cdot \frac{f^{(h)}(x^\star)}{h!} \, y^h \right). \tag{136}$$

The corresponding Stein generator is

$$\mathcal{L}g(y) := -\frac{f^{(h)}(x^\star)}{(h-1)!} \, y^{h-1} g'(y) + \frac{1}{2}\sigma^2 \, g''(y). \tag{137}$$

as it satisfies the Stein identity $\mathbb{E}[\mathcal{L}g(Y)] = 0$ for all sufficiently regular $g$. Then, following the standard generator coupling setup, we have:

$$
\begin{aligned}
d_W\left( Y^{(\alpha)}, Y \right) &= \sup_{h \in \mathrm{Lip}_1(\mathbb{R})} \mathbb{E}\left[ h\left( Y^{(\alpha)} \right) \right] - \mathbb{E}[h(Y)] \\
&= \sup_{h \in \mathrm{Lip}_1(\mathbb{R})} \mathbb{E}\left[ h\left( Y^{(\alpha)} \right) - \mathbb{E}[h(Y)] \right] \\
&= \sup_{h \in \mathrm{Lip}_1(\mathbb{R})} \mathbb{E}\left[ \mathcal{L}g_h\left( Y^{(\alpha)} \right) \right] \\
&= \sup_{h \in \mathrm{Lip}_1(\mathbb{R})} \mathbb{E}\left[ \mathcal{L}g_h\left( Y^{(\alpha)} \right) - \mathcal{L}^{(\alpha)} g_h\left( Y^{(\alpha)} \right) \right] \\
&\leq \sup_{g \in \mathcal{F}} \mathbb{E}\left[ \mathcal{L}g\left( Y^{(\alpha)} \right) - \mathcal{L}^{(\alpha)} g\left( Y^{(\alpha)} \right) \right],
\end{aligned}
$$

where $g_h$ denotes the Stein solution for the Gibbs distribution and $\mathcal{F} := \{g_h : h \in \mathrm{Lip}_1(\mathbb{R})\}$.

### E.1.2. BOUNDING THE GENERATOR DIFFERENCE

Plugging in the definition of $\mathcal{L}^{(\alpha)}$ and $\mathcal{L}$, we have

$$
\begin{aligned}
&\mathcal{L}g_h(y) - \mathcal{L}^{(\alpha)} g_h(y) \\
&= \frac{1}{2}\sigma^2 \, g_h''(y) - \frac{f^{(h)}(x^\star)}{(h-1)!} \, y^{h-1} g_h'(y) - \frac{1}{\alpha^{2-\frac{2}{h}}} \mathbb{E}\left[ g_h(Y_{k+1}) - g_h(Y_k) \,\big|\, Y_k = y \right].
\end{aligned} \tag{138}
$$

We will now Taylor expand $g_h(Y_{k+1}) - g_h(Y_k)$ in the above expression. Abstracting away the third-order remainder term as $R_3(Y_{k+1}, Y_k)$, we have that

$$(138) = \frac{1}{2}\sigma^2 \, g_h''(y) - \frac{f^{(h)}(x^\star)}{(h-1)!} \, y^{h-1} g_h'(y)$$

$$- \frac{1}{\alpha^{2-\frac{2}{h}}} \mathbb{E}\left[ g_h'(Y_k)\,(Y_{k+1} - Y_k) + \frac{1}{2}\, g_h''(Y_k)\,(Y_{k+1} - Y_k)^2 + R_3(Y_{k+1}, Y_k) \,\big|\, Y_k = y \right]$$

$$= W_1 + W_2 + Rem.$$

where the three terms are defined as follows:

$$W_1 := -\frac{f^{(h)}(x^\star)}{(h-1)!}\, y^{h-1} g_h'(y) - \frac{1}{\alpha^{2-\frac{2}{h}}}\, \mathbb{E}\left[ g_h'(Y_k)\,(Y_{k+1} - Y_k) \,\big|\, Y_k = y \right],$$

$$W_2 := \frac{1}{2}\, \sigma^2\, g_h''(y) - \frac{1}{\alpha^{2-\frac{2}{h}}}\, \mathbb{E}\left[ \frac{1}{2}\, g_h''(Y_k)\,(Y_{k+1} - Y_k)^2 \,\big|\, Y_k = y \right],$$

$$Rem := -\frac{1}{\alpha^{2-\frac{2}{h}}}\, \mathbb{E}\left[ R_3(Y_{k+1}, Y_k) \,\big|\, Y_k = y \right].$$

**Bounding $W_1$.**

$$|W_1| = \left| -\frac{f^{(h)}(x^\star)}{(h-1)!}\, y^{h-1} g_h'(y) - \frac{1}{\alpha^{2-\frac{2}{h}}}\, \mathbb{E}\left[ g_h'(Y_k)\,(Y_{k+1} - Y_k) \,\big|\, Y_k = y \right] \right|$$

$$= \left| -\frac{f^{(h)}(x^\star)}{(h-1)!}\, y^{h-1} g_h'(y) - \frac{1}{\alpha^{2-\frac{2}{h}}}\, \mathbb{E}\left[ g_h'(y)\,(Y_{k+1} - Y_k) \,\big|\, Y_k = y \right] \right|$$

$$= \left| -\frac{f^{(h)}(x^\star)}{(h-1)!}\, y^{h-1} g_h'(y) - \frac{g_h'(y)}{\alpha^{2-\frac{2}{h}}}\, \mathbb{E}\left[ Y_{k+1} - Y_k \,\big|\, Y_k = y \right] \right|$$

$$= \left| -\frac{f^{(h)}(x^\star)}{(h-1)!}\, y^{h-1} g_h'(y) - \frac{g_h'(y)}{\alpha^{2-\frac{2}{h}}}\, \mathbb{E}\left[ \alpha^{1-\frac{1}{h}}\left( -f'(x^\star + \alpha^{1/h} y) + \xi_k \right) \right] \right| \qquad \text{(SGD recursion)}$$

$$= \left| -\frac{f^{(h)}(x^\star)}{(h-1)!}\, y^{h-1} g_h'(y) + \alpha^{-1+\frac{1}{h}} f'(x^\star + \alpha^{1/h} y)\, g_h'(y) \right| \qquad (\mathbb{E}[\xi_k] = 0)$$

$$= |g_h'(y)| \left| \alpha^{-1+\frac{1}{h}} f'(x^\star + \alpha^{1/h} y) - \frac{f^{(h)}(x^\star)}{(h-1)!}\, y^{h-1} \right|$$

$$= |g_h'(y)| \left| \alpha^{-1+\frac{1}{h}}\left( f'(x^\star + \alpha^{1/h} y) - \frac{f^{(h)}(x^\star)}{(h-1)!}\, (\alpha^{1/h} y)^{h-1} \right) \right|.$$

Observe that by a Taylor expansion with integral remainder, we have

$$\left| f'(x^\star + \alpha^{1/h} y) - \frac{f^{(h)}(x^\star)}{(h-1)!}\, (\alpha^{1/h} y)^{h-1} \right|$$

$$= \left| \frac{(\alpha^{1/h} y)^h}{(h-1)!} \int_0^1 (1-t)^{h-1} f^{(h+1)}(x^\star + t\alpha^{1/h} y)\, dt \right|$$

$$\leq \frac{\alpha |y|^h}{(h-1)!} \int_0^1 (1-t)^{h-1} \left| f^{(h+1)}(x^\star + t\alpha^{1/h} y) \right| dt$$

$$\leq \frac{\alpha |y|^h}{(h-1)!} \int_0^1 (1-t)^{h-1} M\, dt \qquad \text{(Assumption 5.1)}$$

$$= \frac{\alpha |y|^h}{(h-1)!} \cdot \frac{M}{h}$$

and therefore,

$$|W_1| \leq |g_h'(y)|\, \alpha^{-1+\frac{1}{h}} \cdot \frac{\alpha |y|^h}{(h-1)!} \cdot \frac{M}{h}$$

$$= \frac{M}{h(h-1)!}\, \alpha^{1/h}\, |y|^h\, |g_h'(y)|$$

$$\leq \frac{MR}{h(h-1)!}\, \alpha^{1/h}\, |y|^h \qquad \text{(Conjecture 5.4)}.$$

By Conjecture 5.3, we have that $\mathbb{E}|Y^{(\alpha)}|^h$ is bounded, yielding that

$$\mathbb{E}[|W_1|] \leq \frac{MR}{h(h-1)!}\, \alpha^{1/h}\, \mathbb{E}\left[|Y^{(\alpha)}|^h\right]. \tag{139}$$

is of the correct order.

**Bounding $W_2$.**

$$
\begin{aligned}
|W_2| &= \left| \frac{1}{2}\,\sigma^2\, g_h''(y) - \frac{1}{\alpha^{2-\frac{2}{h}}}\, \mathbb{E}\left[\frac{1}{2}\, g_h''(Y_k)\,(Y_{k+1}-Y_k)^2 \,\big|\, Y_k = y\right]\right| \\
&= \left| \frac{1}{2}\,\sigma^2\, g_h''(y) - \frac{1}{\alpha^{2-\frac{2}{h}}}\, \mathbb{E}\left[\frac{1}{2}\, g_h''(y)\,(Y_{k+1}-Y_k)^2 \,\big|\, Y_k = y\right]\right| \\
&= \frac{|g_h''(y)|}{2}\, \left| \sigma^2 - \frac{1}{\alpha^{2-\frac{2}{h}}}\, \mathbb{E}\left[(Y_{k+1}-Y_k)^2 \,\big|\, Y_k = y\right]\right| \\
&= \frac{|g_h''(y)|}{2}\, \left| \sigma^2 - \frac{1}{\alpha^{2-\frac{2}{h}}}\, \mathbb{E}\left[\alpha^{2-\frac{2}{h}}\left(-f'(x^\star + \alpha^{1/h}y) + \xi_k\right)^2\right]\right| \qquad \text{(SGD recursion)} \\
&= \frac{|g_h''(y)|}{2}\, \left| \sigma^2 - \mathbb{E}\left[\left(-f'(x^\star + \alpha^{1/h}y) + \xi_k\right)^2\right]\right| \\
&= \frac{|g_h''(y)|}{2}\, \left| \sigma^2 - \left(f'(x^\star + \alpha^{1/h}y)^2 - 2f'(x^\star + \alpha^{1/h}y)\mathbb{E}[\xi_k] + \mathbb{E}[\xi_k^2]\right)\right| \\
&= \frac{|g_h''(y)|}{2}\, \left| \sigma^2 - \left(f'(x^\star + \alpha^{1/h}y)^2 + \sigma^2\right)\right| \qquad (\mathbb{E}[\xi_k] = 0,\ \mathbb{E}[\xi_k^2] = \sigma^2) \\
&= \frac{|g_h''(y)|}{2}\, \left| f'(x^\star + \alpha^{1/h}y)^2 \right| \\
&= \frac{|g_h''(y)|}{2}\, \left( f'(x^\star + \alpha^{1/h}y)^2 \right).
\end{aligned}
$$

Note that by a Taylor expansion, we have that

$$
\begin{aligned}
\left| f'(x^\star + \alpha^{1/h}y)\right| &\leq \left| \frac{f^{(h)}(x^\star)}{(h-1)!}\, (\alpha^{1/h}y)^{h-1}\right| + \left| \frac{(\alpha^{1/h}y)^h}{(h-1)!} \int_0^1 (1-t)^{h-1}\, f^{(h+1)}(x^\star + t\alpha^{1/h}y)\, dt\right| \\
&\leq \frac{f^{(h)}(x^\star)}{(h-1)!}\, \alpha^{\frac{h-1}{h}}\, |y|^{h-1} + \frac{\alpha|y|^h}{(h-1)!} \int_0^1 (1-t)^{h-1}\, \left|f^{(h+1)}(x^\star + t\alpha^{1/h}y)\right|\, dt \\
&\leq \frac{f^{(h)}(x^\star)}{(h-1)!}\, \alpha^{\frac{h-1}{h}}\, |y|^{h-1} + \frac{\alpha|y|^h}{(h-1)!} \int_0^1 (1-t)^{h-1}\, M\, dt \qquad \text{(Assumption 5.1)} \\
&= \frac{f^{(h)}(x^\star)}{(h-1)!}\, \alpha^{\frac{h-1}{h}}\, |y|^{h-1} + \frac{M}{h(h-1)!}\, \alpha\, |y|^h.
\end{aligned}
$$

which implies that

$$
\begin{aligned}
|W_2| &\leq \frac{|g_h''(y)|}{2}\, \left( \frac{f^{(h)}(x^\star)}{(h-1)!}\, \alpha^{\frac{h-1}{h}}\, |y|^{h-1} + \frac{M}{h(h-1)!}\, \alpha\, |y|^h \right)^2 \\
&\leq |g_h''(y)|\, \left( \left(\frac{f^{(h)}(x^\star)}{(h-1)!}\right)^2 \alpha^{2-\frac{2}{h}}|y|^{2h-2} + \left(\frac{M}{h(h-1)!}\right)^2 \alpha^2|y|^{2h}\right) \qquad ((a+b)^2 \leq 2a^2 + 2b^2) \\
&\leq C\, |g_h''(y)|\, \alpha^{2-\frac{2}{h}}\, \left(|y|^{2h-2} + |y|^{2h}\right) \qquad (\alpha \in (0,1)) \\
&\leq C\, |g_h''(y)|\, \alpha^{2-\frac{2}{h}}\, \left(1 + |y|^{2h}\right) \qquad (|y|^{2h-2} \leq 1 + |y|^{2h}) \\
&\leq C\, R\, \alpha^{2-\frac{2}{h}}\, \left(1 + |y|^{2h}\right) \qquad \text{(Conjecture 5.4)}.
\end{aligned}
$$

Therefore, by 5.3

$$
\begin{aligned}
\mathbb{E}[|W_2|] &\leq C\, R\, \alpha^{2-\frac{2}{h}}\, \left(1 + \mathbb{E}\left[|Y^{(\alpha)}|^{2h}\right]\right) \\
&\leq C\, R\, \alpha^{2-\frac{2}{h}}\, (1 + C_h)
\end{aligned}
$$

**Bounding** $Rem$**.**

$$
\begin{aligned}
|Rem| &= \left| -\frac{1}{\alpha^{2-\frac{2}{h}}} \, \mathbb{E}\big[R_3(Y_{k+1}, Y_k) \,\big|\, Y_k = y\big] \right| \\
&\leq \frac{1}{\alpha^{2-\frac{2}{h}}} \, \mathbb{E}\big[\, |R_3(Y_{k+1}, Y_k)| \,\big|\, Y_k = y\big] \\
&= \frac{1}{\alpha^{2-\frac{2}{h}}} \, \mathbb{E}\left[\left| \frac{1}{2}\,(Y_{k+1}-Y_k)^3 \int_0^1 (1-t)^2 \, g_h^{(3)}\big(Y_k + t(Y_{k+1}-Y_k)\big) \, dt \right| \,\bigg|\, Y_k = y\right] && \text{(Taylor integral remainder)} \\
&\leq \frac{1}{\alpha^{2-\frac{2}{h}}} \, \frac{1}{2} \left( \int_0^1 (1-t)^2 \, dt \right) \mathbb{E}\left[ |Y_{k+1}-Y_k|^3 \sup_{u\in\mathbb{R}} |g_h^{(3)}(u)| \,\Big|\, Y_k = y \right] \\
&= \frac{1}{\alpha^{2-\frac{2}{h}}} \, \frac{1}{6} \, \sup_{u\in\mathbb{R}} |g_h^{(3)}(u)| \, \mathbb{E}\big[ |Y_{k+1}-Y_k|^3 \,\big|\, Y_k = y\big] \\
&\leq \frac{R}{6\,\alpha^{2-\frac{2}{h}}} \, \mathbb{E}\big[ |Y_{k+1}-Y_k|^3 \,\big|\, Y_k = y\big] && \text{(Conj. 5.4)} \\
&= \frac{R}{6\,\alpha^{2-\frac{2}{h}}} \, \mathbb{E}\left[ \left| \alpha^{1-\frac{1}{h}}\left( -f'(x^\star + \alpha^{1/h}y) + \xi_k \right) \right|^3 \right] && \text{(SGD recursion)} \\
&= \frac{R}{6} \, \alpha^{1-\frac{1}{h}} \, \mathbb{E}\left[ \left| -f'(x^\star + \alpha^{1/h}y) + \xi_k \right|^3 \right] \\
&\leq \frac{2R}{3} \, \alpha^{1-\frac{1}{h}} \left( \left| f'(x^\star + \alpha^{1/h}y) \right|^3 + \mathbb{E}\big[ |\xi_k|^3 \big] \right) && ((|a|+|b|)^3 \leq 4(|a|^3+|b|^3)).
\end{aligned}
$$

Denote $u = \alpha^{1/h}y$, then by a Taylor expansion with integral remainder, we have

$$
\begin{aligned}
|f'(x^\star + u)| &= \left| \frac{f^{(h)}(x^\star)}{(h-1)!} \, u^{h-1} + \frac{u^h}{(h-1)!} \int_0^1 (1-t)^{h-1} f^{(h+1)}(x^\star + tu) \, dt \right| \\
&\leq \frac{f^{(h)}(x^\star)}{(h-1)!} \, |u|^{h-1} + \frac{|u|^h}{(h-1)!} \int_0^1 (1-t)^{h-1} \left| f^{(h+1)}(x^\star + tu) \right| dt \\
&\leq \frac{f^{(h)}(x^\star)}{(h-1)!} \, |u|^{h-1} + \frac{|u|^h}{(h-1)!} \int_0^1 (1-t)^{h-1} M \, dt && \text{(Assump. 5.1)} \\
&= \frac{f^{(h)}(x^\star)}{(h-1)!} \, |u|^{h-1} + \frac{M}{h(h-1)!} \, |u|^h.
\end{aligned}
$$

This implies that

$$
\begin{aligned}
\left| f'(x^\star + \alpha^{1/h}y) \right|^3 &\leq 4\left( \frac{f^{(h)}(x^\star)}{(h-1)!} \right)^3 \left| \alpha^{1/h}y \right|^{3h-3} + 4\left( \frac{M}{h(h-1)!} \right)^3 \left| \alpha^{1/h}y \right|^{3h} && ((a+b)^3 \leq 4(a^3+b^3)) \\
&= 4\left( \frac{f^{(h)}(x^\star)}{(h-1)!} \right)^3 \alpha^{3-\frac{3}{h}} |y|^{3h-3} + 4\left( \frac{M}{h(h-1)!} \right)^3 \alpha^3 |y|^{3h}
\end{aligned}
$$

and therefore

$$
\begin{aligned}
|Rem| &\leq \frac{2R}{3}\, \alpha^{1-\frac{1}{h}} \left( 4\left( \frac{f^{(h)}(x^\star)}{(h-1)!} \right)^3 \alpha^{3-\frac{3}{h}} |y|^{3h-3} + 4\left( \frac{M}{h(h-1)!} \right)^3 \alpha^3 |y|^{3h} + \mathbb{E}\big[|\xi_k|^3\big] \right) \\
&\leq C\,\alpha^{4-\frac{4}{h}} |y|^{3h-3} + C\,\alpha^{4-\frac{1}{h}} |y|^{3h} + C\,\alpha^{1-\frac{1}{h}},
\end{aligned}
$$

where

$$
C := \frac{2R}{3} \max\left\{ 4\left( \frac{f^{(h)}(x^\star)}{(h-1)!} \right)^3, \, 4\left( \frac{M}{h(h-1)!} \right)^3, \, \mathbb{E}\big[|\xi_k|^3\big] \right\}.
$$

Taking the expectation and applying the moment bounds from 5.3 yields

$$
\mathbb{E}[|R|] \leq C\,\alpha^{4-\frac{4}{h}} \mathbb{E}\left[ |Y^{(\alpha)}|^{3h-3} \right] + C\,\alpha^{4-\frac{1}{h}} \mathbb{E}\left[ |Y^{(\alpha)}|^{3h} \right] + C\,\alpha^{1-\frac{1}{h}}.
$$

Therefore, putting things together, we can complete the proof as follows:

$$
\begin{aligned}
d_W(Y^{(\alpha)}, Y) &\leq \sup_{g \in \mathcal{F}} \mathbb{E}\left[ \mathcal{L}g\left(Y^{(\alpha)}\right) - \mathcal{L}^{(\alpha)}g\left(Y^{(\alpha)}\right) \right] \\
&\leq \mathbb{E}[W_1] + \mathbb{E}[W_2] + \mathbb{E}[Rem] \\
&\leq \frac{2MRC_h}{h(h-1)!}\, \alpha^{1/h}.
\end{aligned}
\tag{140}
$$

for $\alpha$ sufficiently small.

### E.2. Numerical Experiments

We gather all the experimental results in this section. We upload the Python code for simulating the tail bounds, justifying our Proposition 5.5, and reproducing all the plots can be found in this GitHub repository.

Recall that we propose a conjecture on determining scaling function and limit distribution for general convex objectives. We first present the experiments supporting this conjecture.

#### E.2.1. CONJECTURE

To empirically verify Proposition 5.5, we consider a class of one-dimensional objective functions that are, in general, neither strongly convex nor globally $L$-smooth. Fix an integer parameter $\ell \in \mathbb{N}$ with $\ell \geq 2$. We define two families of test functions as follows.

**Definition E.1** (Test objective functions). For $\ell \in \mathbb{N}$ with $\ell \geq 2$, define $P_\ell, T_\ell : \mathbb{R} \to \mathbb{R}$ by

$$
P_\ell(x) := \frac{x^{2\ell}}{2\ell},
\tag{141}
$$

$$
T_\ell(x) := \frac{x^{2\ell}}{2\ell} + \frac{\sin^{2\ell}(x)}{2\ell}, \qquad x \in \mathbb{R}.
\tag{142}
$$

Fix $\ell \in \mathbb{N}$ with $\ell \geq 2$, and consider the objectives $P_\ell$ and $T_\ell$ defined in Definition E.1. Running constant-stepsize SGD with stepsize $\alpha > 0$ and additive noise $\{w_k\}_{k \geq 0}$ yields the one-dimensional recursion

$$
X_{k+1}^{(\alpha)} = X_k^{(\alpha)} - \alpha(X_k^{(\alpha)})^{2\ell-1} + \alpha w_k
\tag{143}
$$

$$
X_{k+1}^{(\alpha)} = X_k^{(\alpha)} - \alpha\left[ (X_k^{(\alpha)})^{2\ell-1} + \sin(X_k^{(\alpha)})^{2\ell-1}\cos(X_k^{(\alpha)}) \right] + \alpha w_k
\tag{144}
$$

Both $P_\ell$ and $T_\ell$ are nonnegative and admit the unique global minimizer $x^* = 0$. Let $Y^{(\alpha)} = \frac{X^{(\alpha)} - x^*}{g(\alpha)}$ be the centered, scaled steady state. In Proposition 5.5, the scaling is $g(\alpha) = \alpha^{1/h}$. In the present setup, the local growth order is $h = 2\ell$, hence the prediction is $g(\alpha) = \alpha^{1/(2\ell)}$. Moreover, $P_\ell^{(k)}(0) = 0$ for all $1 \leq k \leq 2\ell - 1$ and $P_\ell^{(2\ell)}(0) = (2\ell - 1)! > 0$. For $T_\ell$, the polynomial term implies the same local behavior: $T_\ell^{(k)}(0) = 0$ for $1 \leq k \leq 2\ell - 1$ and $T_\ell^{(2\ell)}(0) = (2\ell - 1)! + c_\ell$ for some $c_\ell \geq 0$. Consequently, both families satisfy Assumption 5.1 with $h = 2\ell$.

For the setup of numerical experiments, we consider two noise distributions: a light-tailed Gaussian noise and a heavier-tailed noise with finite $2h$-th moment. We define the two choices of noise as follows.

**Definition E.2** (Gaussian noise). Let $\{w_k^N\}_{k \geq 0}$ be i.i.d. with $w_k^N \sim \mathcal{N}(0, 1)$.

**Definition E.3** (Signed Pareto noise). Let $\{B_k\}_{k \geq 0}$ be i.i.d. Rademacher random variables with $\mathbb{P}(B_k = 1) = \mathbb{P}(B_k = -1) = 1/2$, and let $\{Z_k\}_{k \geq 0}$ be i.i.d. $\text{Pareto}(x_m, \beta)$ random variables with support $[x_m, \infty)$ and tail $\mathbb{P}(Z_k > z) = \left(\frac{x_m}{z}\right)^\beta$, with $z \geq x_{m,}$. We let $Z_k$ independent of $\{B_k\}$ and define $w_k^P := B_k Z_k$.

Given the definitions of these two types of noise, without loss of generality, we set the variance of noise to be 1. For pareto noise, we use $\beta = 12$ to ensure the stability of SGD, and set $x_m = \sqrt{1/1.2}$ to ensure its variance is 1.

Our numerical experiments are designed to directly test the conclusions of Proposition 5.5: (i) the scaling law $g(\alpha) = \alpha^{1/(2\ell)}$, and (ii) the stabilization of the scaled steady-state distribution under this scaling as $\alpha \downarrow 0$. In particular, we compare the proposed scaling $g(\alpha) = \alpha^{1/(2\ell)}$ against the classical $\sqrt{\alpha}$ scaling.

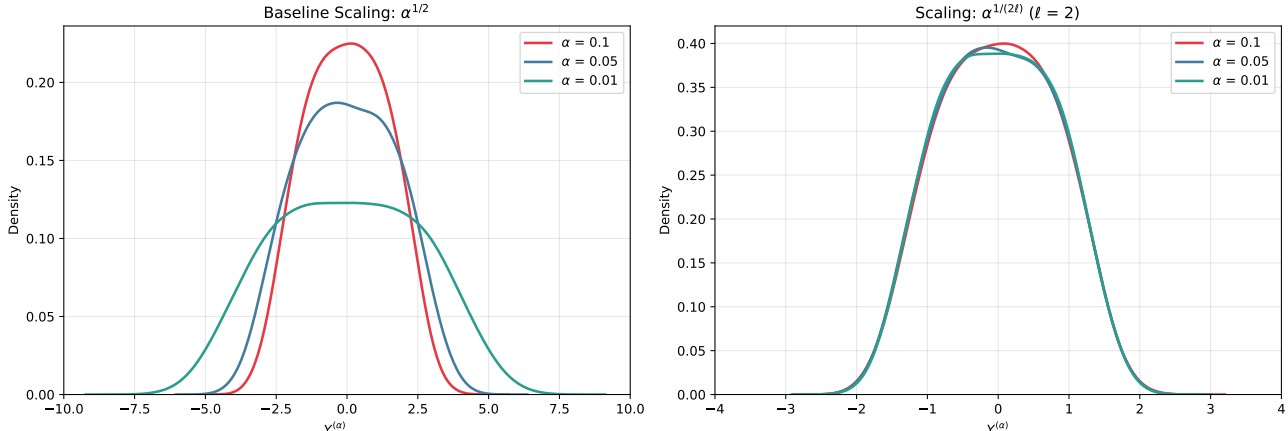

*Figure 2.* Basline versus scaling for Normal Noise

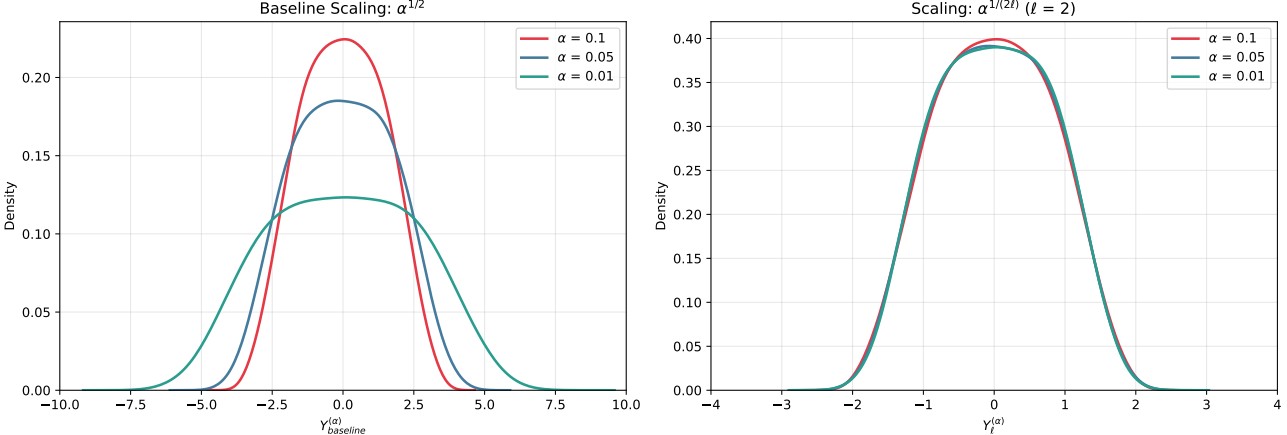

*Figure 3.* Baseline versus scaling for Pareto Noise

Figures 2–3 exhibit density estimates of the empirical distribution of $Y^{(\alpha)}$ for several values of $\alpha$, with $\ell = 2$. Under the correct scaling $g(\alpha) = \alpha^{1/4}$, the curves align closely across $\alpha$, whereas the baseline scaling $g(\alpha) = \alpha^{1/2}$ visibly does not yield convergence of density. Figures 4a–4b extend the comparison to $\ell \in \{2, 3, 4\}$ and to both objective families $P_\ell$ and $T_\ell$, using the predicted scaling $g(\alpha) = \alpha^{1/(2\ell)}$; the resulting densities are consistent across $\ell$ and across noise models. Together, these experiments provide empirical support for the scaling prediction in Proposition 5.5.

All experiments use $n = 100{,}000$ independent runs with trajectories of length $k = 1{,}000 \times \ell$. We approximate the steady-state law using the empirical distribution of the terminal iterate from each run. For flatter objectives, we increase $k$ scaled by $\ell$, since empirical evidence suggests that the time required to reach the stationary distribution grows linearly in $\ell$. As illustrated in the figures, under the correct scaling the empirical distribution appears to stabilize over time, in contrast to the incorrect scaling.

### E.2.2. CONCENTRATION ARGUMENT

We next illustrate empirically our non-uniform Berry-Essen style of tail bounds established in the previous section. As an illustrative example, we visualize the tail bounds obtained in Theorem 3.5 with 1-dimension, which can be concluded as the

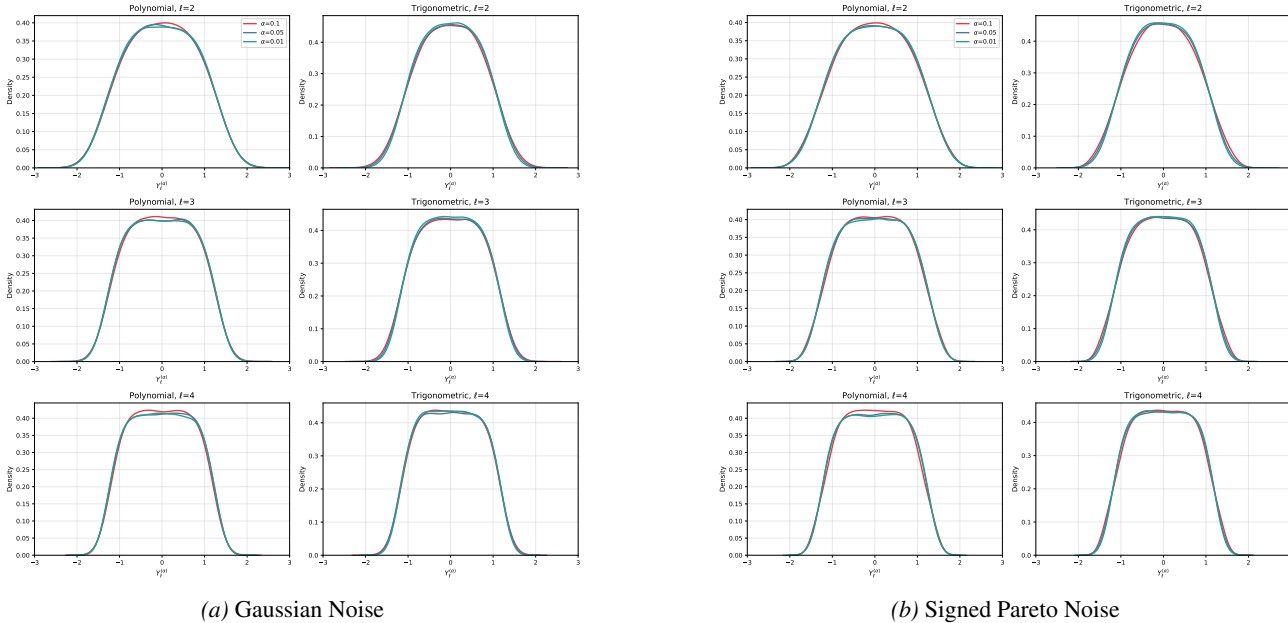

*(a)* Gaussian Noise                   *(b)* Signed Pareto Noise

*Figure 4.* Combined Results for Polynomial and Trigonometric Functions

following form

$$\left| \mathbb{P}(Y^{(\alpha)} > a) - \mathbb{P}(Z > a) \right| \leq O\left( \frac{1}{a} \sqrt{C\alpha^{\frac{1}{2}}} \right). \tag{145}$$

Though the above bound is in order analysis, we have the precise constant instead of merely Big-$O$. In the following, we use the explicit tail bounds for SGD model in Theorem 3.5 to simulate the tightness of this tail in 1-dimensional setting.

We show the sharpness of the non-uniform concentration bounds using the quadratic objective $f(x) = x^2/2$. For simulation setup, we set i.i.d. standard Gaussian noise $w_k \sim \mathcal{N}(0, 1)$. The quadratic objective function allows the constant-stepsize recursion (4) to specialize to the linear SA $X_{k+1}^{(\alpha)} = (1 - \alpha)X_k^{(\alpha)} + \alpha w_k$. Figure 5 illustrates the empirical accuracy of the non-uniform tail bounds by comparing the simulated distribution of the normalized steady state $Y^{(\alpha)}$ against the Gaussian reference. We visualize the corresponding analytic envelope implied by (145). In each panel, the horizontal axis is the threshold $a \geq 0$, and the vertical axis is the cumulative probability $\mathbb{P}(Y^{(\alpha)} \leq a)$. The red markers plot the empirical CDF of $Y^{(\alpha)}$ obtained from a long run of the SA recursion, while the black curve shows the CDF of the limiting Gaussian random variable $Z$ (as specified in Section 3.1). The blue and green curves display, respectively, the upper and lower bounds on $\mathbb{P}(Y^{(\alpha)} \leq a)$ obtained by combining (145) with the identity $\mathbb{P}(Y^{(\alpha)} \leq a) = 1 - \mathbb{P}(Y^{(\alpha)} > a)$ (and similarly for $Z$).

Across the three choices of stepsize $\alpha$, the empirical CDF lies within the predicted envelope and approaches the Gaussian benchmark as $\alpha$ decreases, consistent with the stated upper bound in (145). In particular, the gap between the upper and lower bounds shrinks as $\alpha \downarrow 0$ and is tighter for larger thresholds $a$, reflecting the non-uniform (in $a$) nature of (145). Such explicit, non-asymptotic control of tail probabilities yields an *a priori* guarantee on rare events of the steady-state fluctuation, which is particularly relevant for risk-sensitive learning and decision-making applications (cf. Section 3.1).

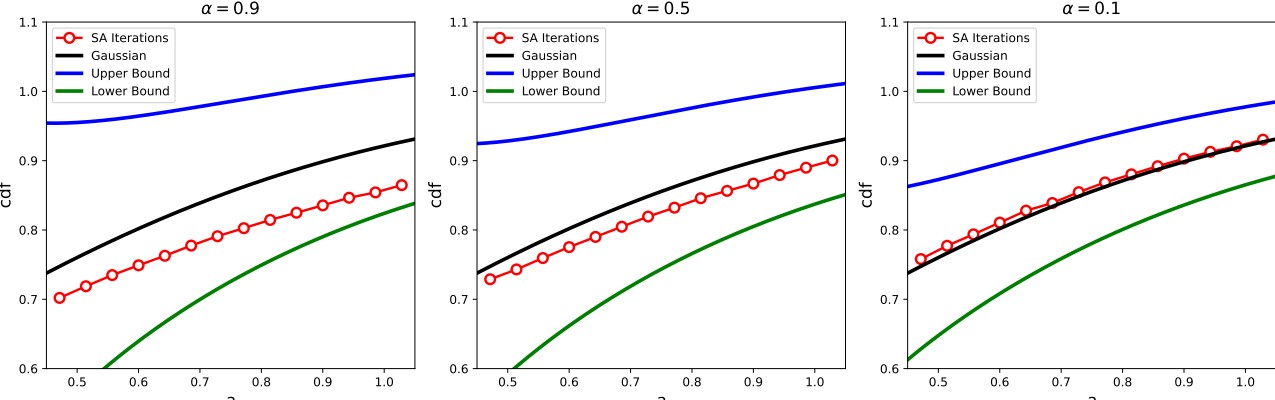

*Figure 5.* SA Iterations and Concentration Bound

