# OpenReview forum: "Steady-State Behavior of Constant-Stepsize Stochastic Approximation: Gaussian Approximation and Tail Bounds"
_ICML.cc/2026/Conference — ICML 2026 regular_

### Official Review · Reviewer_u59R · 2026-03-12

**Soundness:** 3
**Presentation:** 3
**Significance:** 3
**Originality:** 3
**Overall Recommendation:** 5
**Confidence:** 3

**Summary:**

This paper studies the steady-state behavior of constant-stepsize stochastic approximation (SA) as the step size converges to zero. The authors focus on three settings: stochastic gradient descent on smooth and strongly convex objectives, linear SA, and contractive nonlinear SA under both i.i.d. and Markovian noise, and establish explicit upper bounds in Wasserstein distance and Berry-Esseen type tail bound. Moreover, this paper also discusses SGD for non-strongly convex objectives and provides evidence to show that the limit is not Gaussian.

**Compliance With Llm Reviewing Policy:**

Affirmed.

**Final Justification:**

The authors' rebuttal addresses my concerns and I believe this paper makes a contribution to the study of stochastic approximation.

**Key Questions For Authors:**

1. In terms of the proof approach, has Stein's method been applied in previous studies of SA? It seems that Stein's method is a classical method. What is the main novelty and difficulty when applying this method to the study of this paper? Moreover, when applying the Poisson equation technique, what is the main difference compared with previous work (e.g., the reference in Section 4)?

2. Could the approach in this paper be generalized to the case of decreasing step size?

3. In the non-uniform Berry-Esseen bound, the error term is propto $1/a$.When $a$ is close to zero, the error term is very large. Could this dependence $a$ be improved?

**Limitations:**

Yes.

**Strengths And Weaknesses:**

[strengths]
1. This paper is well-organized and easy to follow.
2. This paper considers the combiniation of three settins and two types of noise, and is very rich in scope.
3. Although I do not have sufficient time to verify the details of the very long proofs, I believe the theoretical results are reliable.
4. Compared with previous work, this paper establishes explicit non-asymptotic upper bound with clear dependence on the parameters.

[weaknesses]
1. This paper only controls the distance between the steady-state distribution and the Gaussian limit as the step size converges to zero, but does not characterize the distance between the current scaled iterate and the steady-state distribution. This limits the significance of this paper to some extent.
2. The second part of Section 1.2 (``Our Appoarch'') applies the Poisson equation technique. However, this technique is widely-used in the study of SA under the Markovian noise.

---

> ### Author Rebuttal · Authors · 2026-03-31
>
> Thank you for your comments! Please find our response below.
>
> **Comment: does not characterize the distance between the current scaled iterate and the steady-state distribution.**
>
> We agree that controlling finite time $d_W(Y_k^{(\alpha)}, Y^{(\alpha)})$ is practically important. We discuss this point in our response to Reviewer dVRF (see Comment: The results do not provide any convergence rates), including how finite-time bounds combine with our steady-state result.
>
>
> **Question: has Stein's method been applied in previous studies of SA? What is the main novelty and difficulty?**
>
> Yes, Stein's method has been used in some SA-related works [1], but in a different way. Existing works typically apply Stein's method to a scaled sum, e.g., $\frac{1}{\sqrt{n}} \sum_{i=1}^n \xi_i$, together with Lindeberg arguments. Our setting is different. We study the steady-state law of constant-stepsize SA, which has no explicit summation form. Hence, classical Stein arguments for sums do not directly apply. Our main novelty is to use Stein's method via generator comparison. Though generator-based Stein's method has been used in queueing theory [2], those works do not handle Markovian noise. Thus, the generator-based Stein plus Poisson Equation framework is novel in the SA setting.
>
> The main difficulty for our application is that the recursion is nonlinear and its stationary law is unknown. Addressing this requires a careful combination of Stein equation estimates, moment bounds, and stability arguments. Meanwhile, standard alternatives are not available here, e.g., Stein kernels require explicit density information and direct quantile coupling is unclear and essentially one-dimensional. This is why generator comparison is crucial in our setting.
>
> > [1] Srikant, R. *Rates of convergence in the central limit theorem for Markov chains, with an application to TD learning*. *Mathematics of Operations Research*, 2025.
> >
> > [2] Braverman, A. and Dai, J. G. *Stein’s method for steady-state diffusion approximations of M/Ph/n+M systems*, 2017.
>
>
> **Question: when applying the Poisson equation, what is the main difference?**
>
> Poisson equation is a standard tool for analyzing Markovian noise, which rewrites the noise into a martingale term plus telescoping remainders. This idea is exploited in SA literature, e.g., to apply an ODE argument for SA convergence [3] or bias decomposition [4]. In our paper, however, we combine the Poisson equation with the generator comparison in Stein's method to derive Wasserstein error bound. Particularly, this requires applying the Poisson equation twice (see our treatment of $W_1$ and $W_2$ in Section D.1.2). To the best of our knowledge, such usage of Poisson equation is novel in the SA literature.
>
> > [3] Benveniste, A., Métivier, M., and Priouret, P. *Adaptive Algorithms and Stochastic Approximations*. Applications of Mathematics, Vol. 22. Springer, Berlin, Heidelberg, 1990.
> >
> > [4] Haque, S. U. and Maguluri, S. T. *Stochastic Approximation with Unbounded Markovian Noise: A General-Purpose Theorem*. arXiv preprint arXiv:2410.21704, 2024.
>
>
> **Question: generalized to the case of decreasing step size?**
>
> It is unclear how to extend our generator comparison directly to diminishing stepsizes. Our method relies on a fixed-stepsize, time-homogeneous recursion and its stationary law. Under diminishing stepsizes, the chain is time-inhomogeneous, so the same generator construction is no longer available.  By contrast, [5] uses a blockwise argument by grouping the noise over growing blocks and compares to discretized OU process. The key requirement is that the cumulative stepsize on each group vanishes even as the group size grows, which relies on stepsize $\alpha_k \to 0$. Thus, their technique is specific to the diminishing-stepsize regime.
>
> > [5] Kong, S. T. and Srikant, R. *Finite-Sample Wasserstein Error Bounds and Concentration Inequalities for Nonlinear Stochastic Approximation*. arXiv preprint arXiv:2602.02445, 2026.
>
>
> **Question: Could this dependence on $a$ be improved?**
>
> We agree that the $1/a$ factor becomes large when $a$ is small. This is a standard limitation of non-uniform Berry-Esseen bounds as they are mainly designed for large deviation. For small $a$, one can use Kolmogorov distance $d_K(X, Y) = \sup_{a \in \mathbb{R}} |\mathbb{P}(X \leq a) - \mathbb{P}(Y \leq a)|$ which is uniform in $a$, and take the minimum of the Kolmogorov and our non-uniform bound. A sharper improvement would require $\mathcal{W}_p$ approximation for $p > 1$, leading to $O(\sqrt{\alpha}) \Phi^c(a)$ error instead of the current $O(\alpha^{1/4})/a$ through argument in Section C.5 [6]. However, to the best of our knowledge, such result has not been established even for martingale CLT, and would require substantially stronger techniques.
>
> > [6] Wang, Z. and Maguluri, S. T. *Tail Bounds for Queues with Abandonment: Constant, Moderate, Large Deviations, and Efficient Concentration*. arXiv preprint arXiv:2603.14163, 2026.

---

> > ### Author Rebuttal · Reviewer_u59R · 2026-04-03
> >
> > Thanks for the detailed responses and I will raise my score to 5.

---

> > > ### Author Response · Authors · 2026-04-03
> > >
> > > We thank the reviewer for the valuable questions and comments, and we are glad that our responses addressed the concerns. If the reviewer has any further questions or follow-up comments, we would be very happy to continue the discussion.

---

### Official Review · Reviewer_dVRF · 2026-03-12

**Soundness:** 3
**Presentation:** 3
**Significance:** 2
**Originality:** 2
**Overall Recommendation:** 4
**Confidence:** 2

**Summary:**

This paper provides the non-asymptotic error bounds for approximating the stationary distribution of constant-stepsize stochastic approximation algorithms by their Gaussian limits. For SGD on smooth strongly convex objectives, linear SA, and contractive nonlinear SA (with both i.i.d. and Markovian noise), the authors derive explicit Wasserstein distance bounds between the centered-and-scaled steady state and its Gaussian limit, with errors vanishing as stepsize tends to zero. They also establish tail probability comparisons with explicit error terms decaying in both deviation level and stepsize.

**Compliance With Llm Reviewing Policy:**

Affirmed.

**Final Justification:**

see the discussions.

**Key Questions For Authors:**

N/A

**Limitations:**

Yes

**Strengths And Weaknesses:**

Strengths

- Under standard smoothness assumptions for SA algorithms, Chen et al. (2022) established that the centered and scaled steady state  converges weakly to a Gaussian distribution.  This paper derives explicit Wasserstein distance bounds between the centered-and-scaled steady state and its Gaussian limit, and also establishes tail probability comparisons with explicit error terms decaying in both deviation level and stepsize.

Weakness
- The results require 3-rd moment noise condition and Lipschitzness of the Hessian matrix.
- The results do not provide any convergence rates.
- The paper is rather technical and may be better suited to a probability journal.

---

> ### Author Rebuttal · Authors · 2026-03-31
>
> Thank you for your comments! Please find our response below.
>
> **Comment: The results require 3-rd moment noise condition and Lipschitzness of the Hessian matrix.**
>
> We agree that these assumptions are stronger than those needed for an asymptotic CLT. They are only needed to obtain an explicit non-asymptotic Wasserstein bound.
>
> The third-moment condition is standard in quantitative Gaussian approximation. In Berry--Esseen theory, finite second moments yield convergence in distribution, whereas an explicit rate of convergence typically requires $\mathbb{E}[|\xi_i|^3] < \infty$ [1]. The same issue appears here. After matching the drift and covariance, the remaining one-step generator error is third order, so its control naturally depends on a third-moment bound. This requirement is essentially necessary in the classical i.i.d. setting [2].
>
> The Lipschitz Hessian assumption is a standard smoothness condition for controlling the third-order Taylor remainder [3, H6]. This is what enables an explicit pre-limit rate. Our proof only needs this condition locally around $x^{\star}$. In particular, it holds whenever $f$ is $C^3$ with bounded third derivative in a neighborhood of $x^{\star}$, so it is mild and practically reasonable.
>
> > [1] Petrov, V. V. *Sums of Independent Random Variables*. Springer, 2012.
> >
> > [2] Rio, E. *Upper Bounds for Minimal Distances in the Central Limit Theorem*. *Annales de l'Institut Henri Poincaré, Probabilités et Statistiques* 45(3): 802--817, 2009.
> >
> > [3] Bach, F. and Moulines, E. *Non-Asymptotic Analysis of Stochastic Approximation Algorithms for Machine Learning*. In *Advances in Neural Information Processing Systems 24 (NeurIPS)*, 2011.
>
>
> **Comment: The results do not provide any convergence rates.**
>
> The rate in our paper is a distributional convergence rate quantified in the Wasserstein distance. Particularly, while prior work [4] only provides convergence in distribution, i.e., $Y^{(\alpha)} \xrightarrow{d} \mathcal{N}(0, \Sigma_Y)$, our result gives the quantitative bound $W_1(Y^{(\alpha)}, \mathcal{N}(0, \Sigma_Y)) \leq O(\alpha^{\frac{1}{2} - \epsilon})$ for any $\epsilon > 0$.
>
> In case, if the reviewer's concern is the convergence rate to steady state as $k \to \infty$, we supply finite time convergence results, i.e., $d_W(Y_k^{(\alpha)}, Y^{(\alpha)})$ for $k \geq 1$. We can use an one-step contraction arguments on the SA iteration (1) to obtain a
>  geometric convergence $d_W\left(Y_k^{(\alpha)}, Y^{(\alpha)}\right) \leq O( (1-\Omega(\alpha))^k)$.
> Such finite-time convergence results also exist for specific models in the literature, e.g., [5, Theorem 4.1] for multiplicative linear SA model.   Moreover, combined with our main result, we can obtain a finite-time version of the Wasserstein error,
> $d_W(Y_k^{(\alpha)}, \mathcal{N}(0,\Sigma_Y))
> \leq O((1-\Omega(\alpha))^k) + O(\alpha^{1/2-\epsilon})$. Therefore, $k \asymp \alpha^{-1}\log(1/\alpha)$ already makes the finite time error same order compared to the steady-state Gaussian approximation error, which provides an early stopping rule. After that point, more iterations do not significantly improve the Gaussian approximation. If helpful, we can add this discussion and point to our supplementary finite-time result.
> > [4] Chen, Z., Mou, S., & Maguluri, S. T. Stationary Behavior of Constant Stepsize SGD Type Algorithms: An Asymptotic Characterization. *Proc. ACM Meas. Anal. Comput. Syst.*, 6(1). (2022)
> >
> > [5] Huo, D., Chen, Y., and Xie, Q. *Bias and Extrapolation in Markovian Linear Stochastic Approximation with Constant Stepsizes*. In *Abstract Proceedings of the 2023 ACM SIGMETRICS International Conference on Measurement and Modeling of Computer Systems*, pp. 81--82, 2023.
>
>
> **Comment: The paper is rather technical and may be better suited to a probability journal.**
>
> We agree that the paper is technical. That said, we believe it is appropriate for ICML because the technical tools are developed to answer machine learning questions. We also expect the analytical framework to be reusable, since the Stein-based approach can be applied to other ML stochastic algorithms. Since the questions we address in this paper, i.e., non-asymptotic characteristics, are central to the theory of modern stochastic optimization, we believe the paper fits ICML well.

---

> > ### Author Rebuttal · Reviewer_dVRF · 2026-04-04
> >
> > Thank you for your response. I am willing to raise my score.

---

> > > ### Author Response · Authors · 2026-04-05
> > >
> > > We thank the reviewer for the valuable comments, and we are glad that our responses addressed the concerns. If the reviewer has any further questions or follow-up comments, we would be very happy to continue the discussion.

---

### Official Review · Reviewer_fgvD · 2026-03-13

**Soundness:** 3
**Presentation:** 3
**Significance:** 3
**Originality:** 3
**Overall Recommendation:** 5
**Confidence:** 4

**Summary:**

This paper studies the steady-state behavior of constant-stepsize stochastic approximation (SA), providing explicit non-asymptotic Gaussian approximation bounds for three algorithm classes: SGD on strongly convex objectives, linear SA, and contractive nonlinear SA, under both i.i.d. and Markovian noise. The central contributions are Wasserstein distance bounds between the centered-scaled stationary distribution and its Gaussian limit, alongside non-uniform Berry-Esseen-style tail bounds that decay in both the stepsize and deviation level. The analysis combines generator-based Stein's method with Poisson equation techniques to handle Markovian temporal dependence, and additionally resolves a conjecture from prior work. The paper further extends the framework to general convex objectives, identifying a non-Gaussian Gibbs limiting distribution and a corrected scaling exponent, with numerical experiments supporting the theoretical findings.

**Compliance With Llm Reviewing Policy:**

Affirmed.

**Final Justification:**

I believe the paper makes a solid contribution to existing Gaussian approximation literature and addresses some of the unresolved tasks I raised. I will therefore increase my score.

**Key Questions For Authors:**

1. **On lower bounds.** The paper establishes upper bounds on the Wasserstein distance between the steady-state distribution and its Gaussian limit, but does not provide matching lower bounds. Can the authors comment on whether lower bounds of the same order are achievable?

2. **On covariance estimation.** Have the authors considered any procedures for estimating this asymptotic covariance from the iterates themselves, for instance, through plug-in estimators or batch mean methods? Without such a procedure, it is unclear how practitioners would construct confidence intervals or other interval estimators that fully exploit the Gaussian approximation result.

3. **On admissible stepsizes.** Do the authors have any practical recommendations for how a practitioner might determine whether a given stepsize is admissible, without direct knowledge of these constants? For example, could diagnostic tools based on the iterates themselves be used to verify that the required conditions are approximately satisfied?

**Strengths And Weaknesses:**

## Strength
1. **Resolves an open conjecture.** The paper settles the uniqueness conjecture from Chen et al. (2022) regarding the limiting distribution of the centered-scaled steady state, which had been left open in prior work.

2. **Explicit and actionable constants.** Unlike many asymptotic results, the error bounds carry fully explicit dependence on stepsize, dimension, and problem parameters, making them directly useful for algorithm design and tuning in practice.

3. **Broad coverage of SA models.** The results apply to three practically relevant algorithm families under a unified framework, rather than being tailored to a single special case.

4. **Markovian noise extension.** The paper handles temporally correlated noise via Poisson equation techniques, covering the realistic setting of reinforcement learning where data arrives sequentially along trajectories.

5. **Non-uniform tail bounds.** The Berry-Esseen style tail bounds improve with the deviation level, which is particularly well-suited for risk-sensitive applications where one cares about rare event probabilities.

6. **Non-Gaussian regime.** The extension to general convex objectives identifies the correct scaling and a non-Gaussian Gibbs limiting law, going meaningfully beyond the classical strongly convex setting.

7. **Numerical validation.** Experiments support both the conjectured scaling laws and the tightness of the tail bounds, providing empirical grounding for the theoretical claims.

## Weakness

1. **Reliance on unproven conjectures.** The results for general convex objectives (Proposition 5.5) depend on two conjectures regarding moment bounds and Stein solution regularity that remain unverified, limiting the completeness of the theoretical contribution in that setting.

2. **Logarithmic factor in Wasserstein bounds.** The extra logarithmic factor in the Wasserstein bounds is noted as a technical artifact of the high-dimensional Stein's method framework. It is unclear whether the logarithm can be removed in the multivariate setting without substantially new techniques.

3. **No asymptotic covariance estimation procedure.** The paper does not provide a systematic way to estimate the asymptotic covariance matrix that appears in the Gaussian limit, which is necessary for constructing practical confidence intervals or interval estimators for the steady-state fluctuation. The Lyapunov equation characterizing this covariance depends on the Hessian at the true solution and the noise covariance, both of which are generally unknown in practice. Without addressing this estimation gap, the theoretical Gaussian approximation results are difficult to translate into usable statistical inference procedures.

4. **Restrictive stepsize conditions.** The theoretical guarantees of the paper require the stepsize $\alpha$ to be sufficiently small, falling below problem-dependent thresholds that depend on quantities such as the strong convexity constant and smoothness parameter of the objective. In practice, these thresholds are rarely known in advance, and the regime in which the bounds are guaranteed to apply may be difficult to identify without precise knowledge of the problem constants. Furthermore, the explicit constants appearing in the Wasserstein and tail bounds themselves depend on these same unknown quantities, compounding the difficulty of translating the theoretical results into concrete stepsize selection guidance. This creates a gap between the theoretical framework and practical deployment.

---

> ### Author Rebuttal · Authors · 2026-03-31
>
> Thank you for your comments! Please find our response below.
>
> **Comment: Reliance on unproven conjectures.**
>
> We agree that Proposition 5.5 is conditional on two conjectures. The main missing step is Conjecture 5.4 on Stein solution regularity. Existing Stein-factor results for multivariate Gibbs laws rely on strong log-concavity [1, Theorem 2.1] or dissipativity [2, Assumption 2.1], whereas our polynomial Gibbs potential satisfies neither condition. Therefore, resolving Conjecture 5.4 would require genuinely new techniques.
>
> At the same time, our conjecture already sharpens prior work [3, Condition 3.1]. In particular, it identifies both the Gibbs limit and the scaling from the proof itself, rather than from numerical observation alone. In this sense, the conjectured Stein-factor bound is the remaining missing piece needed to complete the proof.
>
> > [1] Mackey, L., & Gorham, J. Multivariate Stein factors for a class of strongly log-concave distributions.  (2016)
> >
> > [2] Fang, X., Shao, Q.-M., & Xu, L. Multivariate approximations in Wasserstein distance by Stein's method and Bismut's formula. *Probability Theory and Related Fields*, 174(3), 945–979. (2019)
> >
> > [3] Chen, Z., Mou, S., & Maguluri, S. T. Stationary Behavior of Constant Stepsize SGD Type Algorithms: An Asymptotic Characterization. *Proc. ACM Meas. Anal. Comput. Syst.*, 6(1). (2022)
>
>
> **Question: On lower bounds.**
>
> We first justify the sharpness in the limit parameter $\alpha$. In one dimension, the $\sqrt{\alpha}$ rate is already sharp in the simplest quadratic case. For SA with $F(x)=-x$ (Equation 1), $Y^{(\alpha)}$ becomes a weighted sum of independent noises, and classical Edgeworth expansions imply $d_W\left(Y^{(\alpha)}, Z\right)=\Omega(\sqrt{\alpha})$ [4]. Thus, the rate cannot generally be improved, and our one-dimensional upper bound in Section C.9 is sharp. The same example also suggests a lower bound of $\Omega(\sqrt{\alpha})$ in the high-dimensions.
>
> The logarithmic factor comes from the multivariate Stein. This is consistent with existing multivariate CLT upper bounds, where logarithmic losses also appear in Markov-chain CLTs [5]. We therefore believe removing the logarithm would require genuinely new techniques. We do not prove a matching multivariate lower bound, so whether this logarithmic factor is tight remains open.
>
> Turning to the dimension $d$, Kantorovich duality for Wasserstein distance (Equation 3) reduces the high-dimensional lower bound to a one-dimensional lower bound. Concretely, we can construct a multivariate test function
> $
> f(x)=\frac{1}{\sqrt{d}}\sum_{i=1}^d f_i(x_i),
> $ where each $f_i$ is a one-dimensional 1-Lipschitz function.
> Since the dual formulation takes the supremum over all $1$-Lipschitz functions, this yields $d_W(Y^{(\alpha)}, Z)=\Omega(\sqrt{d})$. Thus, some polynomial dependence on $d$ is essential.
>
> > [4] Petrov, V. V. *Sums of Independent Random Variables*. Springer Science. (2012).
> >
> > [5] Srikant, R. *Rates of convergence in the central limit theorem for Markov chains, with an application to TD learning*. Mathematics of Operations Research. (2025).
>
> **Question: On covariance estimation.**
>
> We agree that the long-run covariance is important for a practical confidence interval estimator. In our model, the covariance $\Sigma_M$ can be estimated by first computing $\hat{\xi_k}=\alpha^{-1}(X_{k+1}^{(\alpha)}-X_k^{(\alpha)})-F(X_k^{(\alpha)})$.
> One may then apply a HAC estimator to the empirical autocovariances of $\hat{\xi_k}$. In addition, [6] develops a sequential estimator for $\Sigma_M$ with finite-sample guarantees.
>
> Given such an estimator, the Lyapunov equation determines $\Sigma_Y$. Our theorems then provide Gaussian approximation error bounds in both Wasserstein distance and non-uniform Berry-Esseen form.
>
> > [6] Agrawal, S., A, P. L., and Maguluri, S. T. *Policy Evaluation for Variance in Average Reward Reinforcement Learning*. In *Proceedings of the 41st International Conference on Machine Learning (ICML)*, pp. 471--502. PMLR, 2024.
>
>
> **Question: On admissible stepsizes.**
>
> We agree that the threshold $\alpha \leq \alpha_0$ is problem-dependent and generally unavailable in practice.
> A practical approach is to combine stepsize selection with an iterate-based diagnostic. For example, line-search methods can be used to choose a candidate stepsize, and a Stein discrepancy diagnostic can then be used to test whether the centered-scaled iterates are already close to Gaussian for that stepsize and finite run [7].
>
> These procedures use only the iterates and do not require the unknown constants. The diagnostic targets Gaussian approximation rather than stationarity, and thus complements standard stability diagnostics.
>
> > [7] Anastasiou, A., Barp, A., Briol, F.-X., Ebner, B., Gaunt, R. E., Ghaderinezhad, F., Gorham, J., Gretton, A., Ley, C., Liu, Q., et al. *Stein's method meets computational statistics: A review of some recent developments*. Statistical Science, 38(1):120--139, 2023.

---

> > ### Author Rebuttal · Reviewer_fgvD · 2026-04-02
> >
> > Thanks for the thoughtful responses to the questions I asked. I believe the paper makes a solid contribution to existing Gaussian approximation literature and addresses some of the unresolved tasks I raised. I will therefore increase my score.

---

> > > ### Author Response · Authors · 2026-04-03
> > >
> > > We thank the reviewer for the follow-up and for recognizing the contribution of our work. We are grateful for the valuable questions and comments, and we are glad that our responses addressed some of the concerns. If the reviewer has any further questions or follow-up comments, we would be very happy to continue the discussion.

---

### Decision · Program_Chairs · 2026-04-30

**Decision:**

Accept (regular)

**Comment:**

This paper provides explicit non-asymptotic Gaussian approximation bounds for the stationary distribution of constant-stepsize stochastic approximation algorithms, covering SGD on strongly convex objectives, linear SA, and contractive nonlinear SA under both i.i.d. and Markovian noise. The technical approach was recognized as novel in the SA context, particularly the double application of the Poisson equation and the generator comparison framework for analyzing stationary laws without explicit density information.All three reviewers marked their concerns as fully resolved and raised or maintained their scores. The paper is technically rigorous, well-organized, appropriately scoped for the ICML audience, and makes a clear contribution to the quantitative theory of stochastic optimization that other researchers can build upon.